

# Reactivity study of 3,3-dimethylbutanal and 3,3-dimethylbutanone: Kinetic, reaction products, mechanisms and atmospheric implications

Inmaculada Aranda[1], Sagrario Salgado[1,2], Beatriz Cabañas[1,2], Florentina Villanueva[2] and Pilar Martín[1,2*]

[1]Universidad de Castilla-La Mancha, Departamento de Química Física, Facultad de Ciencias y Tecnologías Químicas, Avda. Camilo José Cela s/n, 13071, Ciudad Real, Spain.

[2]Universidad de Castilla-La Mancha, Instituto de Investigación en Combustión y Contaminación Atmosférica (ICCA), Camino de los Moledores s/n, 13071, Ciudad Real, Spain.

*Corresponding author. Email address: mariapilar.martin@uclm.es*

**Abstract:** 3,3-dimethylbutanal and 3,3-dimethylbutanone are carbonyl compounds that could play a key role in tropospheric chemistry. To better understand the effects of carbonyl compounds in the atmosphere, a kinetic and mechanistic study was conducted on the degradation of 3,3-dimethylbutanal and 3,3-dimethylbutanone with atmospheric oxidants (Cl atoms, OH and $NO_3$ radical). The kinetic experiments were performed at $710 \pm 30$ Torr and at room temperature ($298 \pm 5$ K) using a relative method and FTIR (Fourier Transform Infrared Spectroscopy) to monitor the reactions. The rate coefficients (k in units of $cm^3$ $molecule^{-1}$ $s^{-1}$) obtained were: $k_{Cl+33DMbutanal} = (1.27 \pm 0.08) \times 10^{-10}$, $k_{Cl+33DMbutanone} = (4.22 \pm 0.27) \times 10^{-11}$, and $k_{OH+33DMbutanone} = (1.25 \pm 0.05) \times 10^{-12}$. The reaction products were also determined using FTIR and GC-MS (Gas Chromatography/Mass Spectrometry). The main products observed were short carbonyl compounds, including acetone, formaldehyde and 2,2-dimethylpropanal. In the presence of NO, nitrated compounds are formed, and in large $NO_2$ concentrations peroxyacetyl nitrate (PAN) and peroxy-3,3-dimethylbutyryl nitrate were clearly identified. Other unquantified compounds were multifunctional organic compounds and organic acid of low volatility. Both 33DMbutanal and 33DMbutanone degrade rapidly near emission sources with minimal impact on radiative forcing. However, they may contribute to tropospheric ozone, with a range of $POCP_E$ of 15-69, and secondary organic aerosol formation, potentially worsening air quality and contributing to photochemical smog**.**



## 1 Introduction

Carbonyl compounds are a group of oxygenated volatile organic compounds (OVOCs) that are emitted into the atmosphere from natural and anthropogenic sources (Bao et al., 2022), but they are also formed in the atmosphere as oxidation products of other volatile organic compounds (VOCs) (Mellouki et al., 2015). It is well stablished that OVOCs play an important role in the sequence of chemical reactions that leads to their further oxidation and contributes to the tropospheric ozone formation in polluted and remote environments with important effect on health as is the case of formaldehyde and acetaldehyde (Calvert et al., 2011, Liu et al., 2022, Mellouki et al., 2015, Zhou et al., 2023). In addition, large carbonyl compounds can influence on climate change if they are strong infrared light absorbers by altering the Earth's radiation balance and being an important source of aerosol which could further affect radiation balance and be hazardous for health (Heald and Kroll, 2020, Liu et al., 2022;).

The rising $O_3$ levels in mega-city clusters like Chinese cities underscore the critical need for effective control of ambient carbonyls, significant precursors of $O_3$. Moreover, as intermediate products of hydrocarbon oxidation, carbonyls likely play a pivotal role in minimizing the disparity between atmospheric reactivity in measurements and simulations. Previous studies (Calvert et al., 2011, Liu et al., 2022, Mellouki et al., 2015, Zhou et al., 2023) have provided valuable insights into carbonyls' presence, composition, origins, and impact on $O_3$ and SOA formation, using a combination of field measurements, numerical simulations, and laboratory experiments. Nevertheless, further research is still warranted to achieve a more comprehensive understanding of carbonyls' sources and sinks, given the complexity of their emission and degradation processes (Liu et al., 2022).

In this work, the tropospheric reactivity of two carbonyls compounds whose reactivity is not yet completely established, has been studied: 3,3-dimethylbutanal (33DMButanal) and 3,3-dimethylbutanone (33DMButanone). These two carbonyls are among the reaction products identified in the atmospheric degradation of two alcohols (3,3-dimethyl-1-butanol and 3,3-dimethyl 2-butanol), which reactivity have been previously studied (by our research group) (Colmenar et al 2020). On the other hand, 3,3-dimethylbutanal has also been detected as reaction product in the reaction of 2,4,4-trimethyl-1-pentanol with Cl atoms (Vila et al., 2020) and it could be an intermediate in the synthesis of neotame, a sweetener (Tanielyan and Augustine, 2012). Industrially, 33DMbutanone, known as methyl tert-butyl ketone, is produced for use in fungicides, herbicides and pesticides (Liu et al., 2022) and it might also be a solvent for the extraction of methylphenols from wastewater (Xiong et al. 2018). Specifically, in the study of Byrne et al. (2018) 33DMbutanone has been identified as potential replacements for hazardous volatile non-polar solvents such as toluene, due to low toxicity and good solvation characteristics. In addition to direct emissions, 33DMbutanone could be present in the atmosphere as a reaction product of the gas-phase oxidation of 2,2-dimethylbutane (Jenkin et al., 1997, Saunders et al., 2003) and 3,3-dimethyl-2-butanol (Colmenar et al. 2020).

Specifically, for 33DMButanal and 33DMbutanone few studies about their atmospheric reactivity have been reported in the literature. In the case of the reaction of 33DMbutanal with OH radical, the experimental rate coefficient has been measured by Aschmann et al. (2010) and D'Anna et al., (2001). Only one study on the reaction products with OH radicals has been reported by Aschmann et al., (2010). For the reaction of 33DMButanal with $NO_3$ radical two kinetic studies are available in the literature (D'Anna et al., 2001, D'Anna and Nielsen 1997). Tadic et al., 2012 has reported the photochemical parameters of 33DMbutanal due to the importance of the photodissociation of aldehydes in the atmosphere since it could represent an important source of free radicals. To our knowledge, there are no data about the reaction of 33DMbutanal with Cl atoms.



In the case of 33DMButanone only a kinetic study with Cl atoms has been carried out (Farrugia et al., 2015) and
two studies with OH radicals (Mapelli et al., 2023; Wallington and Kurylo 1987). In the OH studies, the rate
coefficient has been obtained at different temperatures and at low pressure, using absolute methods. No studies have
been carried out on the products of Cl atom and OH radical reaction with 33DMButanone that could help to stablish
the reaction mechanisms.
Taking the above into consideration, the aim of this work is to complete the studies about the reactivity of
33DMButanal and 33DMButanone to further understand their atmospheric chemistry in particular and the carbonyls
in general. For this purpose, the kinetic study has been conducted for the reactions of 33DMButanal and
33DMButanone with Cl atoms and 33DMButanone with OH radicals using a relative method and FTIR technique
as detection system. Additionally, for the reactions of 33DMButanal with Cl atoms, OH and $NO_3$ radicals and for
the reactions of 33DMButanone with Cl atoms and OH radicals a complete reaction product study has been
performed using FTIR and GC-TOFMS techniques. This work is to date the first kinetic study reported in
bibliography for the reaction of 33DMButanal with Cl atoms and the first study on reaction products and mechanisms
for the reactions of 33DMButanone with Cl atoms and OH radicals, and 33DMButanal with Cl atoms and $NO_3$
radicals. Additionally, this work includes a study on the reaction products for the reaction of 33DMButanal with OH
radicals in order to confirm the mechanism proposed by Aschmann et al., (2010).
**2 Experimental Section**
**2.1 Rate coefficients determination: relative method**
Rate coefficients have been determined using a relative rate method on the assumption that the organic compound
(carbonyl: 33DMButanal or 33DMbutanone), and the reference compound (R) are removed solely by their reactions
with the oxidants (Ox: Cl or ·OH):

87             Carbonyl + Ox→ Products ($k_{carbonyl}$)                       (1)

88             R + Ox→ Products ($k_R$)                             (2)

where $k_{carbonyl}$ and $k_R$ are the rate coefficients of the carbonyl and the reference compound, respectively. The kinetic
treatment for the reactions (1) and (2) gives the following relationship:
$$\ln\left(\frac{[carbonyl]_0}{[carbonyl]_t}\right) = \frac{k_{carbonyl}}{k_R} \times \ln\left(\frac{[R]_0}{[R]_t}\right) \qquad (I)$$

where $[carbonyl]_0$, $[R]_0$, $[carbonyl]_t$, $[R]_t$, are the initial concentrations and those at time t for the carbonyl and the
reference compound, respectively. At least three reference compounds were employed for each studied reaction, and
the experiments were performed in triplicate for each one. According to eq. (I), a plot of $\ln([carbonyl]_0/[carbonyl]_t)$
versus $\ln([R]_0/[R]_t)$ (or a proportional property) should give a linear fit with an intercept equal to zero. The slope of
the plot corresponds to the ratio of the rate coefficients ($k_{carbonyl}/k_R$). Therefore, the value of $k_{carbonyl}$ can be determined
if the rate coefficient of the reference compound ($k_R$) is known.
**2.2 Experimental systems and procedure**



Kinetic and product studies were performed at room temperature (298 ± 5 K) and atmospheric pressure (710 ± 18
Torr) employing a 50 L Pyrex® glass cell as reaction chamber coupled to a FTIR spectrometer (Thermo, Nicolet
6700). Additionally, for products identification, samples of the reactions were taken using a Solid Phase
MicroExtration fiber (SPME) as preconcentration sampling method and carried to a Gas Chromatograph coupled to
a Mass Spectrometer with Time Of Flight analyzer (GC-MSTOF) (AccuTOF GCv Jeol). Experimental details can
be found in previous publications (Aranda et al., 2020, Aranda et al., 2024, Colmenar et al., 2018, 2020a, 2020b).
Therefore, only a brief description is provided.
For FTIR experiments, the Pyrex® gas cell used contains a multireflection system that allows a maximum optical
path of 200 m (Saturn Series Multi-Pass cell). For the FTIR spectra collection, a total of 60 interferograms were co-
added over 98 s, usually taken in the range of 650-4000 cm$^{-1}$ with a resolution of 1 cm$^{-1}$. The passive sampling was
carried out using a Grey SPME fiber (DVB/CAR/PDMS). After the adsorption process (5-8 minutes of adsorption),
the fiber was taken to the injection port of the gas chromatograph (GC), where the compounds were desorbed at 250
ºC, separated and detected by GC-MSTOF. Two different capillary columns with the same characteristics were used:
a TRB-1701 (Teknokroma, 30 m × 0.32 mm × 1 µm) and a Equity$^{TM}$ – 1701 (Supelco, 30 m × 0.32 mm × 1 µm).
Once in the mass spectrometer, the compounds were ionized (Electron Ionization (EI) and/or Field Ionization (FI))
and fragmented to obtain their mass spectrum. The chromatographic conditions used for the analysis were as follows:
injection port, 250 ºC; interface, 250 ºC; oven initial temperature of 40 ºC for 4 min; ramp, 25 ºC min$^{-1}$ to 120 ºC,
held for 10 min; second ramp, 20 ºC min$^{-1}$ to 200 ºC, held for 2 min.
The oxidants were generated by photolysis (λ=350 nm) of molecular chlorine (Cl$_2$) for reactions with Cl atoms and
by methyl nitrite (CH$_3$ONO) in the presence of O$_2$ and NO, for reactions with OH radicals. Some experiments have
been carried out using H$_2$O$_2$ as precursor of OH and UV radiation (λ=254 nm) in a Quartz gas cell reactor.
The decomposition of dinitrogen pentoxide was used as source of NO$_3$ radical according to reaction 1.
$$N_2O_5 + M \rightarrow NO_2 + NO_3\cdot + M \hspace{3cm} (R1)$$
The kinetic experiments were conducted in a nitrogen atmosphere for reactions with Cl atoms, while synthetic air
was used for reactions with OH radicals. All the experiments conducted for the study of products were carried out
in synthetic air.
The reactions were followed by measuring the absorbance of the characteristic IR bands of each organic compound
(33DMbutanal/33DMbutanone and the reference compounds in the case of kinetic analysis) at different reaction
times. The IR spectra were processed using OMNIC software, thought a subtraction procedure of the IR bands.
The concentration ranges (in ppm) used in the kinetic experiments were: 10-12 for 33DMbutanal and
33DMbutanone, 9-10 for 1-butene, 10-14 for Propene, 35-40 for 2-methylpropene, 13-14 for Isopropanol, 10-17 for
Cyclohexane, 5-14 for Propanal, 9-9.5 for 2-methyl-2-butanol, 5-6 for Ethyl formate, 11-15 for 1-butanol, 17-22 for
Cl$_2$, 15-20 for NO and 16-20 for methyl nitrite. In the case of the reaction product experiments, the typical
concentration (in ppm) were: 10-14 for 33DMbutanal and 33DMbutanone; 22 for Cl$_2$, 15-20 for NO, 16-20 for
methyl nitrite, 30 for H$_2$O$_2$ and for 14-25 for N$_2$O$_5$.
**2.3 Materials**



Information on the purity and supplier company of the reagents used to carry out the experiments is specified below:
33DMbutanal (95%), 33DMbutanone (97%), and the reference compounds: 1-butene and Propene ($\geq$ 99%); 2-
methylpropene ($\geq$ 99.5%); Isopropanol (70% in $H_2O$), Propanal (97%), Cyclohexane (99.5%), 2-methyl-2-butanol
($\geq$99%), Ethyl formate (97 %) and 1-butanol ($\geq$99 %), 2,2-dimethylpropanoic acid (99%), 3,3-dimethylbutanoic acid
(98%) from Sigma Aldrich, Acetone ($\geq$99.5%) from Supelco. 22DMpropanal (>95%) from TCI. The precursors of
the radicals were: methyl nitrite, $CH_3ONO$, synthesized in the laboratory according to the method of Taylor et al.,
(1980); and $Cl_2$ (99%) from Praxair. NO (98.5%) from Air Liquide, $H_2O_2$ (50 wt. % in $H_2O$, stabilized) from Sigma
Aldrich; $N_2$ (99.999%) and synthetic air (99.999%) from Praxair. Dinitrogen pentoxide ($N_2O_5$) synthesized in the
laboratory according to the procedure described by Schott and Davids (1958).
**3 Results and discussion**
**3.1 Kinetic study**
The reference compounds have been selected according to the following conditions: first that at least one active IR
band does not overlap with those of the compound under study (33DMbutanal or 33DMbutanone) and second, that
$0.1 \leq k_{carbonyl}/k_R) \leq 10$. In addition, a series of experiments was carried out in order to evaluate possible heterogeneous
reactions with the walls, reactions between the compound under study and the reference compound, photolysis of
any of them and/or reactions with the oxidant precursor. The results of these experiments showed that the losses or
the reactants due to these processes were negligible (< 3% dark loss in the case of 33DMbutanal and 0% for
33DMbutanone).
The IR absorption bands used to follow the evolution of the different compounds were: 33DMbutanal, 2700 $cm^{-1}$;
1-butene, 911 $cm^{-1}$; propene, 878-942 $cm^{-1}$; 2-methylpropene 912 $cm^{-1}$; Isopropanol 1070 and 1251 $cm^{-1}$;
cyclohexane, 2862 and 2933 $cm^{-1}$, 33DMbutanona 1137 $cm^{-1}$, propanal 2710 $cm^{-1}$, 2-methyl-2-butanol 883 $cm^{-1}$,
ethyl formate 1192 and 1194 $cm^{-1}$, 1-butanol 1060 $cm^{-1}$ .
The plots of $\ln([carbonyl]_0/[carbonyl]_t)$ versus $\ln([R]_0/[R]_t)$ for each reaction were generated according to equation
(I). As an example, in Figure 1 is shown the plot of equation (I) for the reaction of 33DMbutanal with Cl atoms with
the two reference compounds used. At least three reference compounds were used for each reaction, to assure the
accuracy of value determined. The reference compounds used, and the values of their rate coefficients are included
in Table 1. These values correspond to those recommended by McGillen et al., (2020).





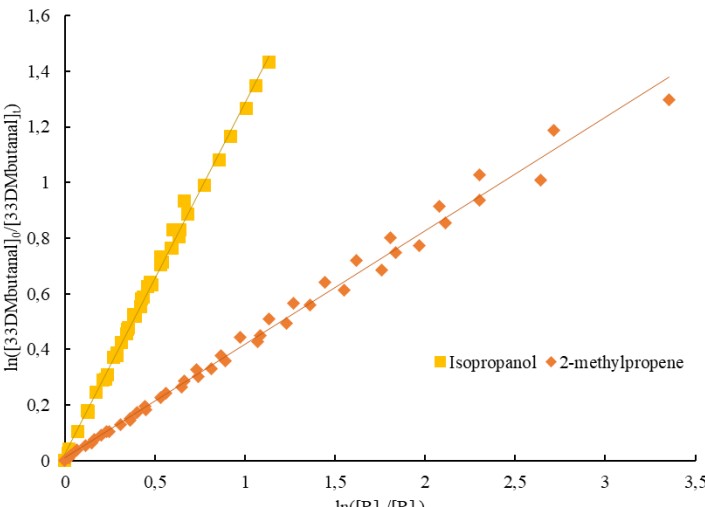


Figure 1. Plot of Eq (I) for the reaction of 33DMbutanal with Cl atoms and for two reference compounds.
The slopes of the plots correspond to the relationship $k_{carbonyl}/k_R$, knowing the value of $k_R$, the rate coefficients of
carbonyls can be determined. It can be seen the good linear fit with an intercept close to zero, indicating the absence
of secondary reactions. The plots for the reaction of 33DMbutanone with Cl atoms an OH radical are shown in
Figures 1S. The results obtained are shown in Table 1.
Table 1. Summary of relative and absolute rate coefficients for the reaction of 33DMbutanal with Cl atoms and
33DMbutanone with Cl atoms and OH radicals. k in units of $cm^3$ molecule$^{-1}$ s$^{-1}$.

| Reaction | Reference compound[d] | $k_{carbonyl}/k_R \pm 2\sigma$ | $k_{carbonyl} \pm 2\sigma$[a,b,c] |
|---|---|---|---|
| | 2-methylpropene | 0.38±0.01 | 1.26±0.27 |
| | $k_{Cl} = 3.3\pm0.7$ | 0.40±0.01 | 1.32±0.28 |
| | | 0.44±0.01 | 1.46±0.31 |
| | Isopropanol | 1.38±0.02 | 1.20±0.31 |
| 33DMbutanal+Cl[a] | $k_{Cl} = 0.87\pm0.23$ | 1.25±0.02 | 1.09±0.28 |
| | | 1.27±0.02 | 1.11±0.29 |
| | Cyclohexane | 0.43±0.01 | 1.42±0.22 |
| | $k_{Cl} = 3.3\pm0.5$ | 0.36±0.01 | 1.19±1.18 |
| | | 0.42±0.02 | 1.40±0.22 |
| | | | |
| **Weighted Average** | | | **1.27±0.08** |
| | Propanal | 0.41±0.02 | 5.10±1.06 |
| | $k_{Cl} = 12.5\pm2.5$ | 0.37±0.01 | 4.62±0.93 |
| | | 0.42±0.01 | 5.22±1.06 |
| | | 0.42±0.01 | 5.31±1.07 |
| 33DMbutanone+Cl[b] | 2-methyl-2-butanol | 0.42±0.01 | 3.03±0.76 |
| | $k_{Cl} = 7.30\pm1.80$ | 0.43±0.01 | 3.10±0.78 |
| | | 0.47±0.01 | 3.40±0.85 |
| | Isopropanol | 0.51±0.02 | 4.41±1.16 |





|  | $k_{Cl}$ =8.7±2.3 | 0.48±0.02 | 4.21±1.10 |
|  |  | 0.49±0.01 | 4.22±1.10 |
|  | Ethyl formate | 4.87±0.36 | 4.97±1.05 |
|  | $k_{Cl}$ = 1.0±0.2 | 4.93±0.18 | 5.03±1.02 |
|  |  | 4.51±0.32 | 4.60±0.98 |
| **Weighted Average** |  |  | **4.22±0.27** |
| 33DMbutanone+OH[c] | Isopropanol | 0.27±0.01 | 1.39±0.15 |
|  | $k_{OH}$ =5.2±0.5 | 0.22±0.01 | 1.17±0.12 |
|  |  | 0.23±0.02 | 1.18±0.12 |
|  | 2-methyl-2-butanol | 0.49±0.03 | 1.67±0.51 |
|  | $k_{OH}$ = 3.4±1.0 | 0.40±0.02 | 1.38±0.42 |
|  |  | 0.56±0.03 | 1.92±0.59 |
|  | 1-butanol | 0.14±0.01 | 1.36±0.28 |
|  | $k_{OH}$ = 9.8±2.0 | 0.16±0.01 | 1.62±0.34 |
|  |  | 0.18±0.01 | 0.96±0.11 |
|  | Cyclohexane | 0.18±0.01 | 1.23±0.13 |
|  | $k_{OH}$ = 6.7±0.7 | 0.15±0.01 | 1.00±0.11 |
|  |  | 0.25±0.03 | 1.67±0.26 |
|  |  | 0.23±0.02 | 1.52±0.20 |
| **Weighted Average** |  |  | **1.25±0.05** |

[a, b, c] $k_{carbonyl}$ and $k_R$ is given in $10^{-10}$, $10^{-11}$, $10^{-12}$, respectively. The data of $k_R$ are values recommended by McGillen et al., (2020). The total absolute error $\sigma(k_{carbonyl})$ is a combination of the statistical errors from the regression analysis ($\sigma_{slope}$) and the quoted error in the value of the rate coefficient of the reference compound ($\sigma_R$). The final values of the rate coefficients and the associated error were calculated as weighted average.

To the best of our knowledge, this is the first work where the rate coefficient for the reaction of 33DMbutanal with Cl atoms is determined. On the other hand, the values of the rate coefficients of the reaction of 33DMbutanone with Cl atoms and OH radicals have been previously determined with values of $(0.48 \pm 0.05) \times 10^{-10}$ cm³ molecule⁻¹ s⁻¹ (Farrugia et al., 2015) for Cl reaction and $(1.21 \pm 0.05) \times 10^{-12}$ cm³ molecule⁻¹ s⁻¹ (Wallington and Kurylo, 1987) and $(1.2 \pm 0.2) \times 10^{-12}$ cm³ molecule⁻¹ s⁻¹ (Mapelli et al., 2023) for OH reaction. These data are in good agreement with the values obtained in this study, thereby contributing to the accurate determination of the rate coefficients.

In the case of 33DMbutanone reactions the rate coefficient of Cl reactions ($4.22 \times 10^{-11}$ cm³ molecule⁻¹ s⁻¹) is one order of magnitude higher than the corresponding to OH reactions ($1.25 \times 10^{-12}$ cm³ molecule⁻¹ s⁻¹). This is the general trend observed in the atmospheric chemistry for the oxidation reactions of organic compounds; $k_{Cl}$>$k_{OH}$>>$k_{NO3}$. This behaviour can be explained by the higher reactivity and lower selectivity of chlorine atoms compared to the OH radical, where the site of attack determines its reactivity (Colmenar et al., 2020).

In Table 2, the rate coefficients for the reaction of different aldehydes and ketones in the butyl series with the main atmospheric oxidants have been tabulated to analyse the influence of the ramifications on reactivity.

Table 2. Rate coefficients for aldehydes and ketones in the butyl series with key atmospheric oxidants. k in units of cm³ molecule⁻¹ s⁻¹.

| **Aldehydes** |
| --- |



| Compounds | $k_{Cl}{}^a$ | $k_{OH}{}^b$ | $k_{NO3}{}^c$ |
|---|---|---|---|
| Butanal | $1.66\pm0.4^d$ | $23.7\pm5^d$ | $1.10\pm0.4^d$ |
| 2-methylbutanal | $2.16 \pm 0.32^e$ | $33.3\pm13^d$ | $2.67\pm0.8^d$ |
| 3-methylbutanal | $2.07 \pm 0.14^f$ | $25.9\pm5^d$ | $2.19\pm0.7^d$ |
| 3,3-dimethylbutanal | **$1.27\pm0.09^g$** | $21.4\pm9^d$ | $1.77\pm0.4^d$ |
| **Ketones** | | | |
| Compounds | $k_{Cl}{}^a$ | $k_{OH}{}^b$ | $k_{NO3}{}^c$ |
| 2-butanone | $0.40\pm0.16^d$ | $1.05\pm0.2^d$ | |
| 3-methylbutanone | $0.68\pm0.07^d$ | $3.00\pm1.2^d$ | $<0.05^h$ |
| 3,3-dimethylbutanone | $0.48\pm0.05^d$ | $1.21\pm0.5^d$ | |
| | **$0.42\pm0.03^g$** | **$1.25\pm0.1^g$** | |
| | | $1.2\pm0.2^i$ | |

[a, b, c] k in $10^{-10}$, $10^{-12}$, $10^{-14}$, respectively. [d] Values recommended in McGillen 2020. [e] Asensio et al., 2022. [f] Bo et al.,
2022. [g] This work. [h] Glasius et al., 1997. [i] Mapelli et al., 2023
For butanals, the trend in rate coefficient values indicates that the presence of a methyl group influences to the
reactivity, resulting in an increase in the rate coefficient compared to the compound without a methyl group. This
could be attributed to the activation of the hydrogen atom at the α- position by a methyl group, as noted in the
literature (Mellouki et al., 2015). Furthermore, the reactivity is influenced by both the position and quantity of methyl
groups. Consequently, the activating influence exerted by the methyl group on the hydrogen bonded to the carbon
adjacent to the aldehyde (α- position) is less pronounced when the methyl group occupies position 3 as opposed to
position 2. Basically, the impact of the methyl group manifests as a short-range activating effect. Regarding the
impact of the number of methyl groups on reactivity, in the case of 33DMbutanal, the significant decrease in the
value of rate coefficient with respect to 3-methylbutanal could be explained by an increase in steric hindrance,
making the hydrogen abstraction process at the α- position less probable, thereby resulting in lower reactivity
compared to the compound with one methyl group (3-methylbutanal).
Concerning butanones, no data are available for reactions with $NO_3$ radicals, with only one value for the upper limit
of 3-methylbutan-2-one (Glasius et al., 1997). The fact that the reactions of ketones with $NO_3$ radicals are too slow
complicates their experimental study and therefore, the determination of their rate coefficients. The available data
of rate coefficients for reactions of butanones with Cl atoms and OH radicals, show again that the presence of a
methyl group attached to a carbon in the α- position with respect to the carbonyl group actives the abstraction of
one hydrogen atom from this carbon, resulting in an increase of the rate coefficient. A comparison of rate coefficients
for 2-butanone and 3-methylbutan-2-one reveals this effect, particularly pronounced in OH reactions. The presence
of two methyl groups (33DMbutanone) produces a significant decrease in the value of the rate coefficient, that again
could be explained by steric hindrance.
As can be observed in Table 2, the type of carbonyl group (aldehyde or ketone) also exerts a significant influence
on the reactivity. The rate coefficients are generally one or two orders of magnitude higher for the aldehyde reactions
compared to the reaction of ketones. The different reactivity of aldehydes and ketones with main atmospheric
oxidants has extensively been studied and documented in the literature (McGillen et al., 2020, Mellouki et al., 2015,).
The different reactivity observed in the reactions of atmospheric radical with saturated carbonyl compounds that are
initiated by hydrogen abstraction, are due to the presence in the carbonyl compound of different types of hydrogens.
In the aldehydic compounds there are two types of hydrogens that can be abstracted, the hydrogen directly attached
to the carbonyl group (aldehydic hydrogen) and the hydrogen attached to the alkyl group (alkyl hydrogen), while in



a ketone only alkyl hydrogens are present. The available kinetic and mechanistic data on the atmospheric degradation
indicate that the H atom abstraction from the aldehydic group (−CHO) is more favoured that H atom abstraction
from the C−H bonds of the alkyl chain. The rate coefficients obtained in this study for 33DMbutanal and
33DMbutanona confirm this argument.
On the other hand, it is well known that functional groups exert an activating or deactivating effect on reactivity,
depending on the type of groups. The reactivity factors (F(R); R=functional group) associated with the functional
group to which a type of carbon is attached (primary ($k_{prim}$), secondary ($k_{sec}$) or tertiary ($k_{tert}$)) can be quantified using
the experimental kinetic database available in the literature. Consequently, rate coefficients for the reactions of
33DMbutanal and 33DMbutanone with Cl atoms and OH and $NO_3$ radicals have been estimated with the SAR
(Structure-Activity Relationship) predictive method (Calvert et al., 2011, Kerdouci et al., 2014, Kwok and Atkinson,
1995). In the case of the two carbonyls compounds of this work, the only possibility of reacting is the abstraction of
one hydrogen atom due to the absence of doble bonds. The global abstraction rate coefficients can be calculated as
$k_{abs} = 3(k_{prim}F(C)) + k_{sec}F(C)F(-CHO) + k_{-COH}F(CH_2)$        for        33DMbutanal        and        $k_{abs} =$
$3(k_{prim}F(-CR_2CO-)) + (k_{prim}F(-CO-)$ for 33DMbutanone.
The rate coefficients (in $cm^3$ molecule$^{-1}$ s$^{-1}$) and factors used to obtain the estimated rate coefficients for
33DMbutanal and 33DMbutanone with Cl atoms are: $k_{prim}$=2.84x10$^{-11}$, $k_{sec}$=8.95x10$^{-11}$, $k_{-COH}$=5.13x10$^{-11}$, F(C)=0.79
proposed by Calvert et al., 2011, F(-CHO)=0.4 proposed by Carter et al 2021, F(-CR₂CO-)=0.563 and F(-CO)=0.037
proposed by Farrugia et al., 2015. Therefore, the estimated rate coefficient for Cl reaction have been 1.36x10$^{-10}$ and
0.49x10$^{-10}$ $cm^3$ molecule$^{-1}$ s$^{-1}$ for 33DMbutanal and 33DM-2-butanone respectively. In the case of OH reactions the
rate coefficients have been estimated using the EPI (Estimation Programs Interface) Suite™, (US EPA), specifically
the AOPWIN™. The rate coefficients estimated have been $k_{estimated}$=22.12x10$^{-12}$ for 33DMbutanal and
$k_{estimated}$=1.69x10$^{-12}$ for 33DM-2-butanone. For the reaction of $NO_3$ radical only the estimated rate coefficient for
33DMbutanal have been done obtaining a $k_{estimated}$=2x10$^{-14}$ $cm^3$ molecule$^{-1}$ s$^{-1}$ using the data of Kerdouci et al.,
(2014). In all cases the estimated rate coefficients are very similar to the experimental values, indicating that the
reactivity factor used for the estimations are well established. Reaction product studies and theoretical calculations
of these reactions will help confirm the arguments presented above.
**3.2 Products study and Mechanisms of reaction**
The products of the reactions of 33DMbutanal and 33DMbutanone with Cl atoms have been studied in the presence
and in the absence of NO to evaluate different atmospheric conditions. In addition, the products of the reaction of
33DMbutanal with OH and $NO_3$ radical and 33DMbutanone with OH radicals have also been studied. All these
reactions have been carried out using the experimental systems described above.
Based on the principles of tropospheric reactivity (Atkinson, 2007, Finlayson-Pitts and Pitts 2000) and in the case
of 33DMbutanal in a previous products study with OH (Aschmann et al. 2010), a complete reaction mechanism has
been proposed for both carbonyls to facilitate the identification of the products. The Scheme 1S and 2S
(supplementary information) show these reaction mechanisms.



In the proposed mechanism for the reactions of 33DMbutanone (Scheme 1S), the initial attack by the oxidant can
occur at the -$CH_3$ in the α-position with respect to the carbonyl (channel I) or at any of the -$CH_3$ of the tert-butyl
group (channel II).

According to the SAR predictions the H-abstraction in the -$CH_3$ of the tert-butyl group is the main channel, for the
reaction with both oxidants (~ 98 % for Cl atoms and ~ 94% for OH radicals, see Table 1S).
In the proposed mechanism of the reactions of 33DMbutanal (Scheme 2S), the initial attack can take place in three
different groups: -CHO (channel I) -$CH_2$- (channel II) and -$CH_3$ (channel III) from tert-butyl group.

Aschmman et al., (2010) proposed the reaction in the -CHO group as the mayor initial channel of this compound
with OH radicals. However, according to the SAR predictions used in this work, the main initial attack depends on
the type of oxidants. Therefore, for the reaction with Cl atoms the main initial attack would take place in the -$CH_3$
(~ 49%) followed by -CHO (~ 30%) and lastly the -$CH_2$- group with ~21%. For the reaction of 33DMbutanal with
$NO_3$ radical the main initial attack would take place in the CHO (~ 63%) followed by -$CH_2$-(~37%). And for OH
reactions the main initial attack of the radical would take place in the -CHO (~ 94%) followed by -$CH_2$- (~ 4%) and
lastly the -$CH_3$ (~2%). Note that the percentage of -$CH_3$ corresponds to three times the % of channel III (see Table
1S).
It is well established (Atkinson, 2007) that alkyl radicals, formed in the initial step of these reactions, rapidly react
with $O_2$ to generate the corresponding peroxyradical ($RO_2 \cdot$). These $RO_2$ radicals can undergo various pathways (see
Schemes 1S and 2S). In the absence of NO, peroxyradicals primarily undergo two self-reaction processes: one
leading to the formation of alkoxyradicals ($RO_2 \cdot + RO_2 \cdot \rightarrow 2RO \cdot + O_2$), and the other producing neutral compounds,
such as hydroxy compounds and carbonyl compounds ($RO_2 \cdot + RO_2 \cdot \rightarrow$ hydroxy compound + carbonyl compound
+ $O_2$). Another significant process is the reaction of $RO_2$ with OH radicals, the likelihood of which depends on the
size and structure of the alkyl group (R) (Berndt et al., 2018; Bottorff et al., 2023; Fittschen, 2019).
In the presence of NO, the $RO_2$ radical may react to form alkoxyradicals and $NO_2$ ($RO_2 \cdot + NO \cdot \rightarrow RO \cdot + NO_2$) or
nitrated compounds ($RONO_2$), and in presence of large concentration of $NO_2$, $RO_2$ generates peroxynitrated



compounds (ROONO$_2$) (pathway less favoured). Under typical tropospheric conditions, alkoxyradicals can react
with oxygen, undergo unimolecular decomposition, or isomerize (Atkinson, 2007). The reaction of RO· radicals
with O$_2$ is only possible if the carbon atom bearing the radical contains at least one hydrogen atom (Atkinson, 2007).
Additionally, in the presence of NO and NO$_2$, alkoxyradicals can also form nitrated compounds.
In an effort to stablish the main reaction paths for the reactions studied in this work, the rate coefficient for
unimolecular decomposition and isomerization have been estimated following the method outlined by Vereecken
and Peeters (2009, 2010). In the case of 33DMbutanone, the rate coefficients estimated for the unimolecular
isomerization of the initial alkoxyradical formed in both pathways were found to be at least one order of magnitude
lower than the estimated rate coefficients for the decomposition of the same alkoxyradical. Consequently, the
reaction products generated from isomerization channel will not be important. In channel II, the estimated rate
coefficient of the decomposition process to obtain acetone has been $1.7 \times 10^{12}$ s$^{-1}$, that is much higher than the
estimated rate coefficient to obtain butane-2,3-dione ($6 \times 10^3$ s$^{-1}$). In the case of 33DMbutanal there is one possibility
of isomerization in channel III where the H-atom implied come from the aldehydic group. However, based on the
estimated rate coefficients, the rate coefficient for decomposition is four times higher than that for the isomerization
process.
Based on the general mechanism proposed in Scheme 1S for 33DMbutanone and Scheme 2S for 33DMbutanal
respectively, the formation of the expected products have been investigated using two analytical techniques. Also,
the influence of the presence or absence of NO in the reaction of these two carbonyls with Cl atoms has been
evaluated.
**3.2.1 FTIR experiments**
The procedure followed to analyse the FTIR spectra have been described in previous works (Aranda et al., 2020,
Aranda et al., 2024, Colmenar et al., 2108, 2020a, 2020b). Therefore, only the main results will be indicated.
The residual IR spectra of the reaction products, obtained after subtracting the spectra of all known compounds
(33DMbutanone, 33DMbutanal, HCl, NO, NO$_2$, CH$_3$NO$_2$, N$_2$O$_5$, HNO$_3$, HNO$_2$, etc.), were compared with IR spectra
of commercial samples or database spectra (Eurochamp 2020 database https://data. eurochamp.org/data-
access/spectra/) last access: 9 July 2024). The identified and quantified reaction products were acetone
(CH$_3$C(O)CH$_3$) and formaldehyde (HCHO) for all reactions (except to 33DMbutanal with NO$_3$); 2,2-
dimethylpropanal ((22DMpropanal, (CH$_3$)$_3$CCHO)) for the reactions of 33DMbutanal with Cl atoms; and nitrated
compounds in those reactions carried out in presence of NO and/or NO$_2$. The nitrated compounds were attributed to
alkoxy nitrates (RONO$_2$ ~1663, 1284, 853 cm$^{-1}$) and peroxy nitrates (ROONO$_2$ ~1718, 1300 and 793 cm$^{-1}$)
(Finlayson-Pitts and Pitts, 2000). A peroxycarbonyl nitrates as PeroxyAcetyl Nitrate (PAN, CH$_3$C(O)OONO$_2$
~1830, 1300 and 793 cm$^{-1}$) was identified and quantified in reactions of 33DMbutanone + Cl conducted in the
presence of NO after 3-5 minutes of reaction time. Figure 2 shows an example of residual spectra from the reactions
of 33DMbutanone and 33DMbutanal with Cl atoms in the absence and presence of NO. The figure includes reference
spectra to corroborate the formation of these compounds.





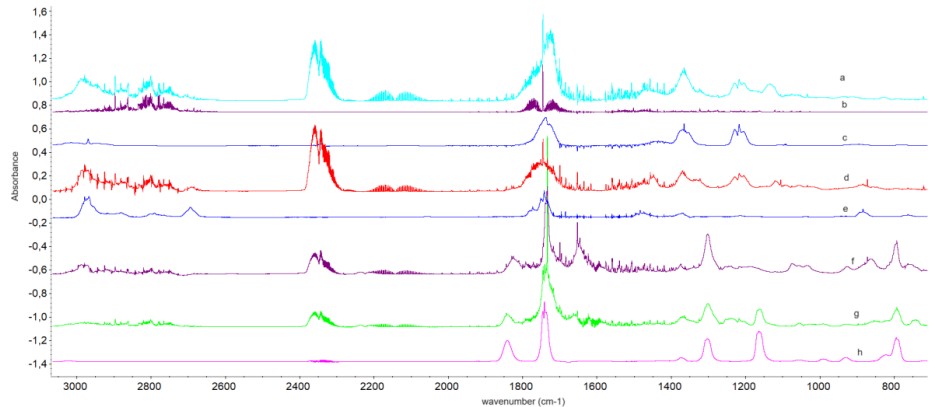

Figure 2. Residual FTIR spectra for the reactions of: 33DMbutanone with Cl in the absence of NO at 21 min (a) and in the presence of NO at 21 min (g), 33DMbutanal with Cl atoms in the absence of NO at 18 min (d) and in the presence of NO at 18 min (f) 64 % of the conversion and a 29% respectively. Reference IR spectrum of HCHO (b) acetone (c) (commercial sample), 22DMpropanal (e) and PAN (h). The spectra have been shifted for clarity.

For the reactions of 33DMbutanal with $NO_3$, nitrated bands attributed to alkoxy nitrates, peroxy nitrates and peroxycarbonyl nitrates are clearly observed. The peroxycarbonyl nitrates could correspond to peroxy-3,3-dimethylbutyryl nitrate ($(CH_3)_3CCH_2C(O)OONO_2$) that is formed due to the large amount of $NO_2$ presents in the reaction mixture from the initial time. Figure 3 shows the characteristic IR absorption bands of nitrated compounds formed in the reaction of 33DMbutanal with $NO_3$.

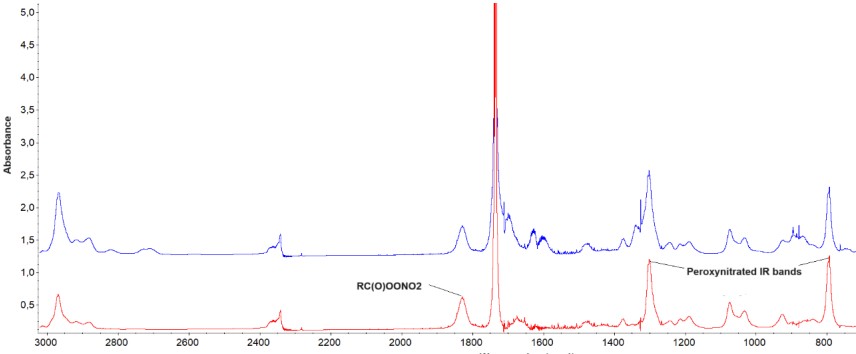

Figure 3. FTIR spectrum of the reaction of 33DMbutanal (~25% of conversion) with $NO_3$ radical (upper). FTIR residual spectra (assigned to peroxy-3,3-dimethylbutyryl nitrate) after elimination of $N_2O_5$, $HNO_3$, 33DMbutanal and $NO_2$ (lower).

It is important to note that for the reactions of 33DMbutanal with Cl atoms (in the presence of NO) and with OH radical at large reaction times, a nitrated compound, as observed in the $NO_3$ reaction, has been detected. Additionally, IR bands of $N_2O_5$ (precursor of $NO_3$) has been observed in the reaction with Cl atoms as consequence of the reaction of $O_3 + NO_2$. This last result indicates the formation of ozone in the degradation process of 33DMbutanone/33DMbutanal in the presence of radiation and $NO_2$.



Figure 2S shows IR spectra of the nitrated compounds together with $N_2O_5$ reference spectrum. The IR bands of the
nitrated compounds show great absorbance in the case of Cl reaction compared to the OH reactions. This fact could
be due to an additional contribution from the 33DMbutanal reaction with $NO_3$. On the other hand, 22DMpropanal
has not been observed in the reactions of 33DMbutanal with OH and $NO_3$ radicals, probably due to the overlapping
of their characteristic IR bands with the ones of nitrated compounds.
To evaluate the amount of nitrated compounds formed in the reactions studied with Cl atoms in the presence of NO,
and in the 33DMbutanal reaction with $NO_3$ an estimation has been made using the average integrated absorption
coefficient of $1.2 \times 10^{-17}$ cm molecule$^{-1}$ corresponding to the IR range 1250-1330 cm$^{-1}$ for similar compounds
(Tuazon and Atkinson, 1990). In the reaction of 33DMbutanal and 33DMbutanone with OH radicals, the yield of
nitrated compounds was not estimated, because there is an additional contribution due to the precursor used
(methylnitrite). For PAN quantification in the reaction of 33DMbutanone with Cl in the presence of NO, the
reference spectrum (Eurochamp 2020 database, https://data. eurochamp.org/data-access/spectra/) last access:
September 2024) has been used.
The time-concentration profiles of the quantified products formed, and the consumption of the carbonyl reactant
have been represented in Figure 4.



(a)

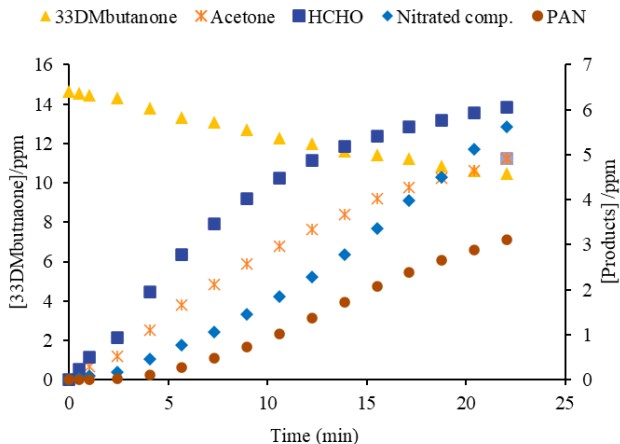


(b)

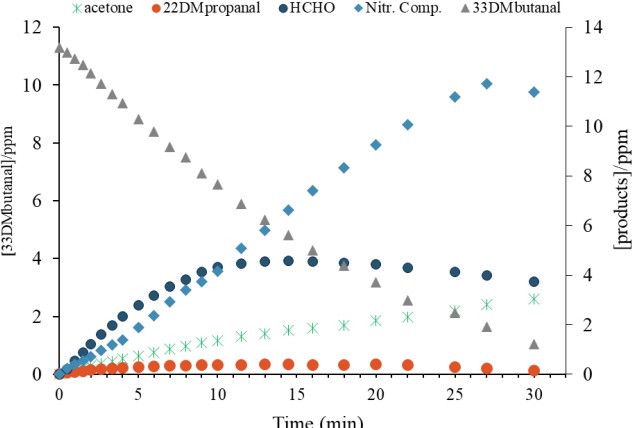


Figure 4. Time-concentration profiles of the products formed, and the carbonyl reacted for the reaction of (a)
33DMbutanone and (b) 33DMbutanal with Cl atoms in the presence of NO. For 33DMbutanone + Cl + NO, the
nitrated compounds profiles are the total nitrated compounds (alcoxy, peroxy and PAN).
For the reaction of 33DMbutanone with Cl atoms, the trends of acetone and HCHO indicate that they are primary
products, although the concentration of HCHO starts to decrease at 20 min of reaction, possibly due to secondary
chemical reactions. The profile of nitrated compounds, especially PAN, shows a significant increase after 5 minutes
of reaction. In the case of the reaction of 33DMbutanal with Cl atoms, all trends (for acetone, HCHO, 22DMpropanal
and nitrated compounds) suggest that they are primary products in the early stages. However, 22DMpropanal and
HCHO seem to undergo further reactions due to secondary chemistry, while the concentrations of acetone and
nitrated compounds increase more than expected, likely due to contributions from other sources (Figures 3S-5S).
For the nitrated compounds, a change in the trend is observed around 2 minutes, likely due to the formation of
nitrated peroxycarbonyl compounds as a result of the presence of $NO_2$ in the reaction mixture. The profile of nitrated



compounds from the reaction of 33DMbutanal with $NO_3$ shows an increase from the initial of the reaction (see
Figure 4S), due to the presence of $NO_2$ in the reaction medium from the beginning.
The yields for primary products have been calculated from the slopes of the plots of the concentration of the products
formed against the variation in the carbonyl reactant consumed using the first data of the reactions to avoid secondary
chemistry contributions. In some cases, the yields have been difficult to obtain due to the overlapping of IR bands
with other unidentified products. In these cases, in which the concentration of the compounds that seems to react
with the main oxidant (such as 22DMpropanal and HCHO) the yields have been recalculated using the formalism
published by Tuazon et al., 1986. Figure 5 showed an example of yield plots for the reactions of 33DMbutanal with
Cl atoms in the absence and the presence of NO. The yield of PAN formed in the reaction of 33DMbutanone with
Cl atoms in the presence of NO has been estimated from the slopes of the plots, where the data show a linear
behavior, corresponding to a $\Delta$[3,3DMbutanone] of approximately 1.5 ppm (see Figure 5a).





(a)

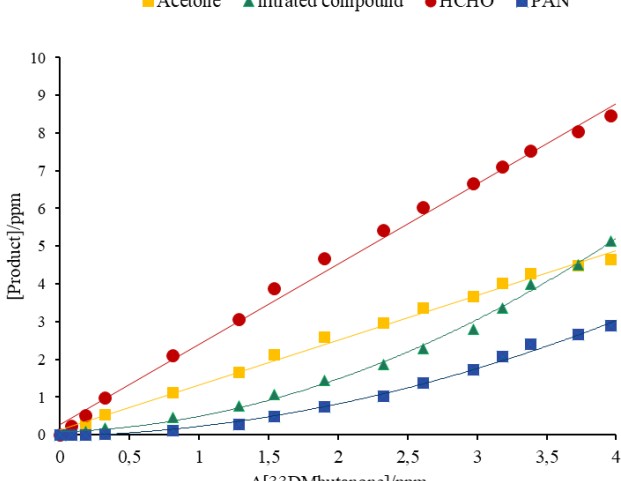


(b)

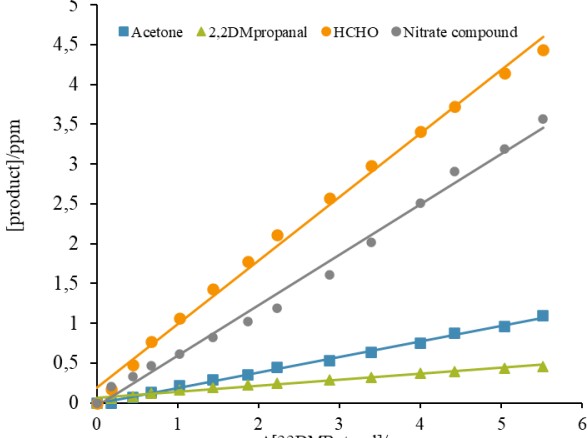


Figure 5. Plots of the reaction product formed versus the consumption of the reactant in the reaction of (a)
33DMbutanone + Cl + NO and (b) 33DMbutanal + Cl + NO.
The Figures 6S and 7S of supplementary material show the yield plots for the reactions of 33DMbutanone and
33DMbutanal with Cl atoms and 33DMbutanone with OH in the absence of NO respectively.
A summary of the estimated yields of reaction products, identified and quantified through FTIR analysis, is presented
in Table 3 for the reaction of 33DMbutanone + Cl, Cl + NO and + OH radical. Table 4 shows the results for the



reactions of 33DMbutanal + Cl; Cl + NO, OH and $NO_3$ radical. Table 3. Estimated yields (%) of reaction products
identified with FTIR analysis and the reaction products tentatively assigned from SPME/GC-TOFMS analysis in
the reactions of 33DMbutanone with Cl atoms and OH radical in the absence and in the presence of NO.
Table 3. Estimated yields (%) of reaction products identified with FTIR analysis and the reaction products tentatively
assigned from SPME/GC-TOFMS analysis in the reactions of 33DMbutanone with Cl atoms and OH radical in the
absence and in the presence of NO.

| Reaction | FTIR | | | | GC-TOFMS | |
|---|---|---|---|---|---|---|
| | Yield acetone (%)±2σ | Yield HCHO[a,b] (%)±2σ | Yield nitrated comp. (%)±2σ | Yield PAN (%)±2σ | $t_r$ min | Reaction Product |
| 33DMbutanone + Cl | 65.0±1.1 | 153.3±7.0 | | | 2.2 | Acetone |
| | 65.9±1.8 | 178.1±4.1 | | | 2.87 | Hydroxyacetone |
| | 69.0±1.3 | 174.7±3.6 | | | 7.03 | Hydroxy-2,3-butanodione |
| **Average** | **66.6±4.2** | **168.7±30.0** | | | 8.28 | 2,2DM3-oxo-butanal |
| **Total carbon[c, d] (%)** | **61** | | | | 10.26 | 4-hydroxy-3,3-DM2butanone |
| 33DMbutanone + Cl + NO | 123.2±3.2 | 212.4±7.4 | 59.2±1.9 | 101.0±7.0 | 2* | Acetone |
| | 111.2±6.3 | 195.4±5.8 | 56.6±6.6 | 112.9±18.4 | 5.59 | Peroxyacetylnitrate (PAN) |
| | 137.4±11.0 | 199.9±11,5 | 66.9±6.1 | 95.5±44.2 | 6.81*(7.03)[(1)] | Hydroxy-2,3-butanodione |
| **Average** | **124.0±26.2** | **202.6±17.6** | **60.9±10.7** | **103.1±8.8** | 8.01*(8.28) | 2,2DM3-oxo-butanal |
| | | | | | 9.04 | 2,2-dimethyl-3-propyl-oxynitrite |
| **Total carbon[c, d] (%)** | **95.6** | | | | 9.68 | Nitrated compound |
| | | | | | 14.57 | Nitrated compound |
| 33DMbutanone + OH[e] | 31.4±2.0 | 59.5±2.4 | | | | - |
| | 31.5±1.0 | 68.2±3.8 | | | | |
| | 34.1±1.6 | 73.5±4.2 | | | | |
| **Average** | **32.3±3.0** | **67±14** | | | | |
| **Total carbon[c, d] (%)** | **28** | | | | | |
| 33DMbutanone + OH+NO[f] | 121.1±6.7 | - | - | | 2.01 | Acetone |
| | 161.9±19.7 | - | - | | 2.55 | Nitrated compound |
| | 89.2±6.4 | - | - | | 9.04 | 2,2-dimethyl-3-propyl-oxynitrite |
| **Average[g]** | **93.0±72.8** | - | - | | 14.62 | Nitrated compound |
| **Total carbon[c, d] (%)** | **47** | | | | | |

[a]Yields have been estimated using the reference IR spectra from the Eurochamp database (Rodenas et al., 2017). [b]The rate
coefficient used to correct the concentration of formaldehyde has been $k_{Cl}$=7.2 × $10^{-11}$ $cm^3$ molecule$^{-1}$ s$^{-1}$ from IUPAC(2017).[c]
$Total\ carbon = \sum_{1}^{i} \frac{n^{\underline{o}}\ of\ carbon\ of\ product_i}{n^{\underline{o}}\ of\ carbon\ of\ 33dmbutanone} \times molar\ yield_i$.[d] Nitrated compounds have not been accounted for total
carbons. [e]Only FTIR experiments using $H_2O_2$ as OH radical precursor. [f]Experiments using methyl nitrite as OH radical precursor.
[g]The acetone yield must be taken with caution due to interference with IR bands of methyl nitrite. The yield of HCHO and
nitrated compounds for the reaction of 33DMbutanal with OH radical has not been determined, as there is a significant
contribution from other sources such as the precursor used to generate the OH radical (which is a nitrated compound, methyl
nitrite) and its degradation (which generates methyl nitrate and formaldehyde). [(1)] Little Peaks *GC-TOFMS experiments. The





retention time shorter than Cl+NO and OH experiment due to the use of a different chromatographic column. The positive
identification and quantification were not possible due to the scarce of commercial standards.



Table 4. Estimated yields (%) for reaction products formed in the reactions of 33DMbutanal with Cl atoms in the
absence and in the presence of NO and with $NO_3$ and OH radical using FTIR and the product identify in the
qualitative analysis using GC-TOFMS.

| Reaction | FTIR | | | | GC-TOFMS | |
|---|---|---|---|---|---|---|
| | Yield 22DMpropanal[a] (%)±2σ | Yield acetone (%)±2σ | Yield HCHO[b,c] (%)±2σ | Yield Nitrated comp. (%)±2σ | $t_r$ min | Reaction Product |
| 33DMbutanal + Cl | 28.4±0.4 | 31.1±0.6 | 37.4±0.3 | - | 2.2 | Acetone |
| | 29.6±0.3 | 26.8±0.4 | 38.5±0.4 | - | 2.65 | Hydroxyacetone |
| | 29.9±2.0 | 27.5±0.3 | 40.5±1.0 | - | 3.42 | 22DMpropanal |
| **Average** | **29.3±0.7** | **28.5±2.4** | **38.8±1.5** | **-** | 5.82 | 22DMpropanol |
| | | | | | 8.23 | 22DMbutanodial/22DM-propanoic acid** |
| | | | | | 9.57 | 33DMbutanoic acid |
| **Total carbon[d] (%)** | **45** | | | | 13.25 | 22DMtetrahydrofuranone |
| | | | | | 15.78 | 4-hydroxi-3,3DMbutanal or (2,3-dihydro-4,4-DMfuran) |
| | | | | | 16.14 | 3-hydroxy-22-DMpropanal |
| | | | | | 20.90 | 22dimethylpropane-1,3diol |
| 33DMbutanal + Cl + NO | 12.8±2.6 | 20.3±0.6 | 83.3±1.9 | 52.1±3.1 | 2.2 | Acetone |
| | 7.7±0.2 | 20.8±0.3 | 89.2±1.4 | 50.3±3.2 | 3.41 | 22DMpropanal |
| | 7.6±0.2 | 22.4±1.0 | 95.3±3.1 | 52.5±2.0 | 5.82 | 22DMpropanol |
| **Average** | **9.4±3.0** | **21.1±1.1** | **89.3±6.0** | **51.6±1.2** | 6.97 | peroxy-3,3-dimethylbutyryl nitrate |
| | | | | | 8.26 | 33DM-oxo-butanal/22DM-propanoic acid** |
| **Total carbon[d,e] (%)** | **33.2** | | | | 9.58 | 33DMbutanoic acid |
| | | | | | 13.25 | 2,2-DMtetrahydrofuranone |
| | | | | | 16.16 | 3-hydroxy-2,2DMpropanal |
| 33DMbutanal + $NO_3$ | Not observed | Not observed | Not observed | 100 | 2 | Acetone |
| | | | | | 3.08*(3.42) | 2,2DMpropanal |
| | | | | | 5.47*(5.82) | 2,2DMpropanol |
| | | | | | 6.68*(6.97) | peroxy-3,3-dimethylbutyryl nitrate |
| | | | | | 7.15 | Nitrated compound |
| | | | | | 7.88*(8.23) | 33DM-oxo-butanal /22DMpropanoic acid** |
| | | | | | 8.94 (9.56) | 33DMbutanoic acid |
| 33DMbutanal + OH | Not observed[(1)] | 24.9±08 | - | - | 2 | Acetone |
| | | 34.6±1.0 | - | - | 7.16 | Nitrated compound |
| | | 37.1±0.7 | - | - | 8.92*(9.56) | 33DMbutanoic acid |
| **Average** | | **32.2±6.4** | **-** | **-** | | |
| **Total carbon[d] (%)** | **16.1** | | | | | |

[a]The rate coefficient used to correct the concentration of 22DMpropanal has been $k_{Cl}=1.42\times10^{-10}$ cm$^3$ molecule$^{-1}$ s$^{-1}$ from Calvert
et al., 2011. [b] Yields has been estimated using the reference IR spectra from the Eurochamp database (Rodenas et al., 2017). [c]The
rate coefficient used to correct the concentration of formaldehyde has been $k_{Cl}=7.2\times10^{-11}$ cm$^3$ molecule$^{-1}$ s$^{-1}$ from IUPAC(2017).[d]
$Total\ carbon = \sum_{1}^{i} \frac{n^{\circ}\ of\ carbon\ of\ product_i}{n^{\circ}\ of\ carbon\ of\ 33dmbutanal} \times molar\ yield_i$.[e] Nitrated compounds have not been accounted for. [(1)] Probable
interference with IR bands of nitrated compounds. The yield of HCHO and nitrated compounds has not been determined, as there



is a significant contribution from sources other than the main reaction, such as the precursor used to generate the OH radical
(which is a nitrated compound, methyl nitrite) and its degradation (which generates methyl nitrate and formaldehyde). *GC-
TOFMS experiments. Retention time shorter than Cl experiments due to the use of a different chromatographic column. The
positive identification and quantification were not possible to scarce of commercial standards. Ony 22DMpropanal was confirmed
with standard. **Secondary reaction product
As shown in Tables 3 and 4, the range of total carbon recovered is less than 100%. Only in the case of the reaction
of 33DMbutanone with Cl atoms in the presence of NO the total carbon is 95% (not accounting for nitrated
compounds), but it is important to note that the residual FTIR spectra (see Figure 8S) indicate the presence of other
compounds that are not accounted for total carbon. The residual spectra magnified (see Figure 9S) shows IR
absorption bands that can be assigned to reaction products proposed in the general Scheme 1S as formic acid and
hydroxyacetone. On the other hand, it is interesting to note that in the residual spectra for the reaction of
33DMbutanal with the three oxidants, showed in Figure 10S, some IR absorption bands appear at the same
wavenumber indicating common reaction products.
Gas chromatography coupled with a time-of-flight mass spectrometer (GC-MSTOF) using EI and/or FI ionization
mode has been employed as a complementary technique to FTIR to identify more reaction products or to confirm
those detected by FTIR. The yields should be considered with caution due to potential systematic errors during the
quantification analysis.
**3.2.2 SPME/GC-TOFMS experiments**
The SPME/GC-TOFMS chromatograms in Electron ionization mode (EI) collected at different reaction times for
the studied reaction, show peaks at different retention times whose areas increase with the reaction time, indicating
that they correspond to reaction products. (Figure 11S). Due to the characteristic of SPME sampling method, only a
qualitative analysis was possible.
For the reaction of 33DMbutanone with Cl atoms in the presence and the absence of NO, also GC-MS analysis using
Field Ionization was carried out. The FI mass spectrum helps to establish the identification of reaction products.
Figure 6 shows an example of the SPME/GC-TOFMS chromatograms collected in FI and EI mode for the reaction
of 33DMbutanone with Cl.



(a)



(b)

(c)




Figure 6. Example of the SPME/GC-TOFMS chromatograms for the reaction of 33DMbutanone + Cl atoms (a) in the absence of NO and FI mode, (b) in the presence of NO and FI mode and (c) in presence of NO and EI mode.

It can be observed that the FI chromatograms in the absence of NO (a) present more peaks than the corresponding ones in the presence of NO (b), with only one peak being common ($t_r$ = 8.29 min). This indicates that the presence of NO influences the reaction mechanism. In the case of the chromatograms obtained under the same reaction conditions but with different ionization modes ((b) and (c) chromatograms), a different number of peaks also is observed, probably because some reaction products were not ionized with FI (peaks at $t_r$=9.68 and 14.57). Additionally, the peak in chromatogram (a) and (b) of 33DMbutanone appears at shorter retention times than in chromatogram (c), due to the use of a different chromatographic column. Taking into account this, the peaks at $t_r$=2.24; 6; and 8.29 min, showed in chromatograms (b) correspond with the peaks at $t_r$=2; 5.73; 8.0 and 9.03 min in chromatograms (c).

The mass spectrum of each peak was analysed using the NIST database of GC-MS or comparing with the mass spectrum of the commercial sample. In some cases, the positive identification with high percentage of similitude index were obtained but in other cases, only a tentative assignation was possible based on profile of fragments m/z generated and the reaction products expected according to the schemes 1S and 2S proposed. In those cases, in which FI spectrum could be obtained, the assignation was made based on fragment m/z of molecular weight. All chromatograms collected in EI mode present a little peak that appear together with the peak of air, that could correspond to acetone. In order to confirm the presence of acetone to this retention time (~2 min) a chromatogram has been created using a specific tool of the software of mass spectrometer. For that, in the software it is specified the desired **m/z** (58 m/z for acetone) and then the chromatogram is generated, displaying the ion intensity versus time, with peaks representing the compounds that correspond to the specified m/z. With this tool the experimental chromatograms have been modified in for a better analysis of all chromatographic peaks. Prior to the mass spectra analysis, it was verified that the experimental and generated chromatograms were identical. In the supplementary material, the GC-TOFMS chromatograms generated with this tool for all reactions are compiled (Figures 12S-16S).

The mass spectra of all chromatographic peaks are presented in Tables 2S-3S for the reactions of 33DMbutanone and 33DMbutanal, respectively. An assignment of the reaction products corresponding to each peak has been made, taking into account schemes 1S, 2S and the results of FTIR experiments. However, due to the unavailability of commercial samples, the formation of some product proposed could not positively confirmed. Only in the reaction of 33DMbutanal, the injection of a real sample of 22DMpropanal, allowed the confirmation of this product assigned to a peak at $t_r$ =3.08/3.41 minutes.

The SPME/GC-TOFMS experiments show the formation of the main compounds quantified in the FTIR experiments, such as acetone and 2,2-dimethylpropanal (22DMpropanal) in the case of the 33DMbutanal reactions. Due to the SPME sampling procedure, formaldehyde could not be detected. The formation of other organic compounds, generally multifunctional of varying alkyl chain lengths, (hydroxycarbonyls, oxocarbonyls, hydroxy-oxo-carbonyls, organic acids) is observed. These compounds could correspond to the unidentified compounds in the FTIR spectra.

Next a discussion of the results on reaction products with both analytical techniques for each of the compounds studied is presented.



**3.2.2.1 33DMbutanone reaction products**
In the SPME/GC-TOFMS chromatograms for the reactions of 33DMbutanone (Figures 6, 12S) can see that the
number of peaks and therefore the reaction products generated is different. Based on the retention times and mass
spectra summarized in Table 2S, similar compounds seem to be formed in these reactions.
As shown in Table 3, the percentage of acetone and formaldehyde in the reactions with Cl and OH in the presence
of NO is higher than in experiments without NO. This indicates that, in the presence of NO, the formation of the
alkoxy radical by reaction of the peroxy radical (RO·) with NO is favored, compared to the formation of this same
alkoxy radical by self-reaction of $RO_2$. Additionally, these alkoxy radicals are chemically activated and undergo
'prompt' decomposition to form acetone, in contrast to alkoxy radicals formed by the self-reaction of $RO_2$, which are
more thermally stabilized (Atkinson et al., 2007). In the reactions of 33DMbutanone in the absence of NO, the
percentage of acetone is lower in the OH reaction than Cl reaction, which could be explained by the possible reaction
of $RO_2$ with OH radical present in the reaction medium, to form $RO_2OH$ rather than $RO_2$ reacting with itself to yield
two alkoxy radicals. The IR band observed at ~1800 cm$^{-1}$ for 33DMbutanone + Cl reaction could correspond with
the stretching vibration of the C=O (carbonyl) bond in acyl chloride. This compound can be formed by the reaction
of $RO_2$ with $Cl_2$ or Cl atoms (Ren et al., 2018). In the Cl reaction without NO, the identification of a product with a
molecular mass of m/z=116 (at $t_r$=10.26 min), assigned to 4-hydroxy-3,3-dimethyl-2-butanone (see Table 2S), is
explained by the self-reaction of two $RO_2$ radicals, leading to the formation of two molecules. In this case, the co-
product molecule to 4-hydroxy-3,3-dimethyl-2-butanone would correspond to a compound with a molecular weight
of m/z=114 (at $t_r$=8.28), assigned to 2,2-dimethyl-2-oxo-butanal. This compound is also observed in the Cl reaction
in the presence of NO, formed by the reaction of the alkoxy radical with $O_2$. The peak at $t_r$=2.55 min with a mass of
m/z=72.08 (Table 2S), assigned to hydroxyacetone, the peak at $t_r$=7.03 min with a molecular mass of m/z=102,
assigned to 1-hydroxybutan-2,3-dione, and the IR absorption band at 1105.42 cm$^{-1}$ (see Figure 10S), characteristic
of formic acid, indicate that the alkoxy radical also undergoes different isomerization processes.
The reaction of 33DMbutanone with Cl and OH in the presence of NO (at short times) and $NO_2$ (at long times) leads
to the formation of chromatographic peaks at $t_r$=5.59, 9.04, 9.68, and 14.62 min for the Cl reaction, and $t_r$=2.55,
9.04, and 14.62 min for the OH reaction, which have been assigned to nitrated compounds (also observed in the
FTIR experiments). Specifically, the peak at 5.59 minutes has been assigned to peroxyacetyl nitrate (PAN).
Additionally, based on the IR bands observed in the residual spectra at times less than 5 minutes for the Cl and OH
reaction in the presence of NO, other nitrated compounds are assigned as alkoxy nitrates (IR band around 850 cm$^{-1}$).
The formation of PAN can only be explained through the channel II, with the decomposition of the initially formed
alkoxy radical (2,2-dimethyl-3-oxobutan-1-yloxy radical). The estimated yield for PAN of 100% indicates that the
percentage of Cl attack on the -CH$_3$ of the tert-butyl group is 100%, a value very close to the 98% estimated by the
SAR method for reactions with Cl (see Table 1S).
For the reactions with OH, only acetone and formaldehyde have been quantified. These products may come from
OH attack on the methyl of the tertiary carbon of 33DMbutanone (channel II) or on the methyl directly attached to
the carbonyl group (channel I). Thus, this study cannot confirm that 94% of the reaction proceeds through channel
II, as the SAR method suggests. However, considering that the rate coefficient estimated by the SAR method is
similar to that obtained in this study, channel II can be considered the main process. From the analysis of the
SPME/GC-TOFMS experiments, the proposed reaction products also are listed in Table 3. Based on the kinetic



results and the main products obtained in this study, the reaction mechanism showed in Figure 7 is proposed for the
degradation of 33DMbutanone.

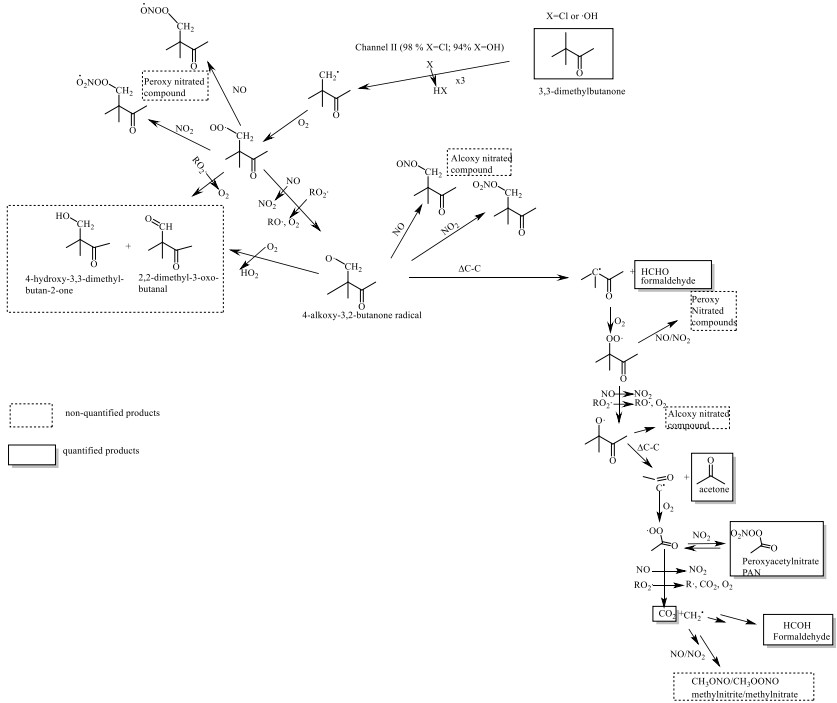


Figure 7. Mechanism proposed by the formation of the main reaction products observed for 33DMbutanone reactions
with Cl atoms and OH radical. The framed products correspond to products identified by FTIR and/or SPME/GC-
TOFMS.



### 3.2.2.2 33DMbutanal reaction products

In the analysis of SPME/GC-TOFMS chromatograms, the number of peaks observed with appreciable intensity in the reaction of Cl atoms in the presence of NO (10 peaks), in the absence of NO (8 peaks), and in the $NO_3$ reaction (7 peaks) is greater than in the reaction with OH, where only 3 peaks are observed (see Figures 13S-16S). This fact may be related to the different attack positions of the oxidant when reacting with 33DMbutanal: 3 for the reaction with Cl atoms, 2 for the reaction with the $NO_3$ radicals, and 1 for the reaction with the OH radicals, as estimated by the SAR method (see Table 1S). Some of these peaks appear at the same retention times, and their mass spectra are very similar (see Table 2S), indicating that they are the same reaction products. Some of these products are also identified in the FTIR analysis (see Table 4), such as acetone ($t_r$ = 2.2 min) and 22DMpropanal ($t_r$ = 3.08 or 3.43 min).

It is interesting to note the presence of a common peak for all reactions that appears at 9.56 min (8.95 min in the experiment with a new chromatographic column). The mass spectrum of this peak is assigned to 3,3-dimethylbutanoic acid by the NIST database software, with a similarity index of approximately 84%. This compound cannot be explained by the general scheme proposed (Scheme 2S), which is based on principles of atmospheric reactivity and bibliographic studies of similar compounds (Aschmann et al. 2010, Atkinson et al. 2007). In order to explain the formation of this peak, other pathway was proposed, where the oxy-3,3-dimethylbutyryl radical (channel I), undergoes isomerization. This pathway leads to the formation of 4-oxo-3,3-dimethylbutanoic acid, whose mass spectrum shows a fragmentation pattern very similar to the peak with a $t_r$ of 9.56 min. In order to establish which compound corresponds to this peak and taking into account that 3,3-dimethylbutanoic acid (33DMbutanoic acid) is a commercial compound, a sample of 33DMbutanoic acid was injected into the SPME/GC-TOFMS system. The analysis showed a chromatogram with a peak at approximately 9 minutes, with a mass spectrum (MS) identical to that of the peaks (9.56/8.95 min), which positively confirms the formation of this 3,3-dimethylbutanoic acid in the reactions of 33DMbutanal with atomic Cl and OH and $NO_3$ radicals (See Figure 17S). The characteristic IR bands of 33DMButanoic acid seem to be present in the FTIR residual spectra obtained for the reaction of 33DMbutanal with Cl atoms (Figure 18S). Recent studies have also detected organic acids from reactions of saturated aldehydes (Asensio et al. 2022, Bo et al. 2022).

The remaining chromatographic peaks showed in Figures13S-16S, have been assigned to reaction products shown in the general Scheme 2S. Table 3S contains all the mass spectra and their assignments. In general, the complete interpretation of mass spectra is complex because the formed products have very similar structures, leading to similar fragmentation patterns. The mass spectrum of chromatographic peaks around 8 min (7.88, 8.23, and 8.26 min) could correspond to several structurally similar products. The injection of a commercial sample of 2,2-dimethylpropanoic acid (22DMpropanoic acid) in the SPME/GC-TOFMS system shows a peak with a retention time of 8 min, whose mass spectrum correspond with the mass spectrum of the peak at 8.23 min observed in the reaction of 33DMbutanal with Cl atoms in the absence of NO. Figure 16S, shows an IR reference spectrum of 22DMpropanoic.

As can be seen in Table 3S, some of the proposed reaction products are dicarbonyls or hydroxycarbonyl compounds. The hydroxycarbonyl compounds tend to cyclize to dihydrofurans via acid-catalyzed heterogeneous reactions (Atkinson et al. 2008). For example, 4-hydroxy-3,3DMbutanal ($t_r$ = 15.78 min) can cyclize to form 2,3-dihydro-4,4-dimethylfuran. Another cyclization process can occur from an alkoxycarbonyl radical, such as the 4-formyl-2,2-dimethylbutan-1-yloxy radical, leading to the formation of 2,2-dimethyltetrahydrofuran-2-one ($t_r$ = 13.25 min).



The common chromatographic peak at 6.97/6.86 min observed in the Cl + NO and NO$_3$ reactions has been assigned
to a nitrated compound, which could correspond to peroxy-3,3-dimethylbutyryl nitrate also detected in the FTIR
analysis. The intensity of these peaks is very low, likely because the compound undergoes thermal decomposition
in the chromatograph injector. Another small chromatographic peak at 7.15 min observed in the NO$_3$ and OH radical
reactions could correspond to two nitrated compounds with the same retention time: peroxy nitrite (in the case of
the OH reaction) and peroxy nitrate (for the NO$_3$ reaction), according to the characteristic IR bands of peroxy
compounds at 793 cm$^{-1}$ observed in the FTIR spectra (see Figure 10S). Figure 10S also shows an IR band at 810
cm$^{-1}$ for the Cl + NO and OH reactions, which is characteristic of alkoxy nitrated compounds. This common product
was not detected in SPME/GC-TOFMS, possibly because it was not adsorbed onto the fiber. In general, SPME/GC-
TOFMS is not an effective sampling method for nitrated compounds.
Table 4 provides a summary of the reaction products tentatively assigned to the chromatographic peaks observed in
the experiments conducted for the reactions of 33DMbutanal with atmospheric oxidants, along with the products
identified and quantified using FTIR.
Finally, the analysis of the quantified compounds in the FTIR experiment can provide insights into the percentage
of each channel or which pathway is favored. Acetone is a reaction product that, according to Scheme 2S, is formed
from all channels. However, the similar yields in the Cl, Cl + NO, and OH reactions (28%, 21%, and 32%,
respectively) may indicate that acetone is formed from all reactions primarily through channel I. HCHO is also a
product formed through all three channels, but the highest yield of formaldehyde, along with the significant decrease
of 22DMpropanal obtained in the reaction of Cl in the presence of NO compared to the yields of these compounds
for the reaction of chlorine in the absence of NO, could indicate that, in the presence of NO, the reaction of the
peroxy species generated in channel I or channel II to form nitrated compounds is favored at the expense of the self-
reaction of RO$_2$.
The total carbon calculated for the Cl reaction in the absence of NO, based on the yields of HCHO, acetone, and
22DMpropanal, accounts for only 45%. The remaining carbon can be explained by the formation of other
compounds identified through SPME/GC-TOFMS. Additionally, the IR band at 1105 cm$^{-1}$ shown in Figure 11S in
the FTIR spectrum of the Cl atoms reaction in the absence of NO, indicates the formation of formic acid, which was
not quantified. The lower yields in the Cl reaction in the presence of NO (33%) and OH (16%) could be attributed
to the significant formation of nitrated compounds, whose yield has not been included in the total carbon calculation.
For the NO$_3$ radical reaction, the linear trend observed in Figure 5 indicates that nitrated compounds are formed
through the main reaction, and therefore, 100% of the reacted 33DMbutanal forms nitrated compounds in this
reaction conditions. The identification of other compounds in the SPME/GC-TOFMS analysis could be due to the
rapid decomposition of peroxynitrated compounds in the chromatograph's injection port or an overestimation of
nitrated compound yields. To determine if these products are generated in significant amounts, quantification is
necessary, but this is not possible due to the lack of standards or the characteristics of the SPME sampling method.
No exclusive compounds to each reaction channel (I, II, or III) have been quantified for 33DMbutanal reaction.
Therefore, it is not possible to determine the percentage of each channel through which the reaction of 3,3DMbutanal
with each oxidant proceeds. The formation of 33DMbutanoic acid in all reactions implies other reaction pathway
that has not been considered. The following reaction has been proposed as possible pathways to explain the
formation of the 3,3-dimethylbutanoic acid ((CH$_3$)$_3$CCH$_2$C(O)OH):



$(CH_3)_3CCH_2C(O)OO\cdot + HO_2\cdot \rightarrow (CH_3)_3CCH_2C(O)OH + O_3$                          (R2)
Similar reaction was proposed for $CH_3C(O)OO$ radical by some authors (Dillon and Crowley 2008, Groß et al.,
2014, Tomas et al., 2001). On the other hand, 22DMpropanoic acid (observed in the 33DMbutanal with Cl atoms)
could be a secondary product formed from the degradation of 22DMpropanal. The yields of organic acids from the
corresponding aldehyde must be very low (Bo et al. 2022), and therefore, the pathway (R2) must be a minority
channel.
From the analysis of the SPME/GC-TOFMS experiments, the proposed reaction products are also listed in Table 3.
Based on the products identified in this work, the previous bibliographic study of 33DMbutanal with the OH radical
(Aschmann et al. 2010) and considering the estimated percentages for each channel using the SAR method, the
reaction mechanism for 33DMbutanal with Cl atoms, OH and $NO_3$ radicals showed in Figure 9, is proposed to
explain the main observed reaction products in each reaction.



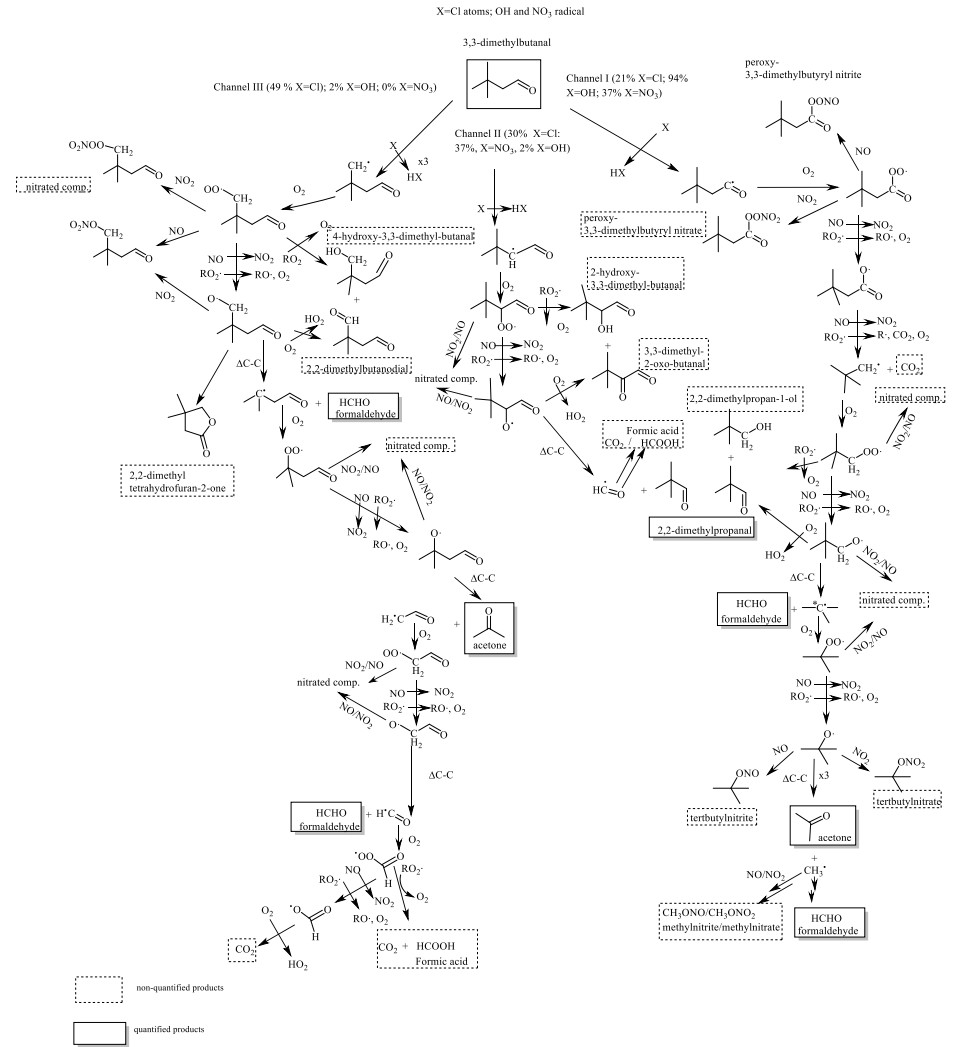

Figure 8: Mechanism proposed by the formation of the main reaction products observed for 33DMbutanal reaction with Cl atoms, OH and $NO_3$ radical. The framed products correspond to products identified by FTIR and/or SPME/GC-TOFMS.

## 4 Atmospheric Implications

The atmospheric implications of the degradation of 33DMbutanone and 33DMbutanal are determined based on their lifetimes, the effects of the reaction products generated, their influence on global warming and air quality, and consequently, their impact on health and living organisms.

The lifetime of a species in the atmosphere is calculated based on its degradation rate through various chemical processes, such as reactions with oxidants, photolysis, or deposition. In general, to estimate the atmospheric lifetime with respect to homogeneous chemical reactions with oxidants, the Eq. (II) is used:



$$\tau_{ox} = \frac{1}{k_{ox}[Ox]}$$
(II)

where $\tau_{ox}$ represents the lifetime for the considered reaction, $k_{ox}$ is the rate coefficient, and [Ox] refers to the typical
atmospheric concentration of the oxidant. The atmospheric lifetimes of 33DMbutanone and 33DMbutanal were
calculated using the rate coefficients of Table 2. For 33DMbutanal the $k_{OH}$ and $k_{NO3}$ have been the averaged from
the two rate coefficients available in the literature.
The ketones and aldehydes can significantly absorb light in the tropospheric actinic region λ>290 nm, and the
photolysis could play an important degradation process (Atkinson, 2003; Mellouki et al., 2015). There are not any
experimental data about UV-Vis absorption cross-sections of 33DMbutanal that could allow the calculation of the
photolysis rate. The value of J (0, θ) = 3× 10⁻⁵ s⁻¹ calculated for 3-methyl-butanal (similar compound to
33DMbutanal) by Lanza et al., (2008), has been used to estimate the photolysis lifetime. For 33DMbutanone a
photolysis rate of J (z, θ) = 2.4 × 10⁻⁶ s⁻¹. was estimated by Mapelli et al. (2023).
Respecting to deposition process the lifetime associated with wet deposition could be estimated with the Eq. (III)
proposed by (Chen et al., 2003):
$$\tau_{wet} = \frac{H_{atm}}{k_H v_{pm} RT}$$
(III)

where $k_H$ is the Henry's law constant; $H_{atm}$ is the height in the troposphere ($H_{atm}$ =630 m), $v_{pm}$ is the average
precipitation rate (536 mm yr⁻¹ for Spain, (http://www.aemet.es, last access: 4 October 2024), R is the gas constant
(8.14 Pa m³/mol K); and T is the temperature, considered to be constant and equal to 298 K. In the literature, there
is only one data of Henry's constant for 33DMButanone of 4.3×10⁻² mol/m³Pa (Hovorka et al., 2019, Sander, 2023).
The same data has been used to 33DMbutanal.
Taking into account all those degradation processes, a global lifetime ($\tau_{global}$) has also been calculated for
33DMbutanone and 33DMbutanal with the the equation (IV):
$$\tau_{global} = \left[ \frac{1}{\tau_{Cl}} + \frac{1}{\tau_{OH}} + \frac{1}{\tau_{NO3}} + \frac{1}{\tau_{phot}} + \frac{1}{\tau_{wet}} \right]^{-1}$$
(IV)

Table 5. Atmospheric lifetimes of 33DMButanone and 33DMButanal.

| | [a]$\tau_{Cl}$ (days) | $\tau_{Cl}$* (days) | [b]$\tau_{OH}$ (days) | [c]$\tau_{NO3}$ (days) | $\tau_{phot}$ (days) | $\tau_{wet}$ (years) | $\tau_{global}$ (days) | *$\tau_{global}$ (days) |
|---|---|---|---|---|---|---|---|---|
| **33DMbutanal** | 91 | 0.7 | 0.5 | 1.2 | 0.4[d] | 11 | 0.2 | 0.14 |
| **33DMbutanone** | 274 | 2.1 | 9.2 | - | 5[e] | 11 | 2.5 | 1.14 |

[a]Determined using the 24 h average [Cl] = 1 × 10³ atoms cm⁻³ (global average) (Platt and Jansen, 1995). *Determined using the
peak of [Cl] in coastal and industrial areas at 1.3 × 10⁵ atoms cm⁻³ (Spicer et al., 1998). [b] Determined using the 12 h average of
[OH·] = 1 × 10⁶ radicals cm⁻³ (12-hour average) (Prinn et al., 2001), [c][NO₃·] = 5 × 10⁸ radicals cm⁻³ (Atkinson, 2000). [d]Data
calculated with a J from Lanza et al., 2008. [e]Data calculated with a J from Mapelli et al., 2023.
It can be observed that the dominant tropospheric loss processes of 33DMbutanone and 33DMbutanal are their
reactions with OH radicals and photolysis process (Note that photolysis lifetime depends on the atmospheric
conditions considered), followed by their reaction with NO₃ radicals at night. However, in places where there is a





peak concentration of Cl atoms (coastal areas), the reaction with Cl atoms may compete with photolysis and reaction
with OH radicals as their main degradation process. The calculated wet lifetime of 11 years indicate that the wet
deposition can be considered negligible.
The shorter global lifetime of ~4 hours for 33DMbutanal and ~ 2 days for 33DMbutanone indicate that these
compounds are degraded near their generation sources. The products created in the degradation reactions of
33DMbutanone and 33DMbutanal may also have environmental implications. Thus, formaldehyde is classified as
potentially carcinogenic to humans (NTP, 2021). Acetone, 22DMpropanal and organic nitrates (PAN and
peroxybutyrylnitrate) are also key components in photochemical smog episodes, a major contemporary
environmental issue. The multifunctional compounds such as oxocarbonyls, dicarbonyls, hydroxycarbonyls and
acids are products with polar groups characterized by low volatility, which could facilitate the formation of
secondary organic aerosols (SOA) (Calvert et al., 2011, Asensio et al. 2022). Morever, the nitrated compounds
generated can act as NOx reservoir species, especially during the night (Altshuller, 1993) and could have an
influence at the global scale.
The potential for ozone formation of 33DMbutanone and 33DMbutanal has been evaluated calculating their
Photochemical Ozone Creation Potential (POCP) according with the method of Jenkin et al., (2017). The
Photochemical Ozone Creation Potential estimated (POCP$_E$), were 68 and 58 for conditions in NW Europe and
urban areas of the USA, respectively for 33DMbutanal and 26 and 15 for conditions in NW Europe and urban areas
of the USA, respectively for 33DMbutanone. Comparing with other series of organic compounds (Jenkin et al.,
2017), the values of POCP$_E$ for 33DMbutanal indicate that it is an important contributor to tropospheric ozone
generation.
Regarding the calculation of the GWP (global warming potential) parameter, the method of Hodnebrog et al., (2020)
and the lifetime calculated above have been used to estimate the GWP of 33DMbutanone. A value of 0.13 for a time
horizon of 20 years has been obtained, and therefore, the direct contribution to the radiative forcing of climate can
be considered negligible. The GWP for 33DMbutanal has not been calculated because its lifetime is shorter than
that of 33DMbutanone, and thus, its expected GWP is likely to be lower.
**5 Conclusions**
In this work, the rate coefficient for the reaction of 33DMbutanal with Cl atoms has been determined for the first
time. Additionally, the rate coefficients for 33DMbutanone with Cl atoms and the OH radical have been measured,
aligning with existing literature data. The kinetic findings, along with previous studies on other carbonyl compounds,
confirm that reactivity is influenced by the type of carbonyl group (aldehyde vs. ketone) and the number and position
of methyl groups. This research has expanded the database on these compounds, especially regarding their reactions
with Cl atoms.
The study of reaction products using FTIR and GC-MSTOF allows to identify and quantify acetone, formaldehyde,
and 22DMpropanal, alongside multifunctional products like hydroxycarbonyls, oxocarbonyls, and nitrated
compounds such as PAN and peroxybutyryl nitrate. The results suggest that the RO$_2$ + OH· reaction in the unpolluted
atmosphere could be significant. The proposed mechanism for 33DMbutanone indicates that hydrogen abstraction
from the tert-butyl methyl group is the primary pathway for Cl and OH, confirming SAR predictions. In the
33DMbutanal reaction, hydrogen abstraction occurs from various functional groups depending on the reacting





species (Cl atoms, NO$_3$, and OH radicals), also aligning with SAR predictions. The positive identification of
33DMbutanoic acid implies a pathway in the reaction mechanisms of 33DMbutanal, that initially have not been
considered.
The atmospheric conditions determine the reaction products obtained in the atmospheric degradation of
33DMbutanal and 33DMbutanone. Thus, in polluted environments with high concentrations of NOx, nitrated
organic compounds (RONO$_2$) are formed. Moreover, when the concentration of NO$_2$ is higher than that of NO,
ozone is formed. In a clean atmosphere, as in the case of the experiments with Cl atoms in the absence of NOx, the
reaction products are hydroxy/oxo carbonyl compounds.
Atmospherically, both 33DMbutanal and 33DMbutanone degrade within a few hours and 2 days respectively during
the day, implying that degradation happens close to the emission sources. Their direct contribution to radiative
forcing is minimal. However, their estimated POCP values suggest a potential role in tropospheric ozone formation,
especially for 33DMbutanal. The multifunctional products formed may contribute to secondary organic aerosol
formation, and their further oxidation in the troposphere could enhance photochemical smog, impacting air quality
and human health.
**Data availability.** The underlying research data are available upon email request from the contact author of this
work.
**Supplement.** The electronic Supplement includes additional tables and figures.
**Author contributions.**
**Inmaculada Aranda:** Formal analysis, validation, investigation, methodology, writing-original draft. **Pilar**
**Martín:** Conceptualization, supervision, methodology, writing-original draft**: Sagrario Salgado** Conceptualization,
supervision, methodology, writing-original draft**. Florentina Villanueva:** Supervision, methodology. **Beatriz**
**Cabañas:** Conceptualization, supervision, funding acquisition.
**Competing interests**
The contact author has declared that none of the authors has any competing interests.
**Acknowledgments**
The authors thank for financial support. I. Aranda thanks UCLM for funding her research contract (Plan Propio de
I+D+i) cofinanced by FSE.
**Financial support.** This research has been supported by the Ministry of Science, Innovation and Universities
(Project RTI 2018-099503-B-I00) and the Junta de Comunidades de Castilla-La Mancha (Project SBPLY/21/180501

725    /000283).

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
