# Peer review of "Reactivity study of 3,3-dimethylbutanal and 3,3-dimethylbutanone: Kinetic, reaction products, mechanisms and atmospheric implications"

_EGUsphere, 2024_

## Referee Comment (RC1)

This study focuses on the atmospheric reactivity of 3,3-dimethylbutanal (3,3-DMButanal) and 3,3-dimethylbutanone (3,3-DMButanone), two carbonyl compounds that may significantly impact atmospheric chemistry. The research investigates their degradation mechanisms with atmospheric oxidants such as Cl atoms, OH, and $NO_3$ radicals. Using Fourier Transform Infrared Spectroscopy (FTIR) and Gas Chromatography-Mass Spectrometry (GC-MS), the study measures reaction rate coefficients and identifies primary reaction products. Additionally, it assesses the environmental implications of these compounds, particularly their roles in tropospheric ozone formation and secondary organic aerosol (SOA) generation. However, there are still some problems with the manuscript.

1. Introduction Part. While the article mentions the environmental and industrial sources of 3,3-dimethylbutanal and 3,3-dimethylbutanone, it lacks specific data on their typical concentrations and distributions in the actual atmospheric environment. It is better to add relevant background information to help readers better understand the importance of these compounds in the atmosphere.

2. Even if the authors use the same reference compound, the parallel experimental results will get a large error in the rate constant. Is this a reasonable margin of error? How does the author explain this error?

3. How is the error calculated for each rate constant? Please elaborate.

4. The clarity of Figures 2 and 3 is not enough. Please adjust the image resolution to make the picture more visible.

5. Is the peak time of each product compared with the standard peak time of the national standard?

6. The mechanism diagrams in Figures 7 and 8 are quite confusing and recommend readjustment.

7. The author has repeatedly mentioned that these aldehydes may contribute to the production of SOA, so does the author consider supplementing the SOA experiment in this system to better evaluate its contribution to SOA?

8. Combined with SAR prediction, the authors point out that methyl groups adjacent to tert-butyl groups are the main reaction path, especially in the oxidation of OH radicals and Cl atoms, but many studies have pointed out that aldehyde groups are the most important and

possibly the only core reaction path in aldehydes. How do the authors consider this problem?

9. In Section 3.1 Kinetic study, the author points out that some wall losses, photolysis losses, etc. can be ignored. Please give specific experimental results and present them in pictures to intuitively understand the wall losses.

---

## Referee Comment (RC4)

This paper deals with the kinetics and mechanisms of two carbonyl compounds (3,3-dimethylbutanone and 3,3-dimethylbutanal) with OH, Cl and NO3. The kinetics of 33DMbutanal with Cl atoms at ambient temperature are determined for the first time, while for the other kinetics (33DMbutanone with Cl atoms and OH radicals), values were already reported in the literature. A set of reaction products has been identified, with a (semi)quantification for a few of them.

My main concern is that many statements consist of suggestions that are not (fully) supported by results. Some of the conclusive remarks are not really convincing. The kinetics present scattering in the results that needs to be clarified.

Overall, the manuscript is quite long and there are many typos that need to be corrected (I reported only a few below).

**Comments**

Line 34: it is stated that "large carbonyl compounds can influence on climate change if they are strong infrared light absorbers by altering the Earth's radiation balance": I don't agree: the atmospheric concentrations of carbonyls are too small to influence the climate by IR light absorption.

3.1 Kinetic study: did you check that the reaction of the compound (resp. the reference) with the oxidant did not form products absorbing at the selected IR bands for the reference (resp. for the compound)?

Why not showing all the relative kinetic plots (in Supplementary Material)?

Schemes S1 & S2: HO2 radicals are formed in the mechanisms (through RO+O2 for example): why not including the RO2+HO2 reactions in the mechanisms (see also line 606)? Especially for secondary and even more tertiary RO2, where RO2+RO2 rate constants are very low... Besides, these mechanisms could be advantageously simplified, by removing low reactivity channels (for example RC(O)O radicals mainly decompose into R+CO2, (CH3)3CO2+(CH3)3CO2 is negligible, etc.). What about direct photolysis of a-dicarbonyls, which has been shown to be significant?

Table 1: there are significant discrepancies between the k values obtained using the different reference compounds, especially with 33DMbutanone+OH (from 0.96 to 1.92) and with 33DMbutanone+Cl (from 3.03 to 5.30). This need to be discussed. How well known are the reference rate constants?

Line 321: Figure 2: what do you mean with "64 % of the conversion and a 29% respectively"? What are the experimental conditions of the experiments? Could you indicate the concentrations of the references? This figure is difficult to read (small size, bright colours, IR spectra mixed).

Line 333: it is stated that "Additionally, IR bands of N2O5 (precursor of NO3) have been observed in the reaction with Cl atoms": referring to Figure S2, we cannot see any N2O5 bands appearing in the product spectra...

Line 365: "the concentrations of acetone and nitrated compounds increase more than expected": what are actually the expected trends??

Figure 3S: what are the differences with Figure 4 (a) and (b)??

Figure 5S: OH reaction in absence of NO: this is from H2O2 photolysis, right? Did you carry out OH reaction in presence of NO (as seems from line 118)? Is there a yield plot also?

Figure 5: comparing (a) and (b): why are the plots of nitrated compounds curved in one case (a) and linear in the other case (b)?

Figure 7S: how do you explain the shift on the x-axis?

Table 3: how did you calculate the uncertainties for the average yields? Why so large differences with those in Table 4??

Lines 421-423: "reaction of 33DMbutanone with Cl atoms in the presence of NO the total carbon is 95% (not accounting for nitrated compounds)," : how is it possible to get almost 100% carbon balance without taking into account nitrated compounds (that seem to be significant from Table 3)?

Table 4: for "33DMbutanal + NO3": no yields are reported: could the authors calculate upper limits (based on their sensitivity)? Especially since acetone is detected by GC... Why is the yield of nitrated compounds exactly = 100%??

Tables 3 & 4: calculation of the carbon budget: I suggest explaining it in the main text, not in the notes of the Tables, since it is not very clear as now...

Lines 424-426: from Figure S9, it is said that formic acid and hydroxyacetone were observed. While this seems to be valid of formic acid, I cannot see any features for hydroxyacetone... Besides, formic acid is often observed in such experiments when formaldehyde is present; it may be formed from secondary wall reactions of CH2O...

Figure 12S: "generated chromatograms for the 33DMbutanone + OH +NO at different reaction times": what are these reaction times? It is noted "8 min" for all last 3 chromatograms... What do you mean with "generated"?

Line 460: "a little peak that appear together with the peak of air, that could correspond to acetone.": why not calibrating directly with pure acetone instead of "creating a chromatogram using a specific tool"??

Lines 462-467: this descriptive part concerning the "specific tool" for creating chromatograms is absolutely unclear. Which software? What about reliability? What is the advantage for acetone compared to direct calibration?

Lines 492-493: "the possible reaction of $RO_2$ with OH radical": could you estimate the contribution of these reactions relative to RO2 + RO2?

Line 508: PAN is known to be unstable at high temperature: how can you attribute the peak at 5.59 min to PAN, knowing that the SPME fiber is heated at 250°C in the GC injector? See also lines 569-570.

Lines 512-514: PAN is formed after a series of almost 12 successive reactions (scheme 1S): how can you affirm that a PAN yield of 100% is consistent with the SAR estimated branching ratio of 98% for channel II??

Line 543: isomerization of oxy-3,3-dimethylbutyryl radical: is this "new" pathway supported by literature studies? Because decomposition of such oxy acyl radicals (into alkyl + CO2) is considered to be extremely fast. I cannot see this suggested reaction on Figure 8, why? Indeed, another pathway was suggested farther (lines 602-607). Why then concluding (lines 696-698) that "The positive

identification of 33DMbutanoic acid implies a pathway in the reaction mechanisms of 33DMbutanal, that initially have not been considered."?

Lines 545-549: mass spectrum of 33DMbutanoic acid compared to mass spectrum of the peak at 9.56 min: you indicated that these 2 spectra match well ("correspond"): could you support this by showing the comparison in Supp Mat?

Lines 550-552: Figure 18S is not convincing...

Lines 558-561: mass spectrum of 22DMpropanoic acid compared to mass spectrum of the peak at 8.23 min: you indicated that these 2 spectra match well ("correspond"): could you support this by showing the comparison in Supp Mat?

Lines 582-583: I don't understand how you conclude that acetone comes mainly from channel I *because* the yields are similar... this may be fortuitous?

Line 596: "100% of the reacted 33DMbutanal forms nitrated compounds": this is not demonstrated; this is only suggested because no other compounds were detected.

Lines 636-638: photolysis rates: it should be specified if the photolysis quantum yield has been taken as 1.

Lines 692-693: the conclusion that "the results suggest that the RO2 + OH· reaction in the unpolluted atmosphere could be significant." is bold, because it is only based on the difference of acetone yields between Cl and OH systems...

**Minor comments/Typos**

Line 30: established

Line 36: remove ';' at the end of the sentence, before the closing bracket

Citations of references are not homogeneous through the MS; please check ("XX et al., XXX" with point and comma).

Lines 129-130: remove capital letters for the compound's names (same thing lines 136-139)

Line 150: "the losses of the reactants..."

Line 151: the loss of < 3% is over how much time?

Table 1: note d is not explained. Notes a,b,c: the statement "kcarbonyl and kR is given in $10^{-10}$, $10^{-11}$, $10^{-12}$, respectively" is not clear: we have 2 subjects (kcarbonyl and kR) and 3 possibilities ($10^{-10}$, $10^{-11}$, $10^{-12}$)!

Lines 181 & 182: use "×" instead of "x" (check all the MS)

Line 182: the statement "This is the general trend observed in the atmospheric chemistry for the oxidation reactions of organic compounds; kCl>kOH>>kNO3" needs at least one reference.

Consistency between Table 1 and 2: the uncertainties in kCl for 33DMbutanal (this work) are not the same. Same for kOH for 33DMbutanone.

Line 210: please homogenize the naming of 3-methylbutan-2-one all through the MS (see Table 2 where it is noted 3-methylbutanone).

Line 217: what is the "atmospheric radical"?

Line 222: "more favoured than H atom"

Line 224: "33DMbutanone"

Line 232: "double bond"

Line 238: "coefficients"

Line 241: "kestimated=22.12x10-12": please keep only 3 significant figures and write in scientific style

Line 245: "factors"

Line 254: "Schemes"

Line 265: "major"

Line 271-272: the sentence is not clear

Line 276: "neutral compounds": you mean "closed-shell"?

Line 286: "establish"

Line 303: "Colmenar et al., 2018"

Figures 3S & 5S, caption: "reacted"

Lines 391-396: Table 3 caption is reported twice.

Table 3: GS-TOFMS, Reaction products: "Hydroxy-2,3-butanedione"

Notes in Table 3: why using note (1) and * instead of continuing with "h"? Same comment for Table 4.

Table 4: "33DMbutanal + OH": "24.9±0.8"

Line 409: "products identified"

Line 419: "only"

Line 429: "GC-MSTOF": sometimes you write "GC-TOFMS": please be consistent!

Line 431: "The yields should be considered with caution": but no yields are reported for GC!!

Lines 438-439: already said in lines 434-437.

Figure 6: (a) and (b) look almost exactly the same while they are reported to be "33DMbutanone + Cl atoms (a) in the absence of NO and FI mode, (b) in the presence of NO"? How can you say that "(a) present more peaks than ... (b)" (line 446)?

Line 447: "This indicates that the presence of NO influences the reaction mechanism.": this is true but not very new...

Line 456: "In some cases": which cases?

Line 483: "can see that": no subject in the sentence

Lines 484-485: not consistent: "the reaction products generated is different" and "similar compounds seem to be formed"

---

## Author Response (AR1)

Ciudad Real, January 28th

Resubmission of Revised Manuscript, egusphere-2024-3241

Dear Editor

We are pleased to resubmit our revised manuscript, titled "Reactivity study of 3,3-dimethylbutanal and 3,3-dimethylbutanone: Kinetic, reaction products, mechanisms and atmospheric implications" for further consideration in Atmospheric Chemistry and Physics journal.

We would like to extend our heartfelt thanks to you and the reviewers for the detailed and insightful feedback provided during the review process. These comments and suggestions have been instrumental in guiding us to significantly improve the manuscript.

The revisions to the manuscript have been substantial, particularly in response to feedback regarding the section on reaction products. Following suggestions from several reviewers, especially Reviewer #5, we have completely restructured this section to enhance its clarity, coherence, and alignment with the high standards of Atmospheric Chemistry and Physics journal. These changes reflect a comprehensive reorganization of the discussion and analysis, which we believe greatly improves the manuscript's overall quality.

It is important to note that our responses to Reviewers #1, #2, and #3 were completed and posted in the online revision, prior to receiving feedback from Reviewers #4, #5, and #6. As a result, our initial responses do not reference the subsequent modifications made in the manuscript based on the later feedback. However, we have carefully addressed all comments from Reviewers #1, #2, and #3 in this revised submission and included these changes in the updated manuscript.

For your convenience, we have prepared a detailed response document outlining how each reviewer's comments have been addressed. We are confident that the extensive revisions have significantly improved the clarity and impact of our work.

Once again, we thank the reviewers for their valuable input and you for overseeing this process. Please do not hesitate to contact us if additional clarifications or revisions are needed.

Sincerely,

Pilar Martin

**The authors thank you for dedicating your time to reviewing our work, and we greatly value the opportunity to discuss the results. Below, we present our point-by-point response to your comments.**

Referee #1.

Introduction Part. While the article mentions the environmental and industrial sources of 3,3-dimethylbutanal and 3,3-dimethylbutanone, it lacks specific data on their typical concentrations and distributions in the actual atmospheric environment. It is better to add relevant background information to help readers better understand the importance of these compounds in the atmosphere.

**Response of authors.**

**Effectively, It would be relevant to indicate the data of concentrations of these compound present in the atmosphere. However, the literature review about this, indicates that these compounds are not emitted in large quantities, or they have not been measured. In the study by Mapelli et al 2023, on 3,3-dimethylbutanone, no data on concentrations found or measured in the atmosphere were reported."**

2. Even if the authors use the same reference compound, the parallel experimental results will get a large error in the rate constant. Is this a reasonable margin of error? How does the author explain this error?

**Response of authors.**

**It cannot be considered surprising this degree of error. Yes, this margin of error is typical for these types of experiments where the rate coefficient is expressed in units of $cm^3$ $molecule^{-1}$ $s^{-1}$, especially when the kinetic method used to determine the rate coefficient is a relative method. In this approach, there is a systematic error associated with the subtraction process that cannot be precisely quantified.**

**This error can significantly affect slow reactions. This is the case for the reaction of 3,3-dimethylbutanone (33DMbutanone) with OH radicals, where the rate coefficient is on the order of $10^{-12}$ $cm^3$ $molecule^{-1}$ $s^{-1}$, making it a slow reaction compared to other OH reactions (typically k~$10^{-11}$ $cm^3$ $molecule^{-1}$ $s^{-1}$). Additionally, in the reaction with OH, where methyl nitrite is used as the OH radical precursor, an additional systematic error may arise due to the overlapping of IR bands from methyl nitrite and the reaction products with those of 33DMbutanone and reference compounds. This is why multiple reference compounds were used in this study.**

**Upon reviewing the data in Table 1, we have identified an error in the reported rate coefficient for the reaction of 33DMbutanone with OH when 1-butanol was used as the reference compound. The correct value is (1.80±0.37), which has already been corrected in Table 1.**

**As shown in Table 1, there are 13 rate coefficient measurements for the reaction of 33DMbutanone with OH. The final rate coefficient can be calculated using either a simple average or a weighted average. However, because these data correspond to different reference compounds and experiments conducted on different days, the authors have chosen to use the weighted average.**

The weighted average obtained is $(1.25\pm0.05)\times10^{-12}$ cm$^3$ molecule$^{-1}$ s$^{-1}$. This value closely aligns with the data reported by Mapelli et al. (2023) $(1.2\pm0.2)\times10^{-12}$ cm$^3$ molecule$^{-1}$ s$^{-1}$ and the recommended value from McGillen (2020) $(1.21\pm0.05)\times10^{-12}$ cm$^3$ molecule$^{-1}$ s$^{-1}$. Therefore, the authors believe that our result provides a reliable rate coefficient for the reaction of 33DMbutanone with OH.

3-How is the error calculated for each rate constant? Please elaborate.

**Response of authors.**

**The total absolute error σ(k$_{carbonyl}$) is a combination of the statistical errors from the regression analysis (σ$_{slope}$) and the quoted error in the value of the rate coefficient of the reference compound (σ$_R$).**

$$\sigma\left(k_{carbonyl}\right) = \sqrt{(k_R * \sigma_{slope})^2 + (slope * \sigma_{k_R})^2}$$

**where pte y σ$_{pte}$ are the slope and the associated error, and k$_R$ and σ$_{k_R}$ are the reference coefficient and its error. The final values of the rate coefficients and the associated error were calculated as a weighted average.**
**I have to mention that when working with different reference compounds it is necessary to consider the error associated with averaging the results obtained with each of them. To do this, we work with weighted mean and standard deviation values with the following equation:**

$$\bar{x} = \frac{\sum_{i=1}^{N} w_i x_i}{\sum_{i=0}^{N} w_i}$$

**Where x$_i$ are the obtained rate coefficients and w$_i$ is the associated weight given by:** $w_i = \frac{1}{\sigma^2}$
**The standard deviation associated with the weighted average is given by:**

$$\sigma = \sqrt{\frac{1}{\sum_{i=1}^{N} w_i}}$$

**We have not considered it necessary to provide all the details of the error calculation in this article since it has already been extensively explained in previous articles, (Colmenar et al. 2020 (https://doi.org/10.5194/acp-20-699-2020) Aranda et al. 2024 https://doi.org/10.1016/j.atmosenv.2024.120420**

4. The clarity of Figures 2 and 3 is not enough. Please adjust the image resolution to make the picture more visible.

**Response of authors.**

**We will attempt to obtain figures with higher resolution and quality. However, in the case of Figure 2, since there are so many spectra together, it has been difficult to improve the quality while maintaining the full spectrum (from 800 cm$^{-1}$ to 3000 cm$^{-1}$).**

5. Is the peak time of each product compared with the standard peak time of the national

standard?

**Response of authors.**

We would need to clarify what we mean by the term 'standard' we are referring to a commercial compound, not a national standard. To avoid confusion, the manuscript will be updated by replacing the word "standard" with "commercial compound."

6. The mechanism diagrams in Figures 7 and 8 are quite confusing and recommend

readjustment.

**Response of authors.**

**We do not really understand what do you mean with the term 'readjustment'. The proposed mechanism corresponds to the possible attack routes of the atmospheric oxidant with the organic molecule and the subsequent evolution to the products. Please, could you please specify what you mean by 'readjustment'?**

7. The author has repeatedly mentioned that these aldehydes may contribute to the production of SOA, so does the author consider supplementing the SOA experiment in this system to better evaluate its contribution to SOA?

**Response of authors.**

**The authors indeed find it highly interesting to conduct studies aimed at determining SOA generation. Currently, the instrumentation required for these experiments, specifically an Aerosol Mass Spectrometer, is an equipment recently acquired but that is not available for use in our laboratories in the near future, it is part of an ongoing project, after a process of calibrating and optimizing this experimental system.**

8. Combined with SAR prediction, the authors point out that methyl groups adjacent to

tert-butyl groups are the main reaction path, especially in the oxidation of OH radicals and Cl atoms, but many studies have pointed out that aldehyde groups are the most important and possibly the only core reaction path in aldehydes. How do the authors consider this problem?

**Response of authors.**

**It may be that this explanation is not clearly conveyed in the manuscript. What we aim to express is that for 33DMbutanone (ketone compound), the SAR method (see Table S1 in the supplementary material) and the results of this study, indicate that the reaction predominantly initiates with the oxidant (atomic chlorine/hydroxyl radical) abstraction of the hydrogen from the methyl group adjacent to the tert-butyl group. In contrast, for the reaction of 33DMbutanal (aldehyde compound), the hydrogen abstraction depends on the specific oxidant. For the reaction with atomic chlorine, the hydrogen abstraction can occur from any group (with a similar percentage of attack on the CH3-, CH2-, and C(O)H groups). For $NO_3$, hydrogen abstraction primarily occurs at the aldehydic hydrogen (63%), followed by the hydrogen in the CH2 group (37%). For the OH radical, 94% of the abstraction occurs at the aldehydic hydrogen.**

**In this regard, if the referee considers it not sufficiently clear, the text in the manuscript will be revised in the final version.**

**We want to indicate that, in the scheme shown in Figure 8, we have detected an error in the percentage for Channel I of the nitrate radical reaction which has already been corrected.**

9. In Section 3.1 Kinetic study, the author points out that some wall losses, photolysis losses, etc. can be ignored. Please give specific experimental results and present them in pictures to intuitively understand the wall losses.

**Response of authors.**

**The methodology used for the kinetic study involved a series of preliminary experiments to determine the loss processes due to wall adsorption, photolysis, etc. The authors considered these loss processes negligible when the percentage variation in the reactant concentration was less than 3%.**

**To determine these losses using FTIR, the procedure involved comparing the initial spectrum of the reactant with the spectrum recorded after a long period (45 minutes, which is the typical time range for the kinetic study of 33DMbutanal and 33DMbutanone). If the variation was less than 3%, the authors deemed these processes insignificant and concluded that they did not need to be accounted for in the kinetic study.**

**Only in cases where the comparison showed a variation greater than 3%, the rate coefficients for these loss processes were determined using the equation:**

**$\ln([\text{reactant}]_0[\text{reactant}]_t)=kt$**

**For the study of 33DMbutanal and 33DMbutanone, the percentage variation in concentration was less than 3% in all cases. Therefore, the loss coefficients were not calculated, and we do not have any graphs of the mentioned equation. However, as an example, I am sharing a screenshot below where it can be observed that the spectrum of 33BMbutanal at time zero and after 45 minutes of photolysis does not change by more than 3%. (subtraction factor 0.9931)**

[Figure]

Since the preliminary experiments conducted for the kinetic study of 33BMbutanone and 33DMbutanal showed that the loss processes of these reactants (including the reference compounds) are negligible, the authors believe it is unnecessary to provide further information on this matter.

However, if the referee considers it necessary, we will include the following sentence in the manuscript revised *"To determine these loss processes, the initial spectrum of the reactants was compared with the spectrum recorded after a long period (45 minutes, which is the typical time range for the kinetic study of 33DMbutanal and 33DMbutanone)"* ..

Referee #2

The manuscript describes results from laboratory based experiments to elucidate the gas phase atmospheric degradation chemistry of two VOC, namely 3,3-dimethylbutanal (henceforth DMBal in this review) and 3,3-dimethylbutanone (DMBone). There is much novel and valuable data presented here, but some improvements in presentation and discussion are required.

The introduction sets out the likely atmospheric sources and sinks of the two VOC, and outlines previous work. The experimental section is a little lacking in detail. For example, it may be that the photochemistry used to generate OH from CH3ONO is familiar to the authors, but it was not to this reviewer (nor many readers). If you are not going to elaborate here, at least include a reference to a literature paper describing these details. Further, I am not familiar with lamps of output 350 nm. Please describe.

**Response of authors.**

**The authors acknowledge that certain details about the experimental process are missing. However, we believe that the manuscript is already sufficiently extensive and further elaboration on aspects described in the literature (Finlayson-Pitts and Pitts 2000). Nevertheless, if the reviewer deems it necessary, the next information will be included in the final version:**

**"The oxidants were generated by photolysis or decomposition of a precursor (Finlayson-Pitts and Pitts 2000).**

**The generation of chlorine atoms was achieved through the photolysis of Cl2 using radiation emitted by actinic lamps ($\lambda_{max}$ = 360 nm).**

**Cl$_2$ + hv $\rightarrow$ 2 Cl                          (R1)**

**Hydroxyl radicals (OH) were generated by the photolysis of methyl nitrite (CH$_3$ONO) in air in the presence of NO, following the reaction sequence below:**

**CH$_3$ONO + hv$\rightarrow$ CH$_3$O$\cdot$ + NO          (R2)**

**CH$_3$O$\cdot$ + O$_2$$\rightarrow$ HCHO + HO$_2$$\cdot$          (R3)**

**HO$_2$$\cdot$ + NO $\rightarrow$ NO$_2$ + $\cdot$OH          (R4)**

**Some experiments have been carried out using H$_2$O$_2$ as precursor of OH and UV radiation ($\lambda$=254 nm) in a Quartz gas cell reactor.**

**H$_2$O$_2$+ hv $\rightarrow$ 2 OH                          (R5)**

**Nitrate radicals (NO$_3$) were generated through the thermal decomposition of dinitrogen pentoxide (N$_2$O$_5$) at room temperature, according to the following equilibrium**

**N$_2$O$_5$ $\rightleftarrows$ NO$_2$+$\cdot$NO$_3$                          (R6)"**

There is a wealth of data presented in the results section, useful both for rate coefficient determinations and for product yields. Data presented in plots (e.g. Fig. 1) appear to be of high quality. My main concern regarding this manuscript concerns the data presented in

Table 1, and the associated discussion. The data for DBBal + Cl looks good, both as presented in Fig. 1 and in Table 1. However, for determinations of k for the other two reactions, there appears to be an unusually wide spread of results. For DMBone + Cl, these values (in 10-11 cm3 molecule-1 s-1) range from 3.03 to 5.22. The results obtained using 2-methyl-2-butanol as a reference VOC all cluster at the lower end of this range of results, whereas results using other reference compounds tend to agree on k ~ 5x10-11 cm3 molecule-1 s-1. These observations are surely worthy of comment in the following text, together with some analysis of the provenance of the reference k-data. Is the data on 2-methyl-2-butanol + Cl (or any other of the reference reactions) well established? All worth a few more sentences, though to conclude on this reaction, if no prior studies were available then results from this work represent a considerable contribution to our understanding of atmospheric chemistry.

More concerning is kinetic data in Table 1 on DMBone + OH (Table 1). Results from this work range from k / 10-12 cm3 molecule-1 s-1 = (0.96 +/- 0.11) to (1.92 +/- 0.59), exactly a factor of two, considerably larger than I would expect from a relative rate study.

There is no clear sense here that the reference reactions are responsible for this inconsistency. Nor (if the data quality in Fig. 1 is in any way exemplary), was this likely a result of random noise from experiment to experiment. So, having ruled out two potential problems, we should consider other sources of error. Was there another experimental factor exerting a malign influence on the data? I note that we cannot tell from Table 1 whether experiments were conducted at 254 nm (where photolytic effects may be serious) or 350 nm, where the more complex precursor chemistry may introduce other secondary effect. Details such as precursors used should be included in Table 1. The uncertainties quoted for each experiment indicate that data quality could be quite varied; were all datasets nonetheless proportional in appearance – all having intercepts of zero and no evident curvature? Perhaps all such plots should be included in the S.I. Were there any difficulties in differentiating FTIR peaks of DMBone from precursors or products? An overall weighted average yielded (1.25 +/- 0.05)x10-12 cm3 molecule-1 s-1. The surprisingly small (4%) uncertainty in k does not appear to reflect the inconsistencies encountered from one experimental determination to the next. There is more analysis needed and more discussion around the above points required to justify the closing statement from line 179 "These data are in good agreement with the values obtained in this study, thereby contributing to the accurate determination of the rate coefficients."

**Response of authors.**

**When the FTIR system is used to determine the rate coefficient, a systematic error associated with the subtraction process arises, which cannot be precisely quantified. To minimize this systematic error, it is important to avoid overlapping of IR bands from the radical precursor, reference compounds, and reaction products. In many cases, it is challenging to find a reference compound whose characteristic IR bands do not overlap with those of the organic compound (in our case, 3,3-dimethylbutanal and 3,3-dimethylbutanone).**

**Additionally, the slower the reactions, the smaller the variations in the absorbance of the characteristic IR bands of the reacting compounds, making the subtraction factors determined at time t and t' very similar. In such cases, small changes in determining**

the subtraction factor can significantly influence the calculation of the relative rate coefficient ($k_{carbonyl}/k_{Rererence}$) and therefore in the calculation of $k_{carbonyl}$. For this reason, four reference compounds were used in the reactions of 33DMbutanone to determine the rate coefficient ($k_{carbonyl}$) as accurately as possible.

As shown in Table 1, there are 13 rate coefficients measurements for reaction of 33DMbutanone with Cl and OH.

Upon reviewing the data in Table 1, we identified an error in the reported rate coefficient for the reaction of 33DMbutanone with OH when 1-butanol was used as the reference compound. The corrected value is (1.80±0.37) instead of 0.96±0.11. It has been modified in Table 1 in a revised manuscript.

The final rate coefficient can be calculated using either a normal average or a weighted average. However, because these data correspond to different reference compounds and experiments conducted on different days, the authors have chosen to use the weighted average.

The total absolute error $\sigma(k_{carbonyl})$ is a combination of the statistical errors from the regression analysis ($\sigma_{slope}$) and the quoted error in the value of the rate coefficient of the reference compound ($\sigma_R$).

$$\sigma(k_{carbonyl}) = \sqrt{(k_R * \sigma_{slope})^2 + (slope * \sigma_{k_R})^2}$$

where pte y $\sigma_{pte}$ are the slope and the associated error, and $k_R$ and $\sigma_{k_R}$ are the reference coefficient and its error. The final values of the rate coefficients and the associated error were calculated as a weighted average.

I have to mention that when working with different reference compounds it is necessary to consider the error associated with averaging the results obtained with each of them. To do this, we work with weighted mean and standard deviation values with the following equation:

$$\bar{x} = \frac{\sum_{i=1}^{N} w_i x_i}{\sum_{i=0}^{N} w_i}$$

Where $x_i$ are the obtained rate coefficients and $w_i$ is the associated weight given by: $w_i = \frac{1}{\sigma^2}$

The standard deviation associated with the weighted average is given by:

$$\sigma = \sqrt{\frac{1}{\sum_{i=1}^{N} w_i}}$$

Regarding your question about the rate coefficients of all reference compounds, these are the corresponding to the database of McGillen et al 2020 (https://essd.copernicus.org/articles/12/1203/2020/ )

In the case of Cl reaction the weighted average obtained is (4.22±0.27)×10$^{-11}$cm$^3$ molecule$^{-1}$ s$^{-1}$. The rate coefficient obtained for Cl atoms is very similar to recommended value from McGillen (2020) (4.48±0.5)×10$^{-11}$ cm$^3$ molecule$^{-1}$ s$^{-1}$.

In the case of OH reaction the weighted average obtained is (1.25±0.05)×10$^{-12}$cm$^3$ molecule$^{-1}$ s$^{-1}$, again closely aligns with the data reported by Mapelli et al. (2023) (1.2±0.2)×10$^{-12}$ cm$^3$ molecule$^{-1}$ s$^{-1}$ and the recommended value from McGillen (2020) (1.21±0.05)×10$^{-12}$ cm$^3$ molecule$^{-1}$ s$^{-1}$.

**Therefore, the authors believe that our result provides a reliable rate coefficient for the reaction of 33DMbutanone with Cl and OH.**

**Regarding the comment on secondary reactions, such as the photolysis of 3,3-dimethylbutanal and 3,3-dimethylbutanone when irradiated at $\lambda_{max}$=360 nm (254 nm was not used in kinetic experiments), or the reaction between carbonyl compounds and radical precursors (Cl$_2$ or CH$_3$ONO), it should be noted that preliminary experiments were conducted to evaluate these processes. The results of these experiments demonstrated that such contributions are negligible, since the variation in the concentration of the reactants (carbonyl compounds and references compounds) at time zero and 45 minutes is less than 3%.**

**Moreover, if these processes were significant, the plot of the equation used to determine the $k_{carbonyl}/k_{reference}$ ratio would exhibit curvature or a significant origin intercept. In our experiments, the plots showed linear fit with $r^2$~0.99 and low origin intercept values, indicating that these secondary processes do not affect the results.**

**The authors will consider including the representations in the supplementary material as suggested by the reviewer. However, given that there are numerous graphs, resulting in a large amount of information we have made a selection. All the information will be available on demand.**

There follows good discussion of k results, both in terms of the reactivity of different oxidants, of different functional groups (aldehyde and ketone) and of the impacts of structural changes within the VOC on reactivity. However, I found the absence of any comparison with one of the most recent and sophisticated SAR formulations surprising. The authors should compare results obtained here with those calculated using Jenkin et al. (2018) doi.org/10.5194/acp-18-9297-2018

**Response of authors.**

**As indicated, the rate constant estimated for the reaction of 3,3-dimethylbutanone and 3,3-dimethylbutanal with OH radicals was calculated using the EPA's EPI Suite™ software (https://www.epa.gov/tsca-screening-tools/epi-suitetm-estimation-program-interface), specifically the AOPWIN™ model within EPI Suite™. This model estimates the gas-phase reaction rate for interactions between hydroxyl radicals (the most prevalent atmospheric oxidant) and a given chemical. The software version used is updated as of 2023. The $k_{estimated}$ (cm$^3$ molecule$^{-1}$ s$^{-1}$) were 22.12x10$^{-12}$ for 33DMbutanal and 1.69x10$^{-12}$ for 33DMbutanone.**

**For the reaction of 33DMbutanone with OH, the rate constant was also estimated based on the values from Jenkin et al. (2018), obtaining a value of $1.45 \times 10^{-12}$ cm$^3$ molecule$^{-1}$ s$^{-1}$. However, we considered that using the software was sufficient.**

**Thank you for the suggestion.**

The product studies for both reactions appear commendably detailed. Methods appear sound. My only concern here was the unidentified problem with the DMBone kinetic data (see above). This may have derived from FTIR retrievals. Might this propagate into errors in product yields?

**Response of authors.**

**Could the reviewer please specify or clarify their question? I am not entirely sure what they are referring to.**

There follows a discussion of atmospheric implications. The points raised here all seem reasonable, as do the various estimates of lifetimes and POCP. One point to note would be that photolysis lifetimes were estimated elsewhere based upon measured spectra, but that no quantum yield data was available. There are consequently large uncertainties in the rate of photolysis for either of these VOC.

**Response of authors.**

**For 3,3-dimethylbutanal (33DMbutanal), the photolysis rate coefficient is indeed an estimated value based on absorption cross-section data for 3-methylbutanal as reported by Lanza et al. (2008), since no data is available for 33DMbutanal. Therefore, we assume that the photolysis rate constant for 33DMbutanal may have a significant uncertainty.**

**In contrast, for 3,3-dimethylbutanone (33DMbutanone), the photolysis rate constant is derived from the data provided by Mapelli et al. (2023), who determined quantum yields and effective absorption cross-sections. Consequently, this value does not carry significant uncertainty**

Typos / minor concerns:

should be an italic "k" throughout the manuscript;

**Ok, all "k" will be changed an italic "*k*" throughout the manuscript;**

Table 1 header is confusing. It states that k is in units of cm3 molecule-1 s-1, but in fact the k values throughout the table are in different units of 10-10, 10-11 or 10-12 cm3 molecule-1 s-1, depending on which reaction is being reported. I suggest that the most sensible way to report this is to list all k values in one consistent set of units, e.g. 10-11 cm3 molecule-1 s-1. At the very least, remove the misleading statement "k in units of cm3 molecule-1 s -1" from the table header.

**Ok, the suggested changes will be taken into account in a revised manuscript**

Similar comments re. k values and powers of ten for Table 2.

**Ok, the suggested changes will be taken into account in a revised manuscript**

Line 166 "an" to "and"

**Ok, the suggested changes will be taken into account in a revised manuscript**

Referee #3

General Comments:

The manuscript investigates the oxidation processes of two VOCs, specifically 3,3-dimethylbutanal and 3,3-dimethylbutanone, by Cl (with and without NO), NO3, and OH radicals. The authors provide extensive and detailed information on the chemical kinetics, reaction pathways, and resulting products. This study makes a significant contribution to our understanding of the oxidation mechanisms of 3,3-dimethylbutanal and 3,3-dimethylbutanone, particularly those involving Clorine atoms. However, I have some concerns regarding the treatment of wall loss. Once this issue is addressed and improve the quality of certain figures, the manuscript will be ready for publication in ACP.

Regarding the wall loss description (Line 149), while the results indicate minimal losses for 3,3-dimethylbutanal (3%) and 3,3-dimethylbutanone (0%), it is essential to account for potential wall loss impacts for oxidants like chlorine, N2O5, and NO3 on the rate constant results. Heterogeneous reactions, such as wall loss, may introduce uncertainties in chamber studies. Previous chamber studies have reported wall loss rates for these Cl, N2O5 , and NO3 (e.g., https://doi.org/10.5194/acp-2020-360 and https://doi.org/10.1016/j.cplett.2009.03.047 ). The authors should provide more detailed data on wall loss rates to reinforce the reliability of their findings.

**Response of authors.**

**In our case, a relative method is used, unlike the study described in https://doi.org/10.1016/j.cplett.2009.03.047, which employs an absolute method.**

**We conducted experiments to determine the wall losses of $CH_3ONO$ and $N_2O_5$ in the reactor (heterogeneous process), finding that this loss is significant for $N_2O_5$. In the case of $Cl_2$ and atomic Cl, we did not measure their losses because we lack a detection system for $Cl_2$ or atomic Cl. However, in the relative methodology, if the loss of the oxidant or precursors is significant, it would result in a reduced concentration of the oxidant available to react with the reactants (reference compound and carbonyl). This affects the reaction rates of both equally but not the rate constant itself. The loss constant due to heterogeneous reactions of the oxidants is not included in the kinetic equation derived from the relative method.**

**On the other hand, for the kinetic study of 3,3-dimethylbutanone with OH and Cl, as well as 3,3-dimethylbutanal with Cl, the prior studies demonstrated that over 45 minutes the loss of 33DMbutanal is less than 3% and 0% 33DMbutanone. Therefore, the losses of 33DMbutanone and 33DMbutanal are considered negligible. This is extensively explained in the responses to referees #1 and #2.**

Some figures need quality improvement. For instance, Figure 2 requires a legend, and Figure 6 shows low resolution with small, hard-to-read text; it may be better suited for the supplementary document. Additionally, Figure 8 is difficult to read and should be rearranged to enhance readability.

**Response of authors.**

In Figure 2, there is indeed a legend, although it is so small that it is difficult to see clearly. Due to the large number of spectra in this figure, it has been difficult to improve the quality while maintaining the full spectrum (from 800 cm$^{-1}$ to 3000 cm$^{-1}$).

For Figure 6, it is also difficult to obtain a higher-resolution image as these are directly obtained from the GCMS software. The authors deemed it appropriate to include some of the chromatograms in the manuscript so that readers can verify the formation of the reaction products. We will attempt to improve the resolution; however, as the referee suggests, this figure could be moved to the supplementary information, where the spectra can be enlarged and visualized more clearly.

The Figure 8, will be rearranged to enhance readability in a revised manuscript

Specific Comments:

Line 259: Does "x3" in Channel II refer to three attack sites?

There is no clear explanation of what "x3" means in this context. The same issue arises in Line 264 for Channel III.

It refers to the fact that the radical attack on the -CH$_3$ group of the tert-butyl group should be multiplied by 3. Since all three positions are equivalent, only one instance is included, and therefore, the products formed through this pathway are the same. A sentence will be added to the manuscript to clarify what "x3" indicates:`x3' indicates that there are 3 equivalent attack positions.`

Footnote Description table1: The footnote description is unclear. It is not specified whether "a" uses 10^-10, "b" uses 10^-11, and "c" uses 10^-12, or what KR represents. Please revise this for clarity.

Ok, the suggested changes will be taken into account in a revised manuscript

Figure 8: In Channel III, (49% X=Cl ); 2% X= OH....NO3), there should be  no ")" inside

Ok, the suggested changes will be taken into account in a revised manuscript

Referee #4

**Review on the paper egusphere-2024-3241 by Aranda et al.**

This paper deals with the kinetics and mechanisms of two carbonyl compounds (3,3-dimethylbutanone and 3,3-dimethylbutanal) with OH, Cl and NO3. The kinetics of 33DMbutanal with Cl atoms at ambient temperature are determined for the first time, while for the other kinetics (33DMbutanone with Cl atoms and OH radicals), values were already reported in the literature. A set of reaction products has been identified, with a (semi)quantification for a few of them.

My main concern is that many statements consist of suggestions that are not (fully) supported by results. Some of the conclusive remarks are not really convincing. The kinetics present scattering in the results that needs to be clarified.

Overall, the manuscript is quite long and there are many typos that need to be corrected (I reported only a few below).

**Comments**

**The authors thank you for dedicating your time to reviewing our work. We greatly value to reviewing our work, and we consider it very valuable to have the opportunity to discuss the results. As we can confirm, the review of the manuscript has been very thorough, allowing us to correct not only typographical errors but also some data that had not been accurately transferred from the spreadsheets to the tables in the manuscript.**

**Below, we present our point-by-point response to your comments.**

**NOTE: It is important to note that, based on the suggestion of Referee #5, the authors have decided to restructure the entire Results and Discussion section regarding the reaction products. This involves changing the figures in both the manuscript and the supplementary material.**

Line 34: it is stated that "large carbonyl compounds can influence on climate change if they are strong infrared light absorbers by altering the Earth's radiation balance": I don't agree: the atmospheric concentrations of carbonyls are too small to influence the climate by IR light absorption.

**Response of authors.**

**The authors agree with your comment, and the sentence has been revised in the manuscript as follows:**

**Lines 35-38**

` In addition, large carbonyl compounds could influence on climate change altering the Earth's radiation balance if they are strong infrared light absorbers and their atmospheric concentrations are sufficiently high. Additionally, carbonyls could be an important source of aerosol which could further affect radiation balance and be hazardous for health (Liu et al., 2022; Heald and Kroll, 2020).'

3.1 Kinetic study: did you check that the reaction of the compound (resp. the reference) with the oxidant did not form products absorbing at the selected IR bands for the reference (resp. for the compound)?

**Response of authors.**

**Among the preliminary tests to be conducted before performing the kinetic experiments, verifying that there is no interference from the IR bands of the reaction products (both from the reaction of the oxidant with the reference compound and with the compound under study) with the IR bands selected to monitor the reaction must be do. However, based on the experience acquired over the past 25 years in kinetic studies using FTIR, a methodology has been adopted for analyzing IR spectra that minimizes the effect of products absorbing at the selected IR. Specifically, different IR bands are analyzed to determine which of them show decreasing subtraction factors as the reaction progresses, and the reaction is avoided at large % of conversion. If it is observed that the subtraction factors do not decrease but instead increase, the reference compound is then replaced. In the case of Cl reactions, the kinetic experiments were carried out with $N_2$ gas (gas carried), in order to minimize the influence of reaction products.**

Why not showing all the relative kinetic plots (in Supplementary Material)?

**Response of authors.**

**The authors think it would involve providing too much information. However, the relative kinetic plots are included in in supplementary material in the revised version.**

Schemes S1 & S2: HO2 radicals are formed in the mechanisms (through RO+O2 for example): why not including the RO2+HO2 reactions in the mechanisms (see also line 606)? Especially for secondary and even more tertiary RO2, where RO2+RO2 rate constants are very low… Besides, these mechanisms could be advantageously simplified, by removing low reactivity channels (for example RC(O)O radicals mainly decompose into R+CO2, (CH3)3CO2+(CH3)3CO2 is negligible, etc.). What about direct photolysis of a-dicarbonyls, which has been shown to be significant?

**Response of authors.**

**Initially, in the mechanisms proposed in Schemes 1S and 2S, the RO2 + HO2 reaction was not considered to avoid overly complicating the reaction mechanisms, as the RO2 + HO2 reaction can lead to different pathways depending on the structure of the RO2 radical. Moreover, these reactions would only be significant in the absence of NO/NO2. However, following your suggestion, Scheme 1S has been simplified by removing the low-reactivity channels, and the main reactions for the RO2 + HO2 pathway according to Jenkin et al. (2019), have been included as explained in the revised manuscript. Conversely, in Scheme 2S, these reactions have not been included to avoid further complicating the proposed mechanism. A statement in lines 309-311 has been added to the revised manuscript to indicate that RO2 + HO2 reactions should also be considered in unpolluted environments (in the absence of NO/NO2).**

**Lines 309-311**

'The RO2 + HO2 reactions are excluded to avoid further complicating the mechanism for 33DMbutanal, as they are significant only in the absence of NOx, that is, in remote unpolluted atmospheres.'

Additionally, direct photolysis is not included in our mechanism, as it is exclusively focused on the reactions with the three atmospheric oxidants studied in this work. Under the conditions in which the product study was conducted, neither 33DMbutanone nor 33DMbutanal undergo photolysis. Therefore, the products identified in the qualitative analysis do not originate from these reactions.

Table 1: there are significant discrepancies between the k values obtained using the different reference compounds, especially with 33DMbutanone+OH (from 0.96 to 1.92) and with 33DMbutanone+Cl (from 3.03 to 5.30). This need to be discussed. How well known are the reference rate constants?

**Response of authors.**

**Upon reviewing the data in Table 1, we have identified an error in the reported rate coefficient for the reaction of 33DMbutanone with OH when 1-butanol was used as the reference compound. Indicate that in the manuscript revised and following the suggestion of Referee #5, the reference rate coefficient value for 1-butanol has been changed to the one recommended by McGillen et al. (2020) using $k_{1-butanol} = 9.14 \times 10^{-12}$ cm$^3$ molecule s$^{-1}$.**

**When the FTIR system is used to determine the rate coefficient, a systematic error associated with the subtraction process arises, which cannot be precisely quantified. To minimize this systematic error, it is important to avoid overlapping of IR bands from the radical precursor, reference compounds and reaction products. In many cases, it is challenging to find a reference compound whose characteristic IR bands do not overlap with those of the organic compound (in our case, 3,3-dimethylbutanal and 3,3-dimethylbutanone). Additionally, the slower the reactions, the smaller the variations in the absorbance of the characteristic IR bands of the reacting compounds, making the subtraction factors determined at time t and t' very similar. In such cases, small changes in determining the subtraction factor can significantly influence the calculation of the relative rate coefficient ($k_{carbonyl}/k_{Rererence}$) and therefore, the calculation of $k_{carbonyl}$. For this reason, four reference compounds were used in the reactions of 33DMbutanone to determine the rate coefficient ($k_{carbonyl}$) as accurately as possible.**

**The rate coefficients obtained in this work (as weighted average) are very similar to those reported in the literature, therefore, the authors think that our results provide a reliable rate coefficient for the reaction of 33DMbutanone with Cl and OH.**

**The reference rate coefficients of all reference compounds, are the corresponding to the database of McGillen et al 2020 (https://essd.copernicus.org/articles/12/1203/2020/ ) as it is indicated in the manuscript.**

Line 321: Figure 2: what do you mean with "64 % of the conversion and a 29% respectively"? What are the experimental conditions of the experiments? Could you indicate the concentrations of the references? This figure is difficult to read (small size, bright colours, IR spectra mixed).

**Response of authors.**

Indicate that, following the suggestion of Referee #5, the authors have restructured the entire *Results and Discussion section* related to the reaction products. The presentation is now organized by compounds, which has led to some modifications in the original figures, including Figure 2. In the new version of the manuscript the Figure 2 shows the residual FTIR spectra for the reactions of 33DMbutanone with Cl in the absence and presence of NO at different time reactions.

The experimental conditions are almost the same in all reaction product experiments, so the concentration ranges of the reactants are indicated in lines 143-148 of the new version of the manuscript. The reference IR spectrum of HCHO and PAN, have been obtained from Eurochamp 2020 database (https://data. eurochamp.org/data-access/spectra/) and from a commercial compound sample for the case of acetone. In the Figure 2, only a qualitative comparison is made to show the presence of these compounds as reaction products.

For the quantization of reaction products, the concentration of reference IR spectrum corresponds to 0.18 ppm for HCHO (Rodenas et al 2017), 11 ppm for Acetone (commercial compound); 2.30 ppm for 22DMpropanal (Eurochamp 2020 database) and 0.018 ppm for PAN (Eurochamp 2020 database).

Due to the large number of spectra in this Figure 2, it has been difficult to improve the quality while maintaining the full spectrum range (from 800 cm$^{-1}$ to 4000 cm$^{-1}$).

Line 333: it is stated that "Additionally, IR bands of N2O5 (precursor of NO3) have been observed in the reaction with Cl atoms": referring to Figure S2, we cannot see any N2O5 bands appearing in the product spectra…

**Response of authors.**

As I mentioned above, the manuscript has been restructured, and in the new version, Figure 2S has been modified. The N$_2$O$_5$ spectrum is now shown in Figure 12S

In the initial Figure 2S, the IR band of N$_2$O$_5$ most clearly observed is around 1250 cm$^{-1}$, while the other characteristic band of N$_2$O$_5$ (744 cm$^{-1}$) is not as clearly visible because it overlaps with other IR bands from the reactants or reaction products.

[Figure]

An amplification of this band is showed below,

[Figure]

Line 365: "the concentrations of acetone and nitrated compounds increase more than expected": what are actually the expected trends??

**Response of authors.**

**The initial figure 4b is Figure 14Sb in the revised manuscript.**

**According to the concentration-time profile for acetone and the nitrated compounds, it seems that after 5 minutes (nitrated compounds) or 25 minutes (acetone), the trend changes, showing a slight increase in concentration beyond what is expected. If these products originate from the main reaction of 33DMbutanal with Cl, the profile should exhibit the same trend throughout the entire period. This is what we mean by expected. However, if a shift occurs after 5 and 25 minutes, respectively, we interpret this as a contribution from the reaction of one or more compounds other than 33DMbutanal (reaction products).**

[Figure]

In the revised manuscript, the sentence in lines 530-540 has been added as follows:

'The trends of the concentration-time profiles of the quantified products for the reaction of 33DMbutanal with Cl atoms (Fig. 14S), suggest that they are primary products in the early stages (linear trend). However, it is observed that after a certain reaction time, the concentrations of 22DMpropanal and HCHO decrease. This decrease could be attributed to loss processes involving reactions with Cl atoms and/or photolysis. Additionally, the concentrations of acetone and nitrated compounds are observed to increase more than expected (profile with an upward trend, Fig. 14Sb), likely due to contributions from secondary chemistry, such as the previously discussed reactions, of 22DMpropanal with Cl or its photolysis. For the nitrated compounds, a change in the trend is observed around 2 minutes likely due to the formation of nitrated peroxycarbonyl compounds because of the presence of large amounts of $NO_2$ in the reaction mixture. The profile of nitrated compounds from the reaction of 33DMbutanal with $NO_3$ (Fig. 14Sd), shows an increase from the start of the reaction, attributed to the presence of $NO_2$ in the reaction medium from the beginning.'

Figure 3S: what are the differences with Figure 4 (a) and (b)??

**Response of authors.**

**The difference is that Figure 4 shows the products generated in the reaction with Cl and NO, while Figure 3S shows the products generated in the absence of NO**

**Now in the new version of the manuscript the figures have been changed, as mentioned above.**

Figure 5S: OH reaction in absence of NO: this is from H2O2 photolysis, right? Did you carry out

OH reaction in presence of NO (as seems from line 118)? Is there a yield plot also?
**Response of authors.**

**Figure 5S ((Figure 6S (b) in the new version of manuscript) corresponds to the reaction of 33DMbutanone with OH in the absence of NO, using H₂O₂ as the OH precursor. These experiments have been conducted in a different experimental system specifically to study this kinetics in the absence of NO. Therefore, the experiments conducted with H₂O₂ as the OH precursor, have not been performed in the presence of NO. For the experiments involving OH in the presence of NO, CH₃ONO was used as the OH precursor, as indicated in the manuscript. These represent two distinct experimental setups: one uses a gas cell with a quartz body that allows photolysis of H₂O₂ at λ = 254 nm, while the other uses a glass gas cell that permits photolysis only at λ = 360 nm.**

**The quartz cell system belongs to another research group, and only a limited number of experiments were conducted with it, as it was not operational full-time. Therefore, the yields in the presence of NO correspond to the experiments performed with the glass cell, while the yields in the absence of NO correspond to the experiments conducted with the quartz cell. It was not possible to perform experiments using H₂O₂ as a precursor in the presence of NO.**

Figure 5: comparing (a) and (b): why are the plots of nitrated compounds curved in one case (a) and linear in the other case (b)?
**Response of authors.**

**The figure 5 has been changed in the new version of manuscript. Now the plots of 33DMbutanone reactions appear separately to the 33DMbutanal.**

**In the initial manuscript, the curved plot in the Figure 5 (a) (33DMbutanone reaction, now Figure 4) is indicating that there are additional reactions that form nitrated compound, and in this case, to minimize error in the calculations, the yield of nitrated compounds, have been obtained from linear trend of first points.**

**In the revised manuscript, the next sentence has been added as follows:**

**Lines 378-379**

'The yields of nitrated compounds were estimated from the slopes of the plots showing linear behavior (using the initial data) to avoid contributions from secondary chemistry.'

Figure 7S: how do you explain the shift on the x-axis?

**Response of authors.**

**The figure 7S has been changed in the new version of manuscript. Now is figure 7Sb.**

**This may be due to the photolysis of HCHO by radiation at λ = 254 nm (radiation used to generate OH radicals from H₂O₂). This phenomenon was observed consistently across the three experiments conducted. However, it does not occur in the case of acetone.**

Table 3: how did you calculate the uncertainties for the average yields? Why so large differences with those in Table 4??

**Response of authors.**

**When reviewing the data in Tables 3 and 4 and comparing them with the data in the Excel**

sheets containing all calculations, it was observed that the same criterion had not been applied in all cases. In some cases, the uncertainties in the individual yields corresponded directly to the error of the slope, while in other cases, the error of the slope was multiplied by 2. A similar issue was found in the average yield values: in some cases, the uncertainty corresponded to the standard deviation, while in others, it was the standard deviation multiplied by 2. This discrepancy explains why the uncertainties in Table 3 are larger than those in Table 4.

It was decided to apply a consistent criterion in all cases: for individual yields, the uncertainty was taken as the error of the slope multiplied by 2, and for the average values, as the standard deviation multiplied by 2. In the revised version of the manuscript, these errors have been corrected.

Additionally, upon reviewing all the values, it was detected that some data were incorrect, i.e., the values in the Excel sheets did not match those in the manuscript. These corrections have been highlighted in red in the revised version of the manuscript. We apologize for the error.

"reaction of 33DMbutanone with Cl atoms in the presence of NO the total carbon is 95% (not accounting for nitrated compounds),": how is it possible to get almost 100% carbon balance without taking into account nitrated compounds (that seem to be significant from Table 3)?

**Response of authors.**

As indicated in the manuscript, we consider that the yields should be interpreted with caution. This is because, due to the overlap of the absorption bands of all the compounds present in the reaction mixture, the subtraction process is highly complex and prone to significant errors, particularly in experiments conducted in the presence of NO where different nitrated compounds are generated. Furthermore, in the case of the reaction of 33DMbutanone, the variations in the concentrations of the formed products are very small, making it even more challenging to accurately determine the subtraction factor.

The fact that the total carbon percentage is so high without accounting for nitrated compounds, combined with the observation of other products in the GC-MS experiments (which were also not included in the total carbon calculations), leads us to believe that in the reaction of 33DMbutanone with Cl in the presence of NO, the yields of acetone and formaldehyde are likely overestimated.

In the revised manuscript, the next sentence has been added as follows:

Lines 406-412

'In the reaction of 33DMbutanone with Cl atoms in the presence of NO the total carbon is 95% (without nitrated compounds yields). This high yield suggests possible overestimation of acetone and/or HCHO. In this regard, it should be noted that the calculated yields for acetone in reactions where nitrated compounds are generated have significant errors due to overlapping bands. On the other hand, as previously mentioned, the yields of the nitrated compound have been estimated using the average integrated absorption of similar compounds. Therefore, the yields of these nitrated compounds should also be interpreted with caution. ´

Table 4: for "33DMbutanal + NO3": no yields are reported: could the authors calculate upper limits (based on their sensitivity)? Especially since acetone is detected by GC… Why is the yield

of nitrated compounds exactly = 100%??

**Response of authors.**

**As in the experiments with NO, the presence of large amounts of NO₂ in the reaction medium and the formation of nitrated compounds make it very challenging to identify the IR absorption bands of acetone or 2,2-dimethylpropanal and, consequently, to quantify them.**

**The yield of nitrated compound is not exactly 100%. Now in the revised manuscript, the 4 individual yields and the average yield of nitrated compounds have been included in the table 4.**

**These values were obtained using a procedure that employs an absorption cross-section within the 1230–1200 cm⁻¹ range. In the case of reactions with NO₃, the formation of numerous other nitrated compounds could be favored, which may differ from the specific organic nitrates that exhibit absorption within the selected range. This could lead to an overestimation of the reported yield.**

Tables 3 & 4: calculation of the carbon budget: I suggest explaining it in the main text, not in the notes of the Tables, since it is not very clear as now…

**Response of authors.**

**As suggested, in the revised manuscript, the calculation of the carbon budget has been included as main text in lines 391-392 as follows**

The yield of total carbon in % is calculated with Eq (II)

$$Total\ carbon(\%) = \sum_{1}^{i} \frac{n^\circ\ of\ carbon\ of\ product_i}{n^\circ\ of\ carbon\ of\ carbonyl\ compound} \times molar\ yield_i \qquad (II)$$

Lines 424-426: from Figure S9, it is said that formic acid and hydroxyacetone were observed. While this seems to be valid of formic acid, I cannot see any features for hydroxyacetone… Besides, formic acid is often observed in such experiments when formaldehyde is present; it may be formed from secondary wall reactions of CH2O…

**Response of authors.**

**It is true that hydroxyacetone is not clearly observed (band at 1288 cm⁻¹), possibly because the amount generated in the reaction is very small and its spectrum is therefore obscured by the bands of other compounds.**

**We agree that formic acid is primarily produced from sources other than the reaction of 33DMbutanone (as demonstrated by the HCOOH concentration-time profile, that is included in the revised manuscript as supplementary material, which curves upward). For greater clarity, a sentence has been included in the revised manuscript as follows:**

**Lines 383-385**

'Formic acid is mainly produced through secondary reactions when formaldehyde is present, as evidenced by typical secondary concentration profile time (see Fig. 9S).'

Figure 12S: "generated chromatograms for the 33DMbutanone + OH +NO at different reaction times": what are these reaction times? It is noted "8 min" for all last 3 chromatograms… What do you mean with "generated"?

**Response of authors.**

**In figure 12S (Figure 10S in the revised manuscript). The reaction times are as follows: t = 0 minutes, t = 60 minutes, t = 130 minutes, and t = 260 minutes, listed from top to bottom. The time has been indicated in the figure for clarity. The *'8 minutes'* refers to the duration of the contact between the SPME and the reaction mixture (sampling time) and not the reaction time. This time is presented in all chromatograms. It has not been considered necessary to add any additional clarification regarding this in the article.**

**The term 'generated' indicates that the chromatograms were created using the software tool provided with the equipment. As previously mentioned, this tool enables the generation of cleaner chromatograms by reducing baseline noise. It is used exclusively for qualitative purposes.**

Line 460: "a little peak that appear together with the peak of air, that could correspond to acetone.": why not calibrating directly with pure acetone instead of "creating a chromatogram using a specific tool"??

**Response of authors.**

**When a pure acetone sample is introduced, it is observed that acetone elutes along with the air peak. Generating a chromatogram using the specific software tool provided with the equipment allows for better visualization of the peak, as this tool produces a chromatogram with a baseline that has less noise, facilitating easier qualitative analysis.**

Lines 462-467: this descriptive part concerning the "specific tool" for creating chromatograms is absolutely unclear. Which software? What about reliability? What is the advantage for acetone compared to direct calibration?

**Response of authors.**

**As indicated, it is a tool of GC-MS software (Mass Center main). It is used exclusively to assist with qualitative analysis. In the revised version of the manuscript, the text has been changed to initial of the section 3.2 and modified for clarity as follows :**

**Lines 321-328**

'A specific software tool of the mass spectrometer was used to generate chromatograms, from the original chromatograms, with a lower signal to noise ratio to improve peaks visualization and thereby facilitating the analysis of the experiments performed using this technique. For that, in the software it is specified the desired m/z, or m/z range, and then the chromatogram is created displaying the ion intensity versus time, with peaks representing the compounds that correspond to the specified m/z (or m/z range). In Fig. 2S a comparation of an original chromatogram and a chromatogram generated with this tool are shown. This tool has been used for all SPME/GC-TOFMS chromatograms obtained in EI mode. ´

Lines 492-493: "the possible reaction of $RO_2$ with OH radical": could you estimate the contribution of these reactions relative to RO2 + RO2?

**Response of authors.**

**To estimate the contribution of $RO_2$ with OH radical reaction relative to RO2 + RO2 it is**

**necessary to quantify the products formed by the reactions of RO₂ with OH radical.**

Line 508: PAN is known to be unstable at high temperature: how can you attribute the peak at 5.59 min to PAN, knowing that the SPME fiber is heated at 250°C in the GC injector? See also lines 569- 570.

**Response of authors.**

**Indeed, PAN is unstable and decomposes in the injection port into CH₃-C(O)OO and NO₂. Considering that PAN is clearly observed in the FTIR spectrum and that the mass spectrum corresponding to the chromatographic peak at a retention time of $t_r$ = 5.59 min does not account for the formation of any other compound proposed in Scheme 1S, it was concluded that this peak likely corresponds to PAN.**

**Upon decomposition, the CH₃-C(O)OO radical can fragment into CH₃-C(O)⁺ (m/z = 43) and CH₃-C(O)O⁺ (m/z = 60), fragments that can be seen in the mass spectra.**

**In the revised manuscript, the next sentence has been included as follow:**

**Lines 460-468**

`The peak at 5.59 minutes is identified as peroxyacetyl nitrate (PAN) which is unstable and decomposes in the injection port of the gas chromatograph. Upon decomposition, the CH₃-C(O)OO radical could fragment generating the ions CH₃-C(O)⁺ (m/z = 43) and CH₃-C(O)O⁺ (m/z = 59). These fragments are like those observed in the EI mass spectrum (see Table 2S). There is not any reference mass spectrum to compare, only two studies determined the chemical ionization mass spectrum of PAN (Phillips et al., 2013; Pate et. al., 1976). Moreover, considering that PAN is clearly detected in the experiments conducted with the FTIR system and that the mass spectrum corresponding to the chromatographic peak with a retention time of $t_r$ = 5.59 min does not account for the formation of any other compound proposed in Scheme 1S, it was concluded that this peak likely corresponds to PAN.'

lines 512-514: PAN is formed after a series of almost 12 successive reactions (scheme 1S): how can you affirm that a PAN yield of 100% is consistent with the SAR estimated branching ratio of 98% fO)or channel II??

**Response of authors.**

**Although it is generated after 12 successive reactions, the stoichiometry of the reaction would be 1:1. The plot of PAN concentration formed versus the variation in 33DMlbutanone concentration shows a linear relationship with a slope close to 1. Furthermore, the formation of PAN in the reaction of 33DMbutanone with Cl cannot be explained by any other pathway. In the revised manuscript, the next sentence has been included as follows:**

**Lines 485-490**

'Considering that the formation of PAN can only be explained via channel II (hydrogen abstraction from the -CH₃ group of the tert-butyl group), which involves the decomposition of the initially formed alkoxy radical (2,2-dimethyl-3-oxobutan-1-yloxy radical), and that the plot of PAN concentration versus the variation in 33DMbutanone concentration shows a linear relationship with a slope close to one (indicating an estimated PAN yield of 100%, see Fig. 5S), it could be concluded that the percentage of Cl attack on the -CH₃ group of the tert-butyl group is nearly 100%. This value closely with the 98% estimated by the SAR method for

reactions with Cl (see Table 1S).'

Line 543: isomerization of oxy-3,3-dimethylbutyryl radical: is this "new" pathway supported by literature studies? Because decomposition of such oxy acyl radicals (into alkyl + CO2) is considered to be extremely fast. I cannot see this suggested reaction on Figure 8, why? Indeed, another pathway was suggested farther (lines 602-607). Why then concluding (lines 696-698) that "The positive identification of 33DMbutanoic acid implies a pathway in the reaction mechanisms of 33DMbutanal, that initially have not been considered."?

**Response of authors.**

**We think that the explanation in the manuscript of this fact should be clarified and corrected. On one hand, the isomerization of the oxy-3,3-dimethylbutyryl radical was initially proposed to explain the formation of a product whose mass spectrum could correspond to the peak at 9.95 minutes.**

**Although this pathway is highly unlikely, initially we could not find another explanation for the formation of the peak at $t_r$ = 9.95 min observed in all reactions (with all oxidants). Subsequently, a commercial sample of 3,3-dimethylbutanoic acid was injected, confirming that its $t_r$ = 9.95 min and that its mass spectrum matches the one corresponding to the 9.95-minute peak.**

**We therefore consider that the only explanation for the formation of 3,3-dimethylbutanoic acid is the inclusion of a "new" reaction pathway not considered in any other bibliographic study (although there are studies that report the formation of an organic acid (Asensio et al. 2022, Bo et al. 2022).**

**In the new version of the manuscript, the paragraph has been rewritten for greater clarity as follows:.**

**Lines 557-571**

'It is interesting to note the presence of a common peak for all reactions that appears at 9.56 min (8.95 min in the experiment with a new chromatographic column). The mass spectrum of this peak is assigned to 3,3-dimethylbutanoic acid (33DMbutanoic acid) by the NIST database software, with a similarity index of approximately 87%. The formation of this compound cannot be explained by the general scheme proposed (Scheme 2S), which is based on principles of atmospheric reactivity and bibliographic studies of similar compounds (Aschmann et al., 2010; Atkinson, 2007). To explain the formation of this peak, other pathway was initially proposed, where the oxy-3,3-dimethylbutyryl radical (channel I), undergoes isomerization (see Scheme S4). This pathway leads to the formation of 4-oxo-3,3-dimethylbutanoic acid, whose mass spectrum shows a fragmentation pattern very similar to the peak with a tr of 9.56 min. Taking into account that 33DMbutanoic acid is a commercial compound, a sample was injected into the SPME/GC-TOFMS system. The chromatogram showed a peak at approximately 9 minutes, with a mass spectrum (see Fig. 21S) identical to that of the peaks (9.56/8.95 min, see Table 3S) which positively confirms the formation of 33DMbutanoic acid in the reactions of 33DMbutanal with Cl atoms and OH and NO3 radicals. The characteristic IR bands of 33DMbutanoic acid seem to be present in the FTIR residual spectra obtained for the reaction of 33DMbutanal with Cl atoms (Fig. 22S). Recent studies have also detected organic acids from reactions of saturated aldehydes (Asensio et al., 2022; Bo et al., 2022).'

**Lines 638-644**

'The formation of 33DMbutanoic acid in all reactions suggest another reaction pathway not considered in Scheme 2S, as RO2 + HO2 reactions are less likely in the presence of NO/NO2. A proposed reaction pathway explains the formation of the 33DMbutanoic acid:

$(CH_3)_3CCH_2C(O)OO\cdot + HO_2\cdot \rightarrow (CH_3)_3CCH_2C(O)OH + O_3$           (R9)

The IR bands of ozone are not observed in the residual spectra of Fig. 13S, likely due to the low amount of $O_3$ and overlapping IR bands. Similar reaction was proposed for $CH_3C(O)OO$ radical by other authors (Groß et al., 2014; Dillon and Crowley, 2008; Tomas et al., 2001).'

**In the revised manuscript, the reaction for the formation of 33-dimethylbutanoic acid has been included in Figure 8.**

Lines 545-549: mass spectrum of 33DMbutanoic acid compared to mass spectrum of the peak at 9.56 min: you indicated that these 2 spectra match well ("correspond"): could you support this by showing the comparison in Supp Mat?

**Response of authors.**

**According to your suggestion, these spectra are already included in the revised supplementary material. Figure 21S(a) shows the chromatogram with the retention time of the commercial sample of 3,3-dimethylbutanoic acid (9 min) and its mass spectrum (MS). Table 3S presents all the MS spectra for the chromatographic peaks at $t_r$ = 8.95/9.56 min (noting that these are two different retention times due to a change in the chromatographic column).**

**As is indicated above, for better clarity the paragraph has been modified in the revised manuscript as follows:**

**Lines 565-569**

'Taking into account that 33DMbutanoic acid is a commercial compound, a sample was injected into the SPME/GC-TOFMS system. The chromatogram showed a peak at approximately 9 minutes, with a mass spectrum (see Fig. 21S) identical to that of the peaks (9.56/8.95 min, see Table 3S) which positively confirms the formation of 33DMbutanoic acid in the reactions of 33DMbutanal with Cl atoms and OH and NO3 radicals.'

Lines 550-552: Figure 18S is not convincing…

**Response of authors.**

**As indicated in the previous point, mass spectrometry analyses confirm that 3,3-dimethylbutanoic acid is formed in the reaction of 33DMbutanal with Cl. Figure 18S (figure 22S in the revised manuscript) presents the residual IR spectrum of this reaction, along with the IR spectra of 3,3-dimethylbutanoic acid and 2,2-dimethylpropanoic acid. This figure aims to show that the residual spectrum exhibits bands in the same IR range as those of these acids. However, the characteristic IR bands of these acids are not clearly visible, because the amounts of these compounds formed are very small. Additionally, there may be overlaps with the bands of other compounds present in the reaction medium that have not been identified. Thus in the revised manuscript the sentence has been modified as follows:**

**Lines 569-571**

´The characteristic IR bands of 33DMbutanoic acid seem to be present in the FTIR residual spectra obtained for the reaction of 33DMbutanal with Cl atoms (Fig. 22S). Recent studies have also detected organic acids from reactions of saturated aldehydes (Asensio et al., 2022; Bo et al., 2022).'

Lines 558-561: mass spectrum of 22DMpropanoic acid compared to mass spectrum of the peak at
8.23 min: you indicated that these 2 spectra match well ("correspond"): could you support this by showing the comparison in Supp Mat?

**Response of authors.**

**The mass spectra for the peak at 8.23/8.26 min are presented in Table 3S. The initial figure 17S, has been updated to include the chromatograms and spectra of the commercial samples of 2,2-dimethylpropanoic (Figure 21S(b) in the revised manuscript).**

Lines 582-583: I don't understand how you conclude that acetone comes mainly from channel I
*because* the yields are similar… this may be fortuitous?

**Response of authors.**

**Indeed, the fact that the acetone yield is similar in the three reactions could be coincidental. On the other hand, in the study by Aschmann et al. (2010), the route I is proposed as the main reaction pathway for 33DMbutanal with OH radicals, leading to the formation of acetone and nitrated compounds, although acetone was not quantified in that study. Therefore, if we consider that, as proposed in the study by Aschmann et al. (2010), acetone originates from channel I, the similar acetone yields obtained for the reaction of 33DMbutanal with OH, Cl, and Cl + NO could indicate that acetone also originates from route I in the case of reactions with chlorine.**

**For better clarity the paragraph has been modified in the revised manuscript as follows:**

**Lines 615-620.**

'Acetone is a reaction product formed from all channels (see Scheme 2S). According to Aschmann et al. (2010) about reaction of 33DMbutanone with the OH radical the percentage of channel I is 94%, with acetone, tert-butyl nitrite and tert-butyl nitrate as the main reaction products. Therefore, considering the results of Aschmann et al. (2010), the acetone quantifies in our work for the reaction of 33DMbutanone with OH (32%) would be formed mainly through channel I. In the reaction with Cl without NO, the acetone yield (28%) is close to the percentage predicted by the SAR method for channel I (30%)'.

Line 596: "100% of the reacted 33DMbutanal forms nitrated compounds": this is not demonstrated; this is only suggested because no other compounds were detected.

**Response of authors.**

**Upon reviewing the manuscript, we have detected that Figure 5, referenced in line 595 of the initial manuscript, does not show the formation of nitrated compounds from the reaction of 33DMbutanal with NO₃, but rather the nitrated compounds generated in the reactions of 33DMbutanone and 33DMbutanal with chlorine atoms in the presence of NO. In the revised manuscript a new figure (Figure 15S) will be included in the supplementary material, showing the linear trend with a yield of ~100%.**

**For better clarity the paragraph has been modified in the revised manuscript as follows:**

**Lines 629-632**

'For the $NO_3$ radical reaction, the linear trend observed in Fig. 15S (d) indicates that 100% of the reacted 33DMbutanal forms nitrated compounds. If only peroxy-3,3-dimethylbutyryl nitrate was generated, the total carbon would be 100%, indicating that no other compounds were formed in the reactions with $NO_3$. However, as in the case of reactions with 33DMbutanone, the yield of nitrated compounds should be considered with caution.'

Lines 636-638: photolysis rates: it should be specified if the photolysis quantum yield has been taken as 1.

**Response of authors.**

**The authors have used the photolysis rate constant reported by Lanza et al. (2008) for 3-methylbutanal, reference included in the manuscript, considering it to be similar to that of 3,3-dimethylbutanal. Then, we do not consider it is necessary to specify the quantum yield.**

Lines 692-693: the conclusion that "the results suggest that the RO2 + OH· reaction in the unpolluted atmosphere could be significant." is bold, because it is only based on the difference of acetone yields between Cl and OH systems…

**Response of authors.**

**Indeed, this statement is based on the difference in acetone yields between the reactions of 33DMbutanone with Cl and OH, both in the absence of NO. We have revised the sentence and modified as follows:**

**Lines 725-726**

'The differing acetone yields from 33DMbutanone with Cl atoms and OH radical suggest the importance of the $RO_2$ + OH reaction in unpolluted atmospheres.'

**Minor comments/Typos**

Line 30: established.

OK

Line 36: remove ';' at the end of the sentence, before the closing bracket

OK

Citations of references are not homogeneous through the MS; please check ("XX et al., XXX" with point and comma).

OK

Lines 129-130: remove capital letters for the compound's names (same thing lines

136-139)

OK

Line 150: "the losses of the reactants…"

OK

Line 151: the loss of < 3% is over how much time?

**Response of authors**
**In the revised manuscript the time has been included as follows:**
**Lines 165-167**
'To determine these loss processes, the initial spectrum of the reactants was compared with the spectrum recorded after a long period (45 minutes, which is the typical time range for the kinetic study).'

Table 1: note d is not explained. Notes a,b,c: the statement "kcarbonyl and kR is given in $10^{-10}$, $10^{-11}$, $10^{-12}$, respectively" is not clear: we have 2 subjects (kcarbonyl and kR) and 3 possibilities ($10^{-10}$, $10^{-11}$, $10^{-12}$)!

**Response of authors**

**Based on the referees' suggestions, Table 1 has been modified for greater clarity, and errors in some data have been corrected. See the revised version of the manuscript.**

Lines 181 & 182: use "×" instead of "x" (check all the MS)

OK

Line 182: the statement "This is the general trend observed in the atmospheric chemistry for the oxidation reactions of organic compounds; kCl>kOH>>kNO3" needs at least one reference.

**Response of authors.**

**Some references have been included:**

**Line 204**

(Mellouki et al., 2015; Calvert et al., 2011; Atkinson, 2007

Consistency between Table 1 and 2: the uncertainties in kCl for 33DMbutanal (this work) are not the same. Same for kOH for 33DMbutanone.

**Response authors**

**In Table 2, the error for 33DMbutanal has been corrected according to the data in Table 1. In the case of kOH for 33DMbutanone, the value of 0.1 has been retained to ensure consistency with the number of significant figures presented for the rate coefficients in the literature.**

Line 210: please homogenize the naming of 3-methylbutan-2-one all through the MS (see Table 2 where it is noted 3-methylbutanone).

OK

Line 217: what is the "atmospheric

radical"?

**Atmospheric radical, have been changed by** 'atmospheric oxidant'

Line 222: "more favoured than H

atom

OK

Line 224: "33DMbutanone"

OK

Line 232: "double

bond"

OK

Line 238:

"coefficients"

Line 241: "kestimated=22.12x10-12": please keep only 3 significant figures and write in scientific style

**Response of authors**
**The data of "kestimated=22.12x10-12": and "kestimated=2x10-14": have been changed by "kestimated=2.21x10-11": and "kestimated=2.01x10-14, in order to keep 3 significant figures .**

Line 245: "factors"

OK

Line 254: "Schemes"

OK

Line 265: "major"

Line 271-272: the sentence is not clear

And for OH reactions the main initial attack of the radical would take place in the -CHO (~ 94%) followed by -CH$_2$- (~ 4%) and lastly the -CH$_3$ (~2%). Note that the percentage of -CH$_3$ corresponds to three times the % of channel III (see Table 1S).

**Response of authors**

**In the revised manuscript the sentence has been modified as follows:**

**Lines 284-286**

'For OH radicals, the principal channel is hydrogen abstraction from the -CHO group (~94%, channel I), followed by the -CH2- group (~4%, channel II), and -CH3 group (~2%, channel III).'.

Line 276: "neutral compounds": you mean "closed-shell"?

**Response of authors**

**By neutral molecules, we mean that they are not radicals. To avoid confusion, in the revised manuscript, 'neutral compounds' has been replaced by 'two molecules.**

Line 286: "establish"

OK

Line 303: "Colmenar et al.,

2018"

OK

Figures 3S & 5S, caption:

"reacted"

OK

Lines 391-396: Table 3 caption is reported twice.
OK

Table 3: GS-TOFMS, Reaction products: "Hydroxy-2,3-butanedione"

OK

Notes in Table 3: why using note (1) and * instead of continuing with "h"? Same comment for Table 4.

OK

Table 4: "33DMbutanal + OH":

"24.9±0.8"

Ok

Line 409: "products identified"

OK

Line 419: "only"
OK

Line 429: "GC-MSTOF": sometimes you write "GC-TOFMS": please be consistent!

OK

Line 431: "The yields should be considered with caution": but no yields are reported

for GC!!

**Response of authors**

**In the revised manuscript this sentence has been eliminated**

Lines 438-439: already said in lines 434-437.

**Response of authors**

**Actually, lines 438-439 and 434-437 (initial version of manuscript) do not mention exactly the same thing. Lines 434-437 state that the spectra were obtained in EI mode, while lines 438-439 specify that for 33DMbutanone, the spectra were obtained in FI mode. However, in the revised version of the manuscript, the paragraph has been modified as follows:**

**Lines 319-320**
`SPME/GC-TOFMS with EI and/or FI ionization was used as a qualitative technique to complement FTIR in identifying reaction products.

Figure 6: (a) and (b) look almost exactly the same while they are reported to be "33DMbutanone + Cl atoms (a) in the absence of NO and FI mode, (b) in the presence of NO"? How can you say that "(a) present more peaks than … (b)" (line 446)?

**Response of authors**

**Effectively, Figure 6 (a) and (b) are the same, meaning it is duplicated. A new figure (Figure 5 in the revised manuscript) has been created and included in the revised version**.

[Figure]

Figure 5. Example of the SPME/GC-TOFMS chromatograms for the reaction of 33DMbutanone + Cl atoms in the absence of NO and FI mode (reaction time t=0 min (a), t=20 min (b)) in the presence of NO and FI mode (reaction time t=0 min (c), t=20 min (d)) and in the presence of NO and EI mode (reaction time t=0 min (e), t=19 min (f)). Chromatograms (g) and (h) correspond to the reaction of 33DMbutanone + OH + NO at t=0 and t= 130 min of reaction respectively.

Line 447: "This indicates that the presence of NO influences the reaction mechanism.": this is true but not very new…

**Response of authors**

**Indeed, it is not new; we simply want to highlight what could be expected.**

Line 456: "In some cases": which cases?

**Response of authors**

**98% Acetone, 87.2 % 3,3-dimethylbutanoic acid, 73.5% 2,2-dimethylpropanol.**

**The percentage of similitude has been included in the revised manuscript in several sentences:**

**Lines 437-438**

'Except for acetone (98% of similarity index), the similarity index for the assignment of the reaction products was below 15%.'

**Lines 558-560**

'The mass spectrum of this peak is assigned to 3,3-dimethylbutanoic acid (33DMbutanoic acid) by the NIST database software, with a similarity index of approximately 87%.'

**Lines 576-577**

` The similitude index for most reaction products is low except for acetone (98%) and of peak at 5.47 min in the $NO_3$ reaction, assigned to 2,2-dimethylpropanol (22DMpropanol) with a similarity index of 73.5%.'

**Lines 592-593**

'A similar cyclization could explain the formation of 3,4-dimethyldihydrofuran-2,3-dione (34DMDHfuran23dione, tr = 16.2 min) observed in the reaction with Cl atoms and assigned by the NIST database with a similarity index of 62%.'

Line 483: "can see that": no subject in the sentence

OK

Lines 484-485: not consistent: "the reaction products generated is different" and "similar compounds seem to be formed"

**Response of authors**

**In the revised manuscript, the sentence has been eliminated**

**Referee#5**

General comments and main concerns:

The article "Reactivity study of 3,3-dimethylbutanal and 3,3-dimethylbutanone: Kinetic, reaction products, mechanisms and atmospheric implications" by Aranda et al., is written in line with several previous studies performed by this research group.

The manuscript consists of two parts related to kinetic and mechanistic investigations on the reaction of two carbonyls (one ketone and one aldehyde) with atmospheric oxidants (Cl, OH, and NO3). The manuscript comprises a first part including three kinetic studies of 33DMbutanal+Cl, and 33DMbutanone with CL and OH radicals and a second part dedicated to product studies performed with chlorine and hydroxyl radical for both carbonyl compounds and gas-phase product studies initiated by NO3 radicals for the reaction with the aldehyde.

The manuscript needs consistent English corrections and also needs to be restructured on the sections related to gas-phase product presentation. The structure of the manuscript is tangled and difficult to follow, especially for the gas-phase product section. I suggest a structure with subsections for results and discussion representative for each studied reaction.

**Response of authors**

**The authors thank you for dedicating your time to reviewing our work, and we greatly value to reviewing our work, and we consider it very valuable to have the opportunity to discuss the results. As we can confirm, the review of the manuscript has been very thorough, allowing us to correct not only typographical errors but also some data that had not been accurately transferred from the spreadsheets to the tables in the manuscript.**

**Below, we present our point-by-point response to your comments.**

**As you suggest the structure of reaction products has been restructured. Now in the revised manuscript the gas-phase product section has been divided in subsections for results and discussion representative for each studied reaction. This–represents a significant change from the original manuscript, as the figures/tables in both the manuscript and the supplementary material have been modified.**

One main concern is related to product studies where, despite the extensive degradation mechanisms shown in the article, the products tentatively quantified are formaldehyde, acetone, and 22DMpropanal. No other products could be really supported by the presented results.

The mechanisms should be redesigned in a much simpler form to emphasize the findings and to highlight the m/z for the compounds identified in the present study. The expected products in the dotted line in the mechanism should be supported by the identified m/z in the mass spectra. Please add in the article, supportive information from mass spectra corresponding to chromatogram peaks.

**Response of authors**

The authors consider that, although it has only been possible to quantify acetone, formaldehyde, 2,2-dimethylpropanal, nitrate compounds, and PAN using the FTIR system, the reaction products identified using GC-TOFMS are noteworthy. These products are proposed based on qualitative analysis of mass spectra obtained with field ionization (which provides the molecular ion) and the fragmentation pattern analysis of mass spectra acquired with electron impact ionization. The authors consider that the expected products, (showed in Figures 6 and 8 in the revised manuscript), are supported by the m/z values identified in the mass spectra. Tables 2S and 3S present the mass spectra of all major chromatographic peaks as suggested.

So, we consider that this work is particularly significant as it provides, for the first time, the mass spectra of these compounds.

Other concern is related to PAN formation and organic nitrate formation which are mainly suggested to be formed by generally accepted mechanisms due to the concentrations of the reactants involved in the present study. Please discuss the formation of this compound in terms of interferences due to the secondary sources of PAN which could be formed, for example, PAN formation from acetone degradation and huge NOx concentration. This source would explain also the delay in the formation of PAN.

**Response of authors**

**As can be seen in the concentration-time profiles and plots of the reaction products formed versus the consumption of the reactant, PAN is exclusively detected in the reaction of 33DMbutanone with Cl when there is no NO in the medium and the concentration of NO2 begins to be significant. If PAN formation were of secondary origin, for example, by the reaction of acetone, PAN should also have been detected in the experiments of the reaction of 33DMbutanal with Cl in the presence of NO. Therefore, we conclude that PAN is formed by the reaction of 33DMbutanone with Cl in the presence of NO2. In the revised manuscript the next paragraph has been included in lines 485-490 as follows:**

‘Considering that the formation of PAN can only be explained via channel II (hydrogen abstraction from the -CH$_3$ group of the tert-butyl group), which involves the decomposition of the initially formed alkoxy radical (2,2-dimethyl-3-oxobutan-1-yloxy radical), and that the plot of PAN concentration versus the variation in 33DMbutanone concentration shows a linear relationship with a slope close to one (indicating an estimated PAN yield of 100%, see Fig. 5S), it could be concluded that the percentage of Cl attack on the -CH$_3$ group of the tert-butyl group is nearly100%.’

**To explain the delay in the formation of PAN, a stated has been added in the revised manuscript in lines 372-373 as follows**

‘The profile of the nitrated compounds, especially PAN, shows a significant increase after 5 minutes of reaction, related to the rise in NO2 concentration in the medium after that time.’

The techniques employed in the present study can identify multiple other products present in the designed degradation mechanism. Could the authors reconsider the product formation interpreting the mass spectrum of each peak in their study? IR spectra did not show an OH absorption band around 3600 cm-1 according to the proposed alcohol products?

There are also reference spectra that could help to identify possible formation products (eg. Biacethyl, etc.).

A statement at line 592 indicates the formation of formic acid "which was not quantified". Why do not provide all the possible information in the article? I suggest discussing the residual spectrum in terms of remaining IR features that are not allocated to the known products.

**Response of authors**

**As previously mentioned, the assignment of reaction products in experiments conducted with the GC-TOFMS system has been based primarily on the analysis of mass spectra. Many of these compounds exhibit very similar IR bands that overlap, making their identification challenging. Additionally, it is believed that the quantities of these compounds are small, resulting in IR spectra with weak bands (very low absorbance).**

**However, a review of the residual IR spectra from the reactions of 33DMbutanal and 33DMbutanone reveals a discernible absorption band around 3600, 3000, 1770 and 1200-1000 cm$^{-1}$. (The IR band at 3600 cm-1 is not present in the reaction of 33DMbutanal with the NO$_3$ radical and 33DMbutanone with Cl + NO).**

[Figure]

The residual IR spectra have been compared with reference spectra of biacethyl in the case of the reactions with 33DMbutanone and with glyoxal for 33DMbutanal, with no characteristic IR bands of these compounds being clearly observed. Furthermore, as discussed in the manuscript, the formation of butane-2,3-dione (biacethyl) is less favored than the formation of acetone (lines 302 in the revised manuscript).

[Figure]

The residual spectra exhibit very low absorbance values, and the bands correspond to overlapping signals from different products. Therefore, we consider that these spectra are not sufficiently reliable to distinguish the reaction products.

Nevertheless, taking your suggestion into account, the following texts have been included in the revised manuscript:

**Lines 351-360**

'After removing the IR absorption bands of formaldehyde and acetone from the spectra in Fig. 2, the newly obtained residual spectra reveal the presence of IR absorption bands characteristic of various organic functional groups, such as carbonyl (-C(O) ~1745-1795 cm$^{-1}$) and hydroxyl (-OH, ~3600 cm$^{-1}$) (see Fig. 4S). As it can be seen, all the spectra show IR bands around 3725-3500 cm$^{-1}$, 3000-2750 cm$^{-1}$ and 1780 cm$^{-1}$, indicating the formation of common reaction products in the reactions of 33DMbutanone with Cl and OH. Other

different IR bands (1136, 1180, 1364 cm$^{-1}$) are also identified, which would suggest different reaction products. Additionally, the IR bands shown in Fig. 4S are consistent with the multifunctional products proposed in Scheme 1S. Confirmation of these compounds is not possible due to their unavailability as commercial standards and the absence of reference spectra in existing infrared databases. Furthermore, the low intensity of their IR bands, likely resulting from low concentrations in the medium, combined with band overlap, hinders their identification'.

**Lines 383-385**

Formic acid is mainly produced through secondary reactions when formaldehyde is present, as evidenced by typical secondary concentration profile time (see Fig. 9S).

**Lines 525-529**

'After removing the IR absorption bands of the identified products in Fig. 11S, the residual spectra reveal the presence of IR absorption bands characteristic of carbonyl (~1700-1790 cm$^{-1}$), hydroxyl (~3600 and 1033 cm$^{-1}$) and organic acid/acyl chloride compound (~1800 cm$^{-1}$) (Fig. 12S). The amplified residual spectra of Fig. 12S (Fig.13S) shows clearly some IR absorption bands that appear at the same wavenumber indicating common reaction products. The IR band at 1105 cm$^{-1}$ shown in Fig. 13S again indicates the formation of formic acid.'

Please include in the paper all the plots for all the reference compounds used in kinetic experiments for 33DMbutanal with Cl. The scientific community wants to see how well the linearity of the kinetic plots and which of the reference compounds worked better. Why the other kinetic plots are not properly represented?

**Response of authors**

**It is standard practice to include only an example of these plots in the manuscript and others in the supplementary material. However, following your suggestion, all the plots for all the reference compounds used in kinetic experiments have been included in the supplementary material of the revised manuscript.**

a)

[Figure]

b)

[Figure]

c)

[Figure]

'Figure 1S. Plot of Eq. (I) for the reaction of 33DMbutanal with Cl atoms using cyclohexane as reference compound (a) and for 33DMbutanone with Cl atoms (b) and OH radical (c) using different references compounds. In some cases, the plots have been displaced for clarity.'

To indicate the good fit of the experimental data, the following sentence has been included in the revised manuscript:

**Lines 183-186**

'Plot of Eq. (I) depicting the reaction of 33DMbutanal with Cl atoms, using cyclohexane as a reference, is presented in Fig. 1Sa. Additionally, the Fig. 1S includes the plots for the reaction of 33DMbutanone with Cl atoms and OH radicals, along with all the reference compounds used. The plots show a linear fit with $r^2 \geq 0.99$ and low origin intercept values, indicating the absence of secondary reactions.'

An important concern is related to the photolysis of 3,3-Dimethylbutanal. Tadic et al., found significant photolysis for 3,3-Dimethylbutanal in the spectral range used in the present study. How do the authors comment on the missing correction for the photolysis in their kinetic and product studies? How significant is the photolysis? There is a competition between photolytic lifetime and reactive lifetime, how do you comment on that?

**Effectively Tadic et al. (2012) conducted a study on the photolysis of 33DMbutanal using radiation in the range of 270 nm to 380 nm, observing that photolysis occurred. However, in our case, the radiation is limited to $\lambda >360$ nm. As demonstrated in the study by Lanza et al. (2008), the maximum absorption cross-section of 3-methylbutanal corresponds to 290 nm, decreasing sharply at wavelengths greater than 320 nm.**

[Figure]

**By analogy with 3-methylbutanal, 33DMbutanal is expected to exhibit similar behavior. This is the primary reason why, under our experimental conditions, 33DMbutanal does not undergo photolysis**

**I am sharing a screenshot below where it can be observed that the spectrum of 33BMbutanal at time zero and after 45 minutes of photolysis does not change by more than 3%. (subtraction factor 0.9931)**

[Figure]

**Therefore, we do not consider any further corrections to the manuscript necessary.**

Specific comments:

Could authors add information about the concentration of the radical precursors?

**Response of authors**

**As it is indicated in lines 141-146 of the revised manuscript, the concentration of atom/radical precursor is (in ppm) 17-22 for $Cl_2$, 16-20 for methyl nitrite, 30 for $H_2O_2$ and 14-25 for $N_2O_5$.**

Table 1 – 33DMbutanal+Cl – the error for the second exp with cyclohexane is wrong.

**Response of authors**

**Effectively the error is wrong. In the revised manuscript, the error has been corrected.**

Please give an explanation to "d" in the table.

**Response of authors**

**The table 1 has been modified in the revised manuscript and "d" is defined**

Please comment in the text for the difference of 25% between the kcarbonyl values for the reaction of 33DMbutanal+Cl.

**Response of authors**

**The authors consider this difference it is normal and it is not necessary to comment it in the text.**

**We really do not consider it necessary to make any additional comments in the text regarding this since this percentage can be considered as usual in the calculation of these rate constants and fallx within the range of experimental errors.**

Could you explain the lowest kcarbonyl values for the reaction of 33DMbutanone+OH using 1-butanol and cyclohexane as reference compounds?

The following rate constants were recommended by McGillen database:

1-butanol + OH: 9.14e-12 cm3molec-1s-1

2-methyl-2-butanol + OH: 3.42e-12 cm3molec-1s-1

2-propanol + OH: 5.24e-12 cm3molec-1s-1

Why the study does not use these values as the authors mentioned in the article but uses other values as the kOH = 9.8±2.0 for 1-butanol for example?

**Response of authors**

**The McGillen database 2020, has been reviewed, and indeed the recommended value for the rate constant of 1-butanol with OH is $9.14 \times 10^{-12}$ cm$^3$ molecule$^{-1}$ s$^{-1}$. The data has been recalculated using this new value, and the modified data is presented in Table 1. It can be verified that the final value of the rate constant for 33DMbutanone with OH remains unchanged. In the other cases, the recommended value has been used (2-methyl-2-butanol + OH: $3.42 \times 10^{-12}$ cm$^3$ molec$^{-1}$ s$^{-1}$ and 2-propanol + OH: $5.24 \times 10^{-12}$ cm$^3$ molec$^{-1}$ s$^{-1}$. See Table 1 of revised manuscript).**

Please comment in the text for the difference of 50% between the kcarbonyl values for the reaction of 33DMbutanone+OH. 0.96 and 1.92 are completely unrealistic for this study.

**Response of authors**

**Upon reviewing the data in Table 1, we identified an error in the reported rate coefficient for the reaction of 33DMbutanone with OH when 1-butanol was used as the reference compound. The corrected value was (1.80±0.37) instead of 0.96±0.11. Taking into account your suggestion to use the constant for 1-butanol as $9.14 \times 10^{-12}$, the averaged value is $1.68 \pm 0.26$ cm$^3$ molecule$^{-1}$ s$^{-1}$.**

**In the revised manuscript, the weighted average of k obtained for 33DMbutanone+OH (presented in table 1) is $(1.25 \pm 0.05) \times 10{-12}$ cm$^3$ molecule$^{-1}$ s$^{-1}$, closely aligns with the data reported by Mapelli et al. (2023) $(1.2 \pm 0.2) \times 10^{-12}$ cm$^3$ molecule$^{-1}$ s$^{-1}$ and the recommended value from McGillen (2020) $(1.21 \pm 0.05) \times 10^{-12}$ cm$^3$ molecule$^{-1}$ s$^{-1}$.**

**Due to this difference, four reference compounds have been used, whereas it is common practice to use only two or three. However, to ensure the accuracy of the constant value, all data obtained using four reference compounds are provided. The average value obtained is similar to that reported by Mapelli and to the value recommended by McGillen.**

**Therefore, the authors believe that this result provides a reliable rate coefficient for the reaction of 33DMbutanone with Cl and OH and consider it unnecessary to include any comment in the text regarding the different data obtained for the individual rate coefficients for the reaction of 33DMbutanone with OH.**

The reaction of acetylperoxy with HO2 could be better represented by a more recent paper: Winiberg, et al., 2016, Direct measurements of OH and other product yields from the HO2 + CH3C(O)O2 reaction, Atmos. Chem. Phys., 16, 4023–4042, https://doi.org/10.5194/acp-16-4023-2016, 2016. This could explain the formation of ozone, peroxides, and OH radicals. Please discuss the effect of OH radicals formed in the reaction with chlorine atoms on the formation products. The reactivity towards OH and chlorine could help with these discussions.

**Response of authors**

**Thank you for the information, this reference has been included in the revised manuscript. In the study of Winiberg, et al., 2016 the experiments are conducted in the absence of NO and the RO$_2$ radical is different from those initially generated in the reactions of 33DMbutanal and 33DMbutanone. In no case under experimental conditions without NO, such as the reactions of 33DMbutanone with Cl and OH or 33DMbutanal with Cl, ozone has been observed. Furthermore, while the formation of the OH radical could play a significant role in the reaction with Cl, the authors consider that, given the rate constant for OH is an order of magnitude lower than that for Cl atoms, the products identified in our experiments are primarily attributed to the reaction with Cl. In the reaction of Cl in the absence of NO, it is possible that RO2 radicals react with OH generated but it is not possible to confirm the reaction product generated (RO2OH) using FTIR by the lack of reference spectrum.**

The yields from nitrated gas-phase products are not very well represented. Could authors explain the curved shape?

**Response of authors**

**We do not understand what do you mean by "not well represented."**

**In figure 4 of revised version of manuscript, the up curved plot is indicating that there are additional reactions that form nitrated compound, and the linear plots indicate that the nitrated compound are formed as main products.**

**The yield of nitrated compounds and PAN in the case of 33DMbutanone reaction, was obtained from linear trend. In the revised manuscript this is indicated as follows**

**Lines 378-380**

'The yields of nitrated compounds were estimated from the slopes of the plots showing linear behavior (using the initial data) to avoid contributions from secondary chemistry. For PAN, the data used were from a Δ[33DMbutanone] of approximately 2 ppm (see Fig. 5S).'

Please give more information about the error calculation for kinetic results and product yields.

**Response of authors**

**The information about the error calculations for kinetic results is included in the foot of table 1 in the revised manuscript as follows:**

The total absolute error σ(kcarbonyl) is a combination of the statistical errors from the regression analysis (σslope) and the quoted error in the value of the rate coefficient of the

reference compound (σR). The final values of the rate coefficients and the associated error were calculated as weighted average (Colmenar et al., 2020).

**and the error calculation for product yields is now included in the foot of tables 3 and 4 as follows:**

'The quoted error in the individual yield (2σ) is two times the statistical errors from the regression analysis (2 × σslope). The quoted error in the average yield (2σ) is two times the Standard deviation (2 × σ).'

Minor comments:

Please consider ";" between the cited reference citation in the article text body.

OK

Please revise the way used to include the cited literature in the article text body.

OK

In the introduction section, the information about the formation of 33DMbutanone as a product from d 3,3-dimethyl-2-butanol degradation is presented a couple of times.

Line 104. Please use only one literature-cited reference for the experimental details found in the previous publication. Multiple citations, in this case five, lead to an increase in self-citations and this is unwanted.

OK

Please avoid commas for Figure 1 and all the kinetic figures that are included in the article body and supplement.

OK

Line 185. Please use the fundamental atmospheric chemistry reference literature (Finlayson-Pitts book for example) to support the attempt related to faster reactions with chlorine radicals.

**Response of authors**

**Some references have been included in line 205 of revised manuscript**
'(Mellouki et al., 2015; Calvert et al., 2011; Atkinson, 2007).

Line 260 Please mention which SAR approach the authors considered

**The SAR approach considered has been indicated in line 250-252 of the revised manuscript as follows:**

'SAR (Structure-Activity Relationship) predictive methods (Carter et al., 2021; Calvert et al., 2011; Jenkin et al., 2018; Kerdouci et al., 2014; Kwok and Atkinson, 1995).'

Referee #6

The authors Aranda et al. have conducted a series of experiments examining the oxidation of a pair of saturated, functionalized carbonyl molecules. The experiments cover the three major relevant atmospheric oxidants (OH, Cl, and NO3 radicals) and include both relative-rate measurements and studies on major product formation to inform chemical mechanisms. The overall significance of different reactions and products is discussed in the context of general atmospheric chemistry. The authors also highlight some current unknowns regarding the chemistry of aldehydes, a significant and reactive class of volatile organic compounds. The authors have put a substantial amount of effort into these tasks and the interpretation of their data. I have several questions related to the oxidation mechanisms discussed in the paper. If these and minor comments are addressed, I believe the paper should be accepted for publication

I suggest the authors take care when discussing the Cl-initiated oxidation of 3,3-dimethyl butanal. Previous works studying the oxidation of aldehydes (Iwasaki et al., 2008; Rayez et al., 2011; Singh et al., 2009) observed a strong deactivating effect of the CHO group that extended over multiple carbons and did not fit well with SAR predictions; Singh et al. (2009) wrote "the substituent factor for -CHO is significantly less than one, and that the group has a deactivating effect over several carbon atoms along the alkyl chain." It is not clear to me whether the work of Carter (2021) used by the authors for estimating functional group factors accurately reflects these prior works. I therefore suggest that the authors evaluate the prior references mentioned here (or others, as relevant) and determine whether the 0.4 substituent factor, and the discussion in sections 3.1-3.2 and the proposed chemistry in Figure 8, are reasonable. It may not be feasible with the current experimental data, but experimental verification of SAR predictions for Cl reaction with aldehydes would be beneficial in understanding Cl/aldehyde chemistry.

The authors discuss major RO2 radical reaction channels in section 3.2. Can the authors make any estimate of the relative prevalence or importance of the different RO2 radical reaction partners (RO2, HO2, OH/Cl) during these experiments? Without some sort of estimate of RO2 reaction branching ratios, it is difficult to determine the significance of the different identified products and their associated proposed formation pathways. To put this another way, how realistic are the reaction conditions and calculated product yields?

The authors mention the clear loss of some primary products during later experiment times (Figure 4). Can the authors provide an estimate of when secondary chemistry and product formation may become relevant during each experiment? This could include oxidation or photolysis reactions of primary products. It is not currently clear whether primary products may be more or less reactive or photolabile than the parent molecules, which complicates the interpretation of the yield results and the variety of products identified through FTIR and GC-MS analysis.

**Response of authors**

**The authors thank you for dedicating your time to reviewing our work, and we greatly value to reviewing our work, and we consider it very valuable to have the opportunity to discuss the results. As we can confirm, the review of the manuscript has been very thorough, allowing us to correct not only typographical errors but also some data that had not been accurately transferred from the spreadsheets to the tables in the manuscript.**

**Below, we present our point-by-point response to your comments.**

**NOTE: It is important to note that, based on the suggestion of Referee #5, the authors have decided to restructure the entire Results and Discussion section regarding the reaction products. This involves changing the figures in both the manuscript and the supplementary material.**

**Regarding to comment about the SAR predictions for Cl reaction with aldehydes, the authors have used the value reported by Carter et al. (2021), as it is a review up to 2020 and is therefore considered to have taken into account the studies on aldehydes mentioned (Iwasaki et al., 2008; Rayez et al., 2011; Singh et al., 2009). The fact that the experimentally obtained rate constant for 33DMbutanal with Cl (1.27x10$^{-10}$ cm$^3$ molecule$^{-1}$ s$^{-1}$) and the one estimated using the SAR method with a factor of 0.4 for the -CHO group (k$_{estimated}$ =1.36x10$^{-10}$ cm$^3$ molecule$^{-1}$ s$^{-1}$) are very similar suggests that the value of 0.4 used for -CHO is appropriate. On the other hand, based on the similarity between the estimated and experimental rate constants, the authors have proposed the reaction mechanism for 3,3-dimethylbutanal.**

**The following paragraphs have been modified in the revised manuscript**

**Lines 265-266**

'In all cases the estimated rate coefficients are very similar to the experimental values (see Table 1), indicating that the reactivity factors used for the estimations are well established'

**Regarding the estimation of the importance of different RO2 reactions, initially in the proposed reaction mechanisms, all possible reactions of RO2 were included based on general atmospheric reactivity (Finlayson-Pitts and Pitts, 2000, Atkinson 2007), as we had no information to rule out any pathway. In the revised version of the manuscript, the RO2 + HO2 reaction has been included, which was initially omitted, considering it to be significant only in polluted atmospheres and to avoid overly complex mechanisms that are difficult to follow. On the other hand, in the revised version of the manuscript (lines 299-303), it is indicated that, based on the estimated rate constants for isomerization and decomposition process of RO radical (Vereecken and Peeters (2009, 2010), some pathways of the mechanism are ruled out. In this way, a more simplified mechanism has been proposed for 3,3-dimethylbutanone. In the case of the mechanism for 3,3-dimethylbutanal, only the isomerization has been excluded.**

**The following paragraphs have been modified in the revised manuscript**

**Lines 288-311**

'It is well established (Atkinson, 2007) that alkyl radicals, formed in the initial step of these reactions, rapidly react with O2 to generate the corresponding peroxyradical (RO2·). These RO2 radicals can undergo various pathways. In the absence of NO, peroxyradicals primarily undergo two self-reaction processes: one leading to the formation of alkoxyradicals (RO2· + RO2· → 2RO· + O2), and the other producing two molecules, such as hydroxy compound and carbonyl compound (RO2· + RO2· → hydroxy compound + carbonyl compound + O2). Another significant process is the reaction of RO2 with OH radicals and HO$_2$ radicals that undergoes different pathways (Bottorff et al., 2023; Berndt et al., 2022; Fittschen, 2019; Jenkin et al., 2019; Berndt et al., 2018; Winiberg et al., 2016).'

In the presence of NO, the RO2 radical may react to form alkoxyradicals and NO2 (RO2· + NO· → RO· + NO2) or nitrated compounds (RONO2), and in presence of large concentration of NO2, RO2 generates peroxynitrated compounds (ROONO2) (pathway less favoured). Under typical tropospheric conditions, alkoxyradicals can react with molecular oxygen (O2), undergo unimolecular decomposition or isomerize. The reaction of RO· radicals with O2 is only possible if the carbon atom bearing the radical contains at least one hydrogen atom. Additionally, in the presence of NO and NO2, alkoxyradicals can also form nitrated compounds (Atkinson, 2007).

The rate coefficients for unimolecular decomposition and isomerization processes have been estimated in this work, using the method by Vereecken and Peeters (2009, 2010). In the case of 33DMbutanone, the rate coefficients for isomerization were much lower than those for decomposition making isomerization products insignificant. The estimated decomposition rate coefficient to form acetone ($1.7 \times 10^{12}$ s$^{-1}$) was much higher than to obtain butane-2,3-dione ($6 \times 10^{3}$ s$^{-1}$). For 33DMbutanal the decomposition rate is four times higher than that for the isomerization process.

Taking all the above into account, and to facilitate the qualitative analysis of reaction products from the reaction of 33DMbutanone and 33DMbutanal with atmospheric oxidants, proposed reaction mechanisms are depicted in Schemes 1S and 2S in the supplementary material. The RO2 + HO2 reactions are excluded to avoid further complicating the mechanism for 33DMbutanal, as they are significant only in the absence of NOx, that is, in remote unpolluted atmospheres.'

**Regarding to comment about secondary chemistry in the revised manuscript the next paragraphs have modified in order to give more information about the loss processes of some primary products and when secondary chemistry and product formation may be relevant.**

**Lines 370-381**

'The trend of the acetone and HCHO profiles indicate that they are primary products, although the concentration of HCHO starts to decrease after 20 min of reaction, possibly due to secondary chemical reactions such as photolysis or by the reaction with the main oxidant. The profile of the nitrated compounds, especially PAN, shows a significant increase after 5 minutes of reaction, related to the rise in NO2 concentration in the medium after that time.

Fig. 4 shows an example of yield plots for the reactions of 33DMbutanone with Cl atoms in the presence of NO.

The yields of nitrated compounds were estimated from the slopes of the plots showing linear behavior (using the initial data) to avoid contributions from secondary chemistry. For PAN, the data used were from a Δ[33DMbutanone] of approximately 2 ppm (see Fig. 5S). In the case of HCHO, the yield has been recalculated using the formalism published by Tuazon et al., 1986.'

**lines 530-533**

'The trends of the concentration-time profiles of the quantified products for the reaction of 33DMbutanal with Cl atoms (Fig. 14S), suggest that they are primary products in the early stages (linear trend). However, it is observed that after a certain reaction time, the

concentrations of 22DMpropanal and HCHO decrease. This decrease could be attributed to loss processes involving reactions with Cl atoms and/or photolysis.'

**lines 541-543.**

'The concentration plots of the products formed against the variation 33DMbutanal, used to obtain the yields, are shown in Fig. 15S. For of HCHO and 22DMpropanal, where the concentration decreases by react with the main oxidant, the yield has been recalculated using the formalism by Tuazon et al., 1986.'

Comments Minor Comments

General: I found some discussion of parent and product structures difficult to follow. I would suggest showing structures of parent molecules in Table 2 somewhere in the text as well as adding a label or number to some products or intermediates in Figures 7 and 8 to make referencing these structures within the text more clear.

**Response of authors**

**Thank you for your suggestion. The structures are included in the tables of the supplementary material, where the names are also provided. To avoid overly crowded and large tables in the manuscript, the authors believe it is not necessary to include this information in Table 2 and Table 3. The names have been included in Figures 7 and 8 (Figures 6 and 8 in the revised manuscript).**

Line 201: might the decrease in reaction rate when moving to 3,3-dimethyl butanal be due to the fact that many of the abstractable H are now primary, with a lower inherent reaction rate towards OH and Cl, rather than steric factors?

**Response of authors**

**The authors agree with you, and indeed the decrease in the rate constant may also be related to the fact that many of the abstractable hydrogens are now primary. However, in the case of reactions with OH, where the structure of the compound has a much greater influence on the reaction rate than in reactions with Cl, the steric factor of an additional methyl group must be important.**

**Taking into account your suggestion, the following sentence has been included in the revised manuscript as follows:**

**Lines 223-224.**

'The decrease in the rate coefficient may also be related to the fact that many of the abstractable hydrogens are now primary'

Line 311, Figure 7, and elsewhere: I'm not familiar with the "alkoxy nitrate" compounds the authors note as forming from reaction of NO or NO2 with alkoxy radicals (e.g., (Atkinson, 2007)). Can the authors provide more background on the formation of these molecules?

**Response of authors**

**In the new figures of the revised manuscript, examples of the structures of these alkoxy nitrates (RONO2 or RONO) have been included.**

Lines 489-491: Atkinson (2007) also wrote that "this ''prompt'' decomposition of alkoxy radicals formed from the exothermic RO2 + NO reaction appears to be important for alkoxy radicals with a barrier to decomposition of approximately 9 kcal mol$^{-1}$ or less, with prompt decomposition being unimportant for alkoxy radicals with higher barriers to decomposition." If the authors assert that alkoxy radicals formed in this work undergo "prompt" decomposition, they must present work supporting this conclusion (for example, utilizing the methods proposed in the already cited work of Vereecken and Peeters, 2009).

**Response of authors**

**This comment is responded above. This is the paragraphs that have been modified in the revised manuscript:**

'The rate coefficients for unimolecular decomposition and isomerization processes have been estimated in this work, using the method by Vereecken and Peeters (2009, 2010). In the case of 33DMbutanone, the rate coefficients for isomerization were much lower than those for decomposition making isomerization products insignificant. The estimated decomposition rate coefficient to form acetone ($1.7 \times 10^{12}$ s$^{-1}$) was much higher than to obtain butane-2,3-dione ($6 \times 10^3$ s$^{-1}$). For 33DMbutanal the decomposition rate is four times higher than that for the isomerization process.'

Line 493: RO2 + Cl à RO is also a potential reaction (Maricq et al., 1994) that may be relevant to consider.

**Response of authors**

**We consider the RO2 + Cl reaction forming RO to be irrelevant for the reactions of 3,3-dimethylbutanone and 3,3-dimethylbutanal, as the rate coefficients obtained in this our work are higher than or of the same order as the RO2 + Cl reaction (as reported in the study by Maricq et al., 1994). Furthermore, the formation of ClO/ClOOCl does not observe.**

Figure 8: For channel III, might intramolecular hydrogen shifts occur from wither the initial RO2 or RO radicals? Based on prior estimates (Vereecken & Nozière, 2020; Vereecken & Peeters, 2010), these aldehydic H-shifts are expected to be relatively fast. Might such reactions contribute to the larger variety of product structures observed for Cl reaction compared to OH, given the greater importance of channel III for Cl reaction?

**Response of authors**

**Indeed, these intramolecular hydrogen shifts from the initial RO2 or RO radicals could occur, but only the reactions that could explain the identified products (based on the IR spectra or their MS) in the different experiments are included. Not all reactions are included to avoid complex mechanisms.**

**Effectively the greater variety of products observed in the reaction with Cl compared to the reaction with OH can be attributed to the ability of Cl to initiate reactions through all three possible channels, leading to a wider range of products. In contrast, the reaction with OH is limited to proceeding predominantly through a single channel, resulting in fewer product variations. No additional explanation has been included in the revised manuscript.**

Technical comments

Line 265: "mayor" à major

OK

Line 561: "...spectrum of 22DMpropanoic [acid]."

OK

References

Atkinson, R. (2007). Rate constants for the atmospheric reactions of alkoxy radicals: An updated estimation method. *Atmospheric Environment*, *41*(38), 8468–8485. https://doi.org/10.1016/j.atmosenv.2007.07.002

Iwasaki, E., Nakayama, T., Matsumi, Y., Takahashi, K., Wallington, T. J., Hurley, M. D., & Kaiser, E. W. (2008). Kinetics and Mechanism of the Reaction of Chlorine Atoms with n -Pentanal. *The Journal of Physical Chemistry A*, *112*(8), 1741–1746. https://doi.org/10.1021/jp077525z

Maricq, M. M., Szente, J. J., Kaiser, E. W., & Shi, J. (1994). Reaction of Chlorine Atoms with Methylperoxy and Ethylperoxy Radicals. *The Journal of Physical Chemistry*, *98*(8), 2083–2089. https://doi.org/10.1021/j100059a017

Rayez, M. T., Rayez, J. C., & Villenave, E. (2011). Theoretical approach of the mechanism of the reactions of chlorine atoms with aliphatic aldehydes. *Computational and Theoretical Chemistry*, *965*(2–3), 321–327. https://doi.org/10.1016/j.comptc.2010.11.025

Singh, S., Hernandez, S., Ibarra, Y., & Hasson, A. S. (2009). Kinetics and mechanism of the reactions of n -butanal and n -pentanal with chlorine atoms. *International Journal of Chemical Kinetics*, *41*(2), 133–141. https://doi.org/10.1002/kin.20383

Vereecken, L., & Nozière, B. (2020). H migration in peroxy radicals under atmospheric conditions. *Atmospheric Chemistry and Physics*, *20*(12), 7429–7458. https://doi.org/10.5194/acp-20-7429-2020

Vereecken, L., & Peeters, J. (2010). A structure-activity relationship for the rate coefficient of H-migration in substituted alkoxy radicals. *Physical Chemistry Chemical Physics*, *12*(39), 12608–12620. https://doi.org/10.1039/c0cp00387e

**Finally, we would like to thank the reviewers for the thorough work done to improve the submitted manuscript. With your suggestions and corrections, the quality is clearly higher.**

---

## Author Response (AR2)

egusphere-2024-3241

Title: Reactivity study of 3,3-dimethylbutanal and 3,3-dimethylbutanone: Kinetic, reaction products, mechanisms and atmospheric implications

Author(s): Inmaculada Aranda et al.

Public justification (visible to the public if the article is accepted and published):

Dear Dr Pilar Martin,

Thank you for providing a revised document with your responses to the reviewers' comments.

As you can see below, the reviewers still feel that the manuscript requires substantial improvement.

**Report# 2:** The manuscript improved addressing the provided suggestions and comments. A significant part of the manuscript has been restructured. This represents a substantial change from the original manuscript. The results and discussion section adds new information on the interpretation of FTIR and MS data. Proofs of the IR and MS data were included in the supplementary information files. Also, the kinetic data has been added to the SI. However, some minor revisions need to be applied before publication.

**Question:** Please add a statement in the text to explain the scattering of points in Fig S1 c.

**Response of author:** A paragraph has been added in the manuscript to explain the scattering of points in Fig S1 c. Although we consider that the data presented in Fig. S1c. do not show a greater dispersion than usual in this type of kinetic studies with slow reaction rates, the presence of methyl nitrite could also affect the analysis. It should be noted that the values of the rate constants presented in Table 1 for each reference compound correspond to the average value of the constants obtained in each of the experiments performed, but Graph 1S shows all the data together as an example, and although it shows dispersion for the reasons previously mentioned, it is not the graph used to calculate the rate constant. In any case, in accordance with your suggestion, we have decided to include a paragraph in the article, which is the following:

Pag 6, line 187: *"the dispersion of the data observed in the ·OH reaction with 33DMbutanone (Fig. 1Sc) is due to the complexity of the analysis, especially for slow reactions, and the overlapping of the bands of the precursor (methyl nitrite)."*

**Question:** There is still unresolved concern related to the photolysis of 33DMbutanal. The answer is not accepted in the way presented by the authors, however, the test presented by the authors could be mentioned in the text to explain somehow that even though there is a possibility of photolysis, this was tested and found not to affect the results. The authors mentioned "radiation emitted by actinic lamps ($\lambda$max = 360 nm)", and this is in contradiction with the answer "the radiation is limited to $\lambda$ >360 nm". Actinic lamps emit light in a range from around 290 to 450 nm with a maximum of 360nm. There

is a range of light that could induce the photodegradation of 33DMbutanal. Tadic et al., suggested in their article that "It is obvious that in case of … 3,3-dimethylbutanal photolytic and reactive lifetimes are very similar, and that those two removal processes compete among each other".

**Response to the comments:** Firstly, respecting to: 'Actinic lamps emit light in a range from around 290 to 450 nm with a maximum of 360nm. There is a range of light that could induce the photodegradation of 33DMbutanal. Tadic et al., suggested in their article that "It is obvious that in case of … 3,3-dimethylbutanal photolytic and reactive lifetimes are very similar, and that those two removal processes compete among each other'.

We must answer that **this is not the case in our experiments.** As demonstrated by the emission spectrum of the lamps used (Philips Actinic BL TL-K 40W/10-R, Wavelength Range: 340-400 nm, Peak at 365 nm, see below) and the results of our photolysis experiments, 3,3-dimethylbutanal does not undergo photolysis under our experimental conditions. That is, the radiation used to generate atomic chlorine and the OH radical does not induce photolysis of 3,3-dimethylbutanal as it has been explicated in the manuscript. Naturally, when determining the atmospheric lifetime to evaluate its implications, we must account for the fact that the radiation reaching the troposphere is energetic enough for the photolysis process to be comparable to the loss caused by the reaction with the OH radical, as explained in the manuscript.

We show here the emission Spectrum of the lamps used in our experiments. Philips Actinic BL TL-K 40W/10-R. Wavelength Range: 340-400 nm. Peak at 365 nm

[Figure]

As indicated in the manuscript (line 168-170), the possible photolysis of 3,3-dimethylbutanal is negligible under the conditions used in our experiments. Therefore, we do not consider it necessary to add more information on this matter in the manuscript.

**Authors' response to the comment '**The authors mentioned "radiation emitted by actinic lamps ($\lambda$max = 360 nm)", and this is in contradiction with the answer "the radiation is limited to $\lambda > 360$ nm'.

**Answer:** The previous response we made to the referee could not be entirely appropriate, and we apologize for that. In fact, we did not mean to imply that the radiation was limited to wavelengths greater than 360 nm. As confirmed by the emission spectrum of **our lamps** (attached photo), the emission range is 340-400 nm, with a peak wavelength of 365 nm. To be rigorous, we will also change the maximum value from 360 nm to 365 nm in the manuscript (line 124).

**-Question:** Another unresolved comment: "Please comment in the text for the difference of 25% between the kcarbonyl values for the reaction of 33DMbutanal+Cl." This aspect is even more annoying for the reaction with OH radical where the difference is dropping to 50%. Please add a statement in the text to explain such scattering.

**Response of authors:** We have included a short explanation of the fact (line 199). "*Upon examining the rate coefficients and their associated uncertainties, it is evident that most of the experimental values fall within the expected error margins. Furthermore, the 50% difference observed between the rate coefficient values obtained for the reaction of 33DMbutanone with the OH radical is considered usual, given the level of analytical complexity, particularly in the case of slow reactions*". This is because variations in determining the subtraction factor significantly influence the constant's value.

Thus, this can be verified in the following graphs, which show the error range of the obtained values. These graphs are not included in the manuscript.

[Figure]

[Figure]

[Figure]

[Figure]

**Report 3:** There remain inconsistencies in presentation (see below). Some of the kinetic data remains unusually scattered, though the presentation and explanatory text around these has been improved. Also heartening to see the products sections has been substantially revised.

**Response of authors.** As previously responded to Report #2, We have included a short explanation of the fact (line 199): "*Upon examining the rate coefficients and their associated uncertainties, it is evident that most of the experimental values fall within the expected error margins. Furthermore, the 50% difference observed between the rate coefficient values obtained for the reaction of 33DMbutanone with the OH radical is considered usual, given the level of analytical complexity, particularly in the case of slow reactions*".

**Question:** There remain inconsistencies in presentation throughout the manuscript, with muddling of units, parentheses etc., and whether dots are used to denote radical species or not.

**Response of authors.** The manuscript has been revised again eliminating the inconsistencies, muddling of units, denoting radical species, parentheses etc.,

**Report #1**: The revised manuscript is technically sound, but the reason the author chose 3,3-dimethylbutanal and 3,3-dimethylbutanone to research is still unclear, as the authors stated these compounds are not emitted in large quantities or not measured in the atmosphere, so if these compounds could play an important role in the tropospheric chemistry? So the related description in the manuscript should be revised.

**Response of authors**

The authors firmly believe that the work carried out is fully justified, as explained in the introduction. These two compounds have been identified as reaction products in the atmospheric degradation of alcohols (Colmenar et al., 2020).

*Colmenar, I., Martín, P., Cabanas, B., Salgado, S., Tapia, A., Aranda, I.: Atmospheric fate of a series of saturated alcohols: kinetic and mechanistic study, Atmos. Chem. Phys. 20, 699–720. https://doi.org/10.5194/acp-20-699-2020, 2020.*

Additionally, 33DMButanal could be an intermediate in the synthesis of neotame, a sweetener. Industrially, 33DMButanone is produced for use in fungicides, herbicides, and pesticides and could also serve as a solvent substitute for hazardous solvents like toluene, due to its low toxicity and good solvation characteristics.

We have included this sentence in the manuscript: (line 60) "*Although there are no atmospheric concentration measurements for these compounds, their current and potential uses justify the need to understand their atmospheric reactivity and degradation processes.*"

Indeed, several studies have already addressed the determination of rate constants with OH and $NO_3$ radicals and photolysis under atmospheric conditions. Our work further contributes new data on the reactivity not only with these oxidants ·OH and $NO_3$· but also with Cl atoms. Moreover, the research presented in this article helps to better understand the atmospheric chemistry of carbonyl compounds in general.

Therefore, we do not consider it necessary to introduce more modifications into the article.

Finally, we thank the reviewers for their thorough work, which undoubtedly further improves this article.